# Control of feeding by a bottom-up midbrain-subthalamic pathway

Fernando M. C. V. Reis[1] ✉, Sandra Maesta-Pereira [1], Matthias Ollivier[2], Peter J. Schuette[1], Ekayana Sethi[1], Blake A. Miranda[1], Emily Iniguez[1], Meghmik Chakerian[1], Eric Vaughn[3], Megha Sehgal[4], Darren C. T. Nguyen[1], Faith T. H. Yuan[1], Anita Torossian[1], Juliane M. Ikebara[5], Alexandre H. Kihara [5], Alcino J. Silva[1,4,6], Jonathan C. Kao[7], Baljit S. Khakh [2] & Avishek Adhikari [1] ✉

Investigative exploration and foraging leading to food consumption have vital importance, but are not well-understood. Since GABAergic inputs to the lateral and ventrolateral periaqueductal gray (l/vlPAG) control such behaviors, we dissected the role of vgat-expressing GABAergic l/vlPAG cells in exploration, foraging and hunting. Here, we show that in mice vgat l/vlPAG cells encode approach to food and consumption of both live prey and non-prey foods. The activity of these cells is necessary and sufficient for inducing food-seeking leading to subsequent consumption. Activation of vgat l/vlPAG cells produces exploratory foraging and compulsive eating without altering defensive behaviors. Moreover, l/vlPAG vgat cells are bidirectionally interconnected to several feeding, exploration and investigation nodes, including the zona incerta. Remarkably, the vgat l/vlPAG projection to the zona incerta bidirectionally controls approach towards food leading to consumption. These data indicate the PAG is not only a final downstream target of top-down exploration and foraging-related inputs, but that it also influences these behaviors through a bottom-up pathway.

Feeding is one of the most important aspects of an animal's life. Searching for food, animals must conduct careful exploration to gain access to caloric resources and to become familiar with their environment. Such exploratory activities are composed of a broad range of behaviors that include approach to novel stimuli, pursuit of prey and investigation of the environmental layout[1–4]. These processes are thus a critical feature of hunting for prey and also for finding caloric resources.

The neural basis for this vital behavior is not well-understood, but such investigative and exploratory curiosity can be reduced by inactivation of zona incerta (ZI) neurons[5,6]. Furthermore, activation of GABAergic ZI cells causes both exploratory approach and curiosity towards novelty, predation and eating of non-prey foods (i.e., high fat food)[7], indicating that the circuits controlling exploration and food consumption[8] may be linked. Another circuit that controls similar exploratory investigation and hunting is the medial preoptic area, as activation of camk2a cells from this region causes pursuit and following of both crickets and even of inedible objects[9]. Both GABAergic ZI cells and preoptic area camk2a cells project to the midbrain periaqueductal gray (PAG) and activation of both of these projections induces cricket hunting in mice[7,9]. Activation of the camk2a preoptic projection to the PAG also elicits following of objects and crickets,

[1]Department of Psychology, University of California, Los Angeles, Los Angeles, CA 90095, USA. [2]Department of Physiology, University of California, Los Angeles, Los Angeles, CA 90095, USA. [3]Department of Molecular and Cellular Biology, Harvard University, Cambridge, MA 02138, USA. [4]Department of Neurobiology, University of California, Los Angeles, Los Angeles, USA. [5]Centro de Matemática, Computação e Cognição, Universidade Federal do ABC, São Bernardo do Campo, SP 09606-070, Brazil. [6]Department of Psychiatry & Biobehavioral Sciences, University of California, Los Angeles, Los Angeles, USA. [7]Department of Electrical and Computer Engineering, University of California, Los Angeles, Los Angeles, CA 90095, USA. ✉e-mail: freis@ucla.edu; avi@psych.ucla.edu

suggesting there are local PAG cells that can trigger pursuit of prey, foraging and environmental investigative exploration[9]. Moreover, increases in cricket hunting were observed following activation of PAG inputs originating in the lateral hypothalamus[7,9], the bed nucleus of the stria terminalis[10], the central amygdala nucleus[11] and the basal forebrain[12]. These data suggested that the PAG is a common downstream target of exploratory foraging and hunting circuits, and suggest the existence of PAG cells that induce food-seeking. Indeed, to this end it has been shown that pharmacological inactivation of the PAG decreases food approach and consumption[13].

Activation of diverse upstream PAG inputs from higher brain regions induce hunting[7,9,11,12,14], but it is unknown if bottom-up feeding circuits originating in the PAG play a role in this behavior. The PAG is also known to affect higher brain regions in the context of defensive behaviors with bottom-up (midbrain to forebrain) projections[15,16]. For example, the PAG provides a teaching signal to the amygdala to provide CS-US associations during fear conditioning[17]. Additionally, amygdala lesions completely blocked fleeing responses evoked by dorsal PAG stimulation[18] suggesting that the basolateral amygdala is downstream of the dorsal PAG in mediating such behaviors. These data indicate that the PAG may also influence hunting and foraging by affecting higher brain regions, though such a mechanism has not been described. It is reasonable that the elaborate display of PAG-evoked behavioral responses would be organized by forebrain circuits[15,19]. In addition, the lateral PAG is also well-positioned to influence reward-seeking through its projections to forebrain sites such as the lateral hypothalamic area[20] or the subthalamic ZI[19,21]. Considering this view, and taking into account that GABAergic PAG activity has been linked to hunting, we hypothesized that these cells control not only hunting, but general exploratory foraging behavior, pursuit of prey and objects as well as approach towards non-prey food sources through a bottom-up projection.

Here, we show that the activity of vgat-expressing GABAergic l/vlPAG cells encode approach towards and consumption of prey and non-prey food. The activity of these cells also bidirectionally controls appetitive approach drive for both food types in mice. Furthermore, activation of l/vlPAG vgat cells induces a wide range of appetitive-related behaviors, such as exploratory foraging and compulsive eating without altering innate defensive behaviors. These data, along with the higher activity displayed by l/vlPAG vgat cells during approach to food indicate an important role for this population in foraging-related approach towards food. We also show that l/vlPAG vgat cells receive inputs from numerous known feeding and exploration nodes, and they project to regions that control eating as well, such as the ZI. Intriguingly, activity in the l/vlPAG vgat projection to the ZI is necessary and sufficient to induce hunting and consumption of non-prey food. Taken together, our data indicate that common pathways control exploratory foraging and pursuit of prey and non-prey foods and that midbrain circuits can influence this vital behavior by acting on higher brain regions.

## Results

### l/vlPAG vgat cells represent approach and consumption of food

To investigate how l/vlPAG vgat cells encode exploratory approach towards food sources and eating, we injected the vector AAV-DIO-GCaMP6s in the l/vlPAG of vgat-Cre mice to express the genetically encoded calcium indicator GCaMP6s in l/vlPAG vgat cells. This approach allowed us to obtain calcium transients from these cells through implanted miniaturized microscopes (Fig. 1a) during exposure to live crickets and walnuts (Fig. 1b). Eating onset aligned calcium activity shows that cells are often more active prior to eating both food sources (Fig. 1c). To verify if l/vlPAG vgat cells encoded relevant behaviors in this assay, we performed principal component analysis and determined that commonly occurring behaviors, such as approach to food, eating and rearing, appear predominantly in discrete clusters in principal component space (Fig. 1d, e). The clustering silhouette score, measuring quality, is higher than zero for all mice in both assays (Fig. 1d, e), was higher than expected by chance for each individual mouse (Supplementary Fig. 1) and did not correlate with speed (Supplementary Fig. 2), suggesting the specific encoding of feeding-related processes in vgat l/vlPAG cells. Supporting this idea, it was possible to use five-fold cross validated multinomial logistic regression to decode approach to food and eating behaviors from l/vlPAG vgat cell activity above chance levels (Fig. 1f). We next explored if a conserved pattern of neural activation was used to encode these behaviors across cricket and walnut assays. To do so, we co-registered cells recorded in both behavioral assays (Fig. 1g and Supplementary Fig. 3). We then plotted the data in n-dimensional space, where n is the number of neurons, and quantified the degree of clustering of a given behavior across both assays. A scheme illustrating this procedure is shown in Fig. 1h. The left graph shows the eating time points plotted for n = 3 neurons. Dark blue and light blue points correspond to data from the two assays (walnut and cricket). Note that the dark and light blue time points occupy a similar space, and are thus highly overlapping. Consequently, the average distance between walnut eating and cricket eating points is low. Conversely, in the right graph in Fig. 1h the approach points from walnut and cricket assay are more well-separated from each other, and thus the average distance between points from the two assays is higher. Thus, in this analysis, a higher distance between points from the two assays indicates that a particular behavior has a more independent and less conserved representation across the two assays. The actual experimentally-observed data is shown in Fig. 1i. Note that the distance between food approach points is higher than that of eating points. We also analyzed data from both assays in principal component space (Fig. 1j), and show that the average distance between approach cluster points across walnut and cricket assays is greater than that of eating cluster points. Finally, the mean Euclidean distance between n-dimensional behavioral samples and their cluster center across coregistered cells from the cricket and walnut assays is smaller for the eating cluster than the approach cluster, indicating a greater conservation of neural representation for eating than approaching food (Fig. 1k). Taken together, these data show that, both in n-dimensional neuronal space as well as in principal component space, the neural encoding of eating behavior (consummatory) has a more well-conserved representation across assays than the representation of the more variable approach behavior (appetitive) to food.

### l/vlPAG vgat cells show conserved encoding of consumption across prey and non-prey foods

We next investigated the temporal dynamics of l/vlPAG vgat calcium traces during the cricket and walnut assays. Mice were exposed to the walnut assay once and to the cricket assay twice (cricket 1 and 2) (Fig. 2a). Example activity traces show that these cells are more active prior to eating than during eating (indicated by shaded blue boxes in Fig. 2b). Averaged group data show that neural activity was consistently higher prior to eating onset in both assays (Fig. 2c, d). Accordingly, there were more negatively modulated than positively modulated cells during eating ('eat -' and 'eat +' cells, respectively, in Fig. 2e, f). We then evaluated if vgat cells show consistent eating-related modulation across assays (see example co-registration across days for cricket 1 and cricket 2 assays in Fig. 2g). We used a generalized linear model (GLM) to calculate the eating weight for each individual cell. The example cell in Fig. 2h shows a negative weight of similar magnitude across assays, indicating that l/vlPAG vgat cells may show conserved eating-related modulation. Indeed, eating GLM weights for the entire population of recorded co-registered cells were significantly correlated both across cricket 1 and cricket 2 exposures as well as across cricket and walnut assays (Fig. 2i). Approach and eating GLM weights were not correlated (Supplementary Fig. 2),

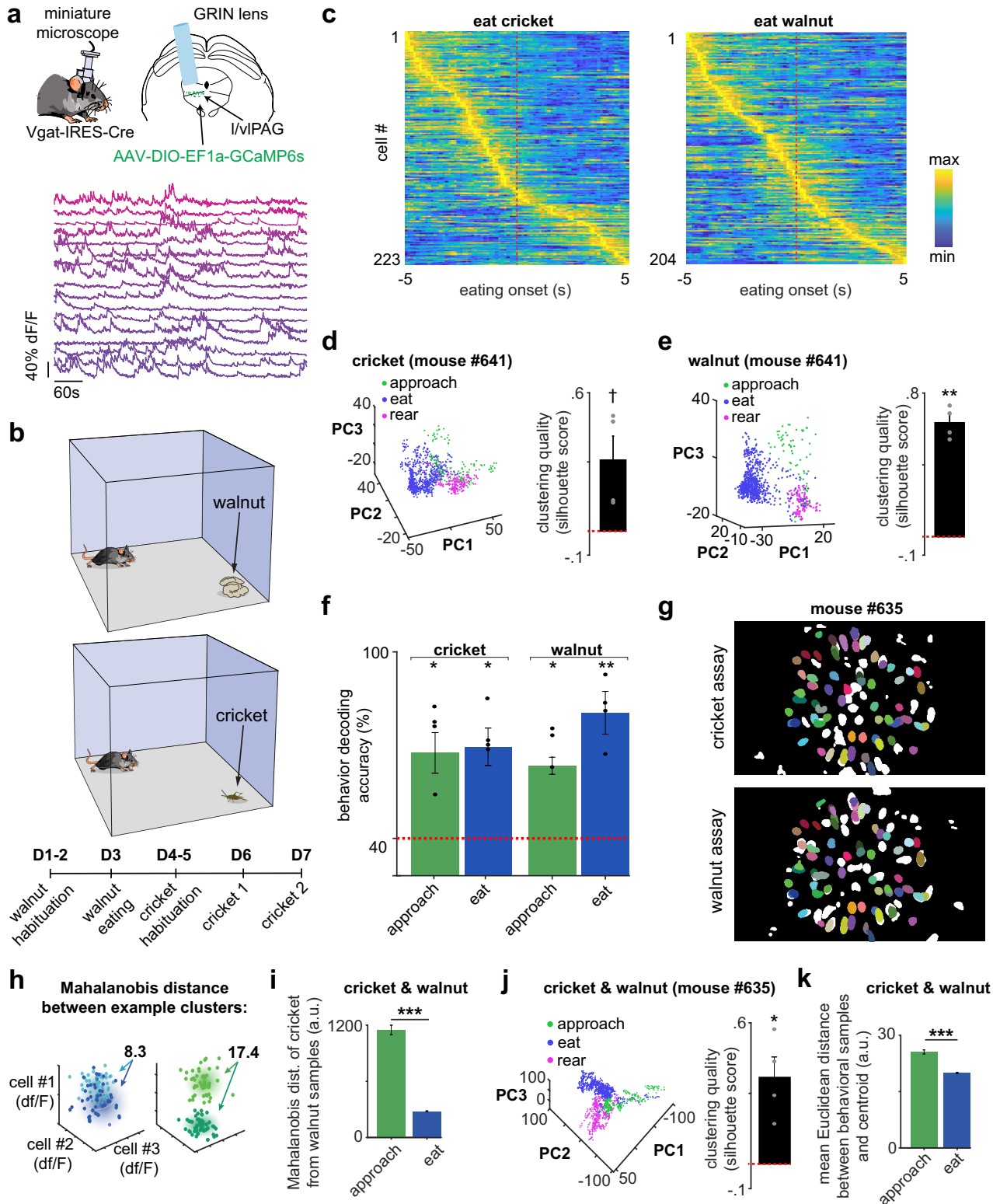

indicating that ensembles encode these behaviors independently. To further characterize this result, we measured the decrease in df/F during eating for each cell across assays (calculated as post-eating df/F - pre-eating df/F). The example cell in Fig. 2j shows a similar decrease in activity following eating onset. This effect was also observed across the population of recorded cells. Cells classified as negatively modulated eat- cells in the walnut experiment showed a decrease in activity during eating in both walnut and cricket 1 assays. Similarly, cricket 1 eat- cells also showed less activation during consumption of

food in the cricket 2 assay (Fig. 2k). We also observed that cricket 1 eat- cells showed a similar pattern of neural activation in cricket 2 during eating (Fig. 2l). Indeed, the activity profiles of co-registered eat- and of eat+ cells in cricket 1 and cricket 2 were highly positively correlated (Fig. 2l). Performing the same analysis across walnut and cricket assays revealed that eat- cells also showed highly correlated activity profiles while eating these two different types of food. Similar results were also seen after performing the same analysis using walnut and cricket 2 assays (Supplementary Fig. 4). Interestingly, eat+ activity

**Fig. 1 | Vgat l/vlPAG cells encode approach to food and eating. a** Surgery for recording l/vlPAG vgat calcium transients (top). Example l/vlPAG vgat traces (bottom). **b** Order of assays. **c** L/vlPAG vgat activity centered at eating onset (cricket assay $n = 223$ neurons; walnut assay $n = 204$ neurons). **d** Representation of behaviors in principal component space (PC) during cricket hunting (see example on left). Clustering quality was measured by silhouette score, which was higher than the chance level of zero (dotted red line) ($n = 4$ mice; one-sample two-tailed $t$-test, t-statistic = 2.79, $p = 0.059$). **e** Same as **d**, but for walnut ($n = 4$ mice; one-one-sample two-tailed $t$-test, t-statistic = 14.37, $p = 0.001$). **f** Behaviors were decoded above chance (red dotted line) ($n = 4$ mice; one-sample two-tailed $t$-test, cricket t-statistics: approach = 3.87, $p = 0.031$; eat = 5.59, $p = 0.011$; walnut t-statistics: approach = 4.75, $p = 0.018$, eat = 7.53, $p = 0.005$). **g** Cells co-registered across assays. **h** Mahalanobis distance between points from two clusters that display higher overlap (left panel with light and dark blue points) and two clusters that are well-separated (right panel with light and dark green points). **i** Mahalanobis distance

between approach and eating clusters across the cricket and walnut assays in n-dimensions ($n = $ # of co-registered neurons). The distance between eating clusters across assays is smaller than the distance between approach clusters, indicating the representation of eating is more conserved than food approach (approach sample $n = 527$ time points, eat sample $n = 1958$ time points; two-tailed Wilcoxon rank-sum test, z-score = 22.88, $p < 0.001$). **j** Same as **d**, but for co-registered cells, showing conserved representation of behaviors across assays ($n = 4$ mice; one-one-sample two-tailed $t$-test, t-statistic = 4.22, $p = 0.024$). **k** The distance between individual points and cluster center is smaller in the eating cluster than the approach cluster, indicating the representation of eating is more conserved across assays (approach sample $n = 1049$ time points, eat sample $n = 5492$ time points; two-tailed Wilcoxon rank-sum two-tailed test, z-score = 10.35, $p < 0.001$). ***$p < 0.001$, **$p < 0.01$, *$p < 0.05$, †$p = 0.059$. Data are presented as mean values +/- SEM. Source data are provided as a Source Data File.

profiles were not correlated across walnut and cricket assays, showing that the negative modulation during the consummatory phase is well-conserved across different assays, while positive modulation during eating is specific to the consummatory phase of each food type (Fig. 2m).

## l/vlPAG vgat cell activity is necessary and sufficient for food foraging and consumption

In order to investigate if l/vlPAG vgat cell activity is necessary for appetitive drive and food consumption, we optogenetically inhibited these cells during cricket and walnut exposure. To do so, we expressed the inhibitory opsin Arch in these cells (Fig. 3a, b). Optogenetic inhibition of l/vlPAG vgat cells reduced walnut consumption and successful predation, and also increased the latency to attack the cricket (Fig. 3c, d). We also showed that optogenetic excitation of ChR2-expressing l/vlPAG vgat cells produces the opposite effect. This manipulation increased both hunting and food consumption, decreased the latency to eat in both assays (Fig. 3e, f) and was not influenced by sex differences (Supplementary Fig. 5). Activity of l/vlPAG vgat cells is higher during exploration and approach to food, and is lower during consumption (Fig. 2c–f). We next sought to mimic this pattern of activation optogenetically. To do so, we restricted the optogenetic activation of ChR2-expressing l/vlPAG vgat cells only to exploration and approach epochs, and we turned the laser OFF at consumption onset (Supplementary Fig. 6). This manipulation also increased consumption, and thus shows that optogenetic PAG vgat activation during eating is not necessary to increase consumption (Supplementary Fig. 6). Importantly, neither activation or inhibition of l/vlPAG vgat cells altered locomotion speed in an empty open field (Supplementary Fig. 7c). The observed optogenetic effects were not due to viral spread outside the l/vlPAG or into the dorsal raphe (Supplementary Fig. 8, 9). It is noteworthy that a prior report showed that activation of PAG gad2 cells decreases feeding[10]. Since both gad2 and vgat are commonly used GABAergic markers, these data appear to be in disagreement. We now show that only 42% of the gad2-expressing PAG cells also co-express vgat (Supplementary Fig. 10). Thus, gad2 and vgat are substantially non-overlapping populations within the PAG, explaining the discrepancy between the current study and the prior report on consumption-suppressing gad2 PAG cells[10].

Prior studies have shown that activation of other hunting circuits, such as the GABAergic projection from lateral hypothalamus to l/vlPAG, induces both hunting and aggression towards conspecifics[14]. We thus tested if activation of l/vlPAG vgat cells induces aggression towards other mice, but no behavioral changes were found (Supplementary Fig. 11), indicating that the biting and attack elicited by l/vlPAG vgat activation is specific to predation. These data show that l/vlPAG vgat cell activity is necessary and sufficient for triggering appetitive drive, leading to food approach and consumption of prey and non-prey food types.

To further characterize the role of l/vlPAG vgat cells in foraging and exploration, we optogenetically activated these cells in a variety of situations. First, we show that activation of ChR2-expressing (Fig. 4a) l/vlPAG vgat cells in an empty chamber induced more rearing, which is a common exploratory behavior (Fig. 4b). To verify if this manipulation would induce more complex exploration we added a climbing ladder to the environment and observed that optogenetic activation decreased the latency to climb (Fig. 4c). To further investigate the variability of context-specific exploratory behaviors, we then exposed the mouse to the holeboard task, which contained holes in the floor that the mouse could explore by performing head dips. Activation of vgat cells consistently increased the number of hole visits in this assay (Fig. 4d). To determine if activity in these cells produced positive motivation, we performed the real time place preference task. Mice displayed a preference for the chamber paired with optogenetic stimulation, indicating that this manipulation may be rewarding or engage positive motivational processes (Fig. 4e). We next reasoned that vgat activation may induce chasing, as hunting is a motivated action. We attached a ball to a long stick and moved the ball in a trajectory that spelled the letters "BG" as previously described[9] (Fig. 4f, left panel in upper row; also see Supplementary Movie 1). During blue light delivery, control YFP-expressing mice did not follow the ball (note in Fig. 4f, middle panel in upper row, that the behavioral mouse track, shown in black does not overlap with the 'BG' track made by the ball). Strikingly, ChR2-expressing mice closely followed the ball during the light ON period (note in Fig. 4f, right panel in the upper row, that mouse and ball tracks are highly overlapping). Several measurements averaged across mice indicate that vgat stimulation induced following of the ball, as this manipulation decreased the average distance from the ball, increased correlation between mouse and ball position and decreased the angular offset between mouse head direction and ball (Fig. 4f, lower row). These data show that optogenetic stimulation induces context-specific exploration (Fig. 4b–d) as well as approach towards a moving object (Fig. 4f). Considering these results, we next tested if this manipulation would cause directed exploration and climbing to approach a food source that is only accessible by climbing a wire mesh ladder. Indeed, stimulation of vgat cells decreased the latency to climb and bite a walnut and increased time spent near the walnut (Fig. 4g). Vigorous climbing was visible fairly quickly after onset of light stimulation (see Supplementary Movie 2), similar to the rapid induction of cricket hunting following activation of l/vlPAG vgat cells (Supplementary Movie 3). No changes in locomotion were observed in this experimental condition (Supplementary Fig. 12). Following this assay, we tested if vgat activity caused preferential approach to food compared to an object. To do so, we hung a walnut and an object in opposite corners of the same chambers. Both items were outside of the reach of the mouse. Activation of l/vlPAG vgat cells increased the time spent near the walnut corner (see heatmap in Fig. 4h). Additionally, this manipulation also increased the time spent near the walnut,

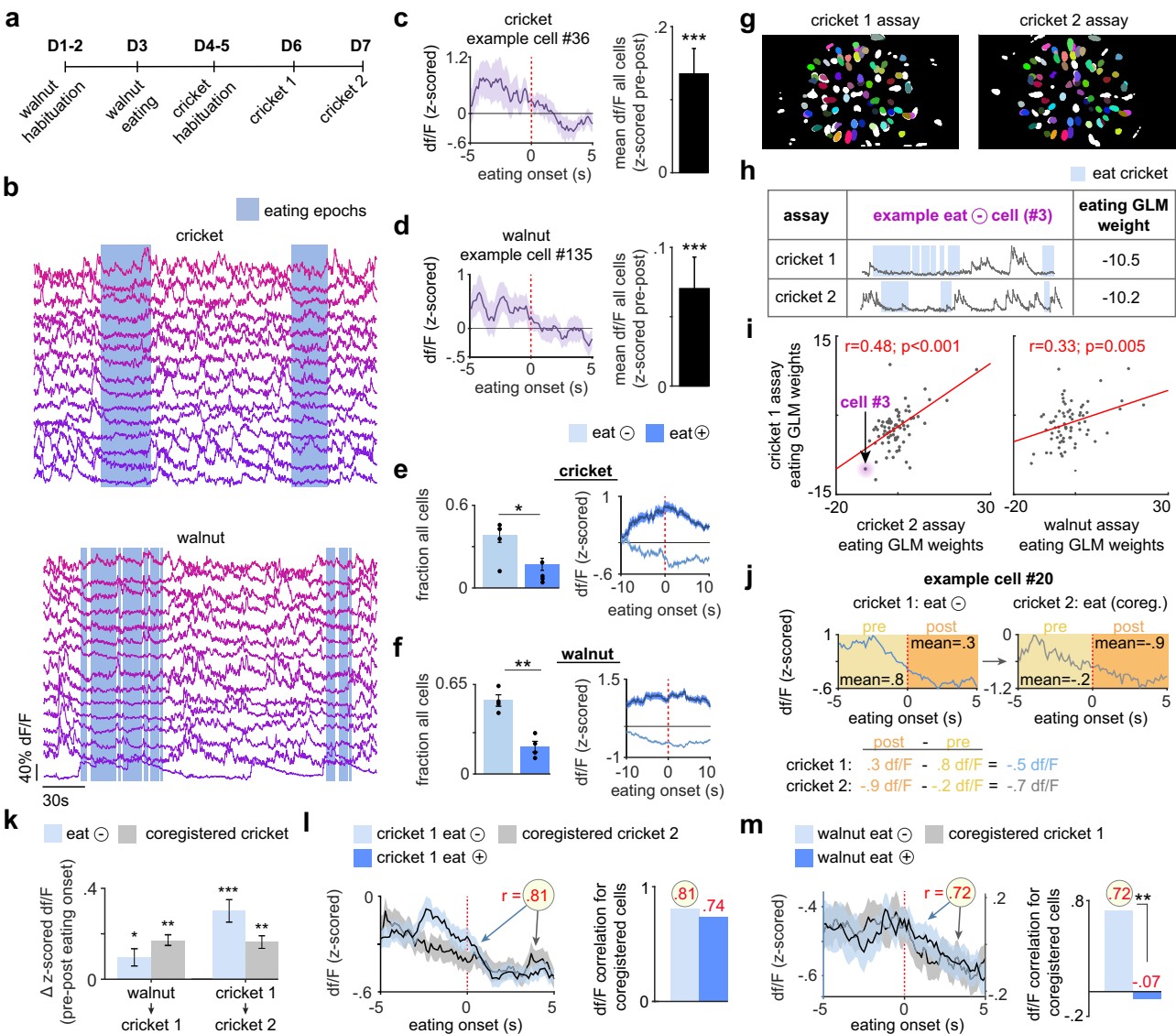

**Fig. 2 | L/vlPAG vgat cells are more active immediately prior to eating. a** Order of assays. **b** L/vlPAG vgat cells are more active prior to eating (blue rectangle). **c** Left: Representative cell showing activation prior to eating cricket. Right: pre-post eating activity of all cells (*n* = 223 cells; two-tailed Wilcoxon signed rank test, z-value = 4.36, *p* < 0.001). **d** Same as **c**, but for walnut (*n* = 204 cells; two-tailed Wilcoxon sign rank test, z-value = 3.78, *p* < 0.001). **e** Left: More cells were negatively modulated by eating cricket (*n* = 4 mice; two-one-sample two-tailed *t*-test, t-statistic=3.15, *p* = 0.026). Right: Mean traces (± 1 s.e.m.) of all cells modulated by eating. (eat- *n* = 86 cells, eat+ *n* = 37 cells). **f** Same as **e**, but for walnut (left: *n* = 4; two-one-sample two-tailed *t*-test, t-statistic=5.88, *p* = 0.009; right: eat - *n* = 110 cells, eat + *n* = 36 cells). **g** Cells co-registered in cricket 1 and cricket 2. **h** General linear model (GLM) weights for representative cell in crickets 1 and 2. **i** Correlation of eating GLM weight across assays (coregistered cricket *n* = 78 cells, coregistered walnut *n* = 74 cells; Spearman correlation, left: *p* < 0.001, right: *p* = 0.005). **j** Example cell co-registered in cricket 1 and 2 shows decreased activity after eating onset in both

assays. **k** (left pair of bars) Eat- walnut cells show decreased activity during eating of both walnut (blue) and cricket 1 (grey) (*n* = 53 cells; two-tailed Wilcoxon signed rank test, walnut z-value = 2.18, *p* = 0.029; coregistered cricket z-value = 2.72, *p* = 0.007) (right pair of bars). Same as the two bars on the left, but for cells co-registered in cricket 1 and 2 (*n* = 35 cells; two-tailed Wilcoxon signed rank test, cricket 1 z-value = 4.49, *p* < 0.001; coregistered cricket 2 z-value = 3.44, *p* = 0.001). **l** Left: Activity of eat- cells in cricket 1 (light blue). Activity for the same cells co-registered in cricket 2 (grey). Right: Correlation of activity centered at eating onset for eat- (light blue) and eat+ (dark blue) cells co-registered in cricket 1 and 2 (eat- *n* = 35 cells, eat+ *n* = 17 cells; Spearman correlation). **m** Same as **l**, but for cells co-registered in walnut and cricket 1 (eat - *n* = 53 cells, eat + *n* = 15 cells; Spearman correlation; for r-value comparison, Fisher r-to-z transformation, *p* = 0.004). ***p* < 0.001, ***p* < 0.01, **p* < 0.05. All error bars and error bands correspond to +/- SEM. Source data are provided as a Source Data File.

but not the object (Fig. 4h). These optogenetic activation data show that l/vlPAG vgat stimulation produced exploratory foraging (Fig. 4g, h) and positive motivation or preference (as shown in the real time preference task in Fig. 4e). We thus investigated if vgat activation would increase food consumption, even if this behavior is associated with negative consequences, which is a cardinal feature of compulsive eating. We created an environment in which part of the floor was covered by an active shocking grid. To gain access to the walnut, mice would have to endure the negative consequence of exposure to foot

shocks (see scheme in Fig. 4i). In both YFP and ChR2-expressing mice, delivery of shocks decreased the amount of walnut consumed. Furthermore, activation of ChR2-expressing vgat l/vlPAG cells rescued the decrease in eating in the presence of the foot shocks. It is worth pointing out that none of these effects were due to analgesia, as excitation of l/vlPAG vgat cells did not alter pain reactions (Supplementary Fig. 13). These data show that vgat stimulation is sufficient to induce eating, even in the presence of aversive consequences—which is a cardinal feature of compulsive eating. We also show that stimulating

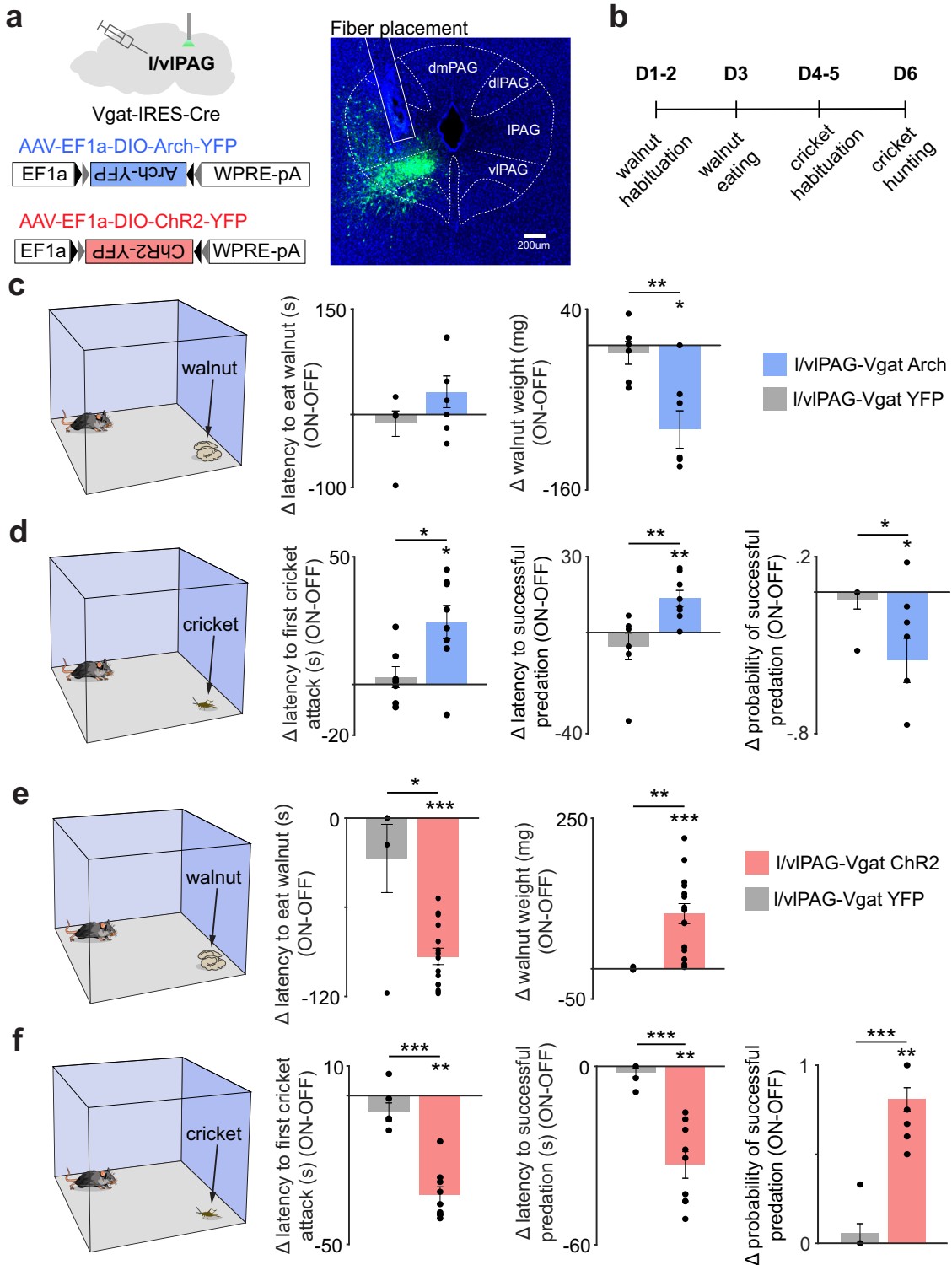

l/vlPAG vgat cells for a longer period (30 min) induces consumption of a large amount of food (Supplementary Fig. 14), which may also contribute to compulsive feeding symptoms. Taken together, these optogenetic activation data show that l/vlPAG vgat activity can induce exploration, foraging, place preference, chasing of moving objects, proximity to food and certain aspects of compulsive eating.

Though this work's focus is on the PAG's role in appetitive drive and eating behaviors, the l/vlPAG is also known to be a central node for the coordination of innate defensive behaviors[22–24]. Furthermore, it is plausible that food consumption may be inhibited in high anxiety states. These views raise the possibility that higher l/vlPAG vgat activity

increases consumption by inhibiting anxiety and other defensive states. We then tested if l/vlPAG vgat activity is sufficient and necessary for defensive states elicited by the anxiogenic elevated plus maze or by a live predatory rat. Bidirectional manipulation of these cells failed to induce changes in open-arm exploration or in the number of risk-evaluation head dips in the elevated plus maze (Supplementary Fig. 15a–d). Similarly, optogenetic excitation or inhibition of l/vlPAG vgat activity also did not alter a wide number of behavioral metrics in the predatory rat exposure assay, including escape speed, percent time freezing and distance to the predator rat (Supplementary Fig. 15e–j). These data indicate that l/vlPAG vgat activity does not

**Fig. 3 | L/vlPAG vgat cell activity bidirectionally modulates eating of cricket and walnut. a** Left: Scheme showing experimental approach for expressing inhibitory (Arch) and excitatory (ChR2) opsins in l/vlPAG vgat cells. Right: Representative Image showing fiber placement and viral expression in l/vlPAG vgat cells. Similar images were obtained in all 13 animals used. **b** Experimental timeline. **c** Optogenetic inhibition of l/vlPAG vgat cells decreases amount of walnut eaten (YFP $n = 6$, Arch $n = 7$; right: Wilcoxon rank-sum, z-value = 2.50, $p = 0.006$; Wilcoxon signed rank (Arch), z-value = −2.20, $p = 0.028$). **d** Inhibition of these cells also increases latency to first cricket attack, latency to successful predation and the probability of successful predation (YFP $n = 7$, Arch n = 8, left: Wilcoxon rank-sum, z-value = −2.03, $p = 0.042$; Wilcoxon signed rank (Arch), z-value = 2.38, $p = 0.013$; middle: Wilcoxon rank-sum, z-value = −2.72, $p = 0.007$; Wilcoxon signed rank (Arch), z-value = 2.52, $p = 0.006$; right: Wilcoxon rank-sum, z-value = 2.10, $p = 0.036$; Wilcoxon signed rank (Arch), z-value = −2.25, $p = 0.024$). **e** Optogenetic excitation of l/vlPAG vgat cells decreased latency to eat walnut and increased amount walnut eaten (YFP $n = 5$, ChR2 $n = 15$, left: Wilcoxon rank-sum, z-value = 1.96, $p = 0.049$; Wilcoxon signed rank (ChR2), z-value = −3.41, $p < 0.001$; right: Wilcoxon rank-sum, z-value = −3.14, $p = 0.002$; Wilcoxon signed rank (ChR2), z-value = 3.41, $p < 0.001$). **f** Excitation of ChR2-expressing l/vlPAG vgat cells decreased latency for the first cricket attack and the latency to successful predation. This manipulation also increased the probability of successful predation. (YFP $n = 6$, ChR2 $n = 9$, left: Wilcoxon rank-sum, z-value = 3.13, $p < 0.001$; Wilcoxon signed rank (ChR2), z-value = −2.67, $p = 0.008$; middle: Wilcoxon rank-sum, z-value = 3.15, $p < 0.001$; Wilcoxon signed rank (ChR2), z-value = −2.67, $p = 0.008$; right: Wilcoxon rank-sum, z-value = −3.21, $p < 0.001$; Wilcoxon signed rank (ChR2), z-value = 2.69, $p = 0.007$). ***$p < 0.001$, **$p < 0.01$, *$p < 0.05$. Data are presented as mean +/- SEM. Source data are provided as a Source Data File.

modulate defensive states caused by open spaces or predators. To test if l/vlPAG vgat activity is modulated by proximity to innate threats we expressed the calcium indicator GCaMP6s in l/vlPAG vgat cells and obtained recordings of calcium transients during exposure to the elevated plus maze and a live predatory rat (Supplementary Fig. 15k, l). Heatmaps of neural activity show that vgat cells do not consistently encode distance to threat (Supplementary Fig. 15m, n), as higher activity was not displayed consistently either near or far from either threat category. Indeed, l/vlPAG vgat activity was not correlated with distance to either the open arms or the rat (Supplementary Fig. 15o). Taken together, these data suggest it is unlikely that l/vlPAG vgat cells affected eating as an indirect consequence of anxiety suppression.

### l/vlPAG vgat cells are bidirectionally connected with feeding circuits

Our data show that l/vlPAG vgat cells regulate food consumption. We thus hypothesized that these cells would be interconnected with other regions known to regulate eating. To test this idea, we injected AAV5-hSyn-FLEX-TVA-P2A-eGFP-2A-oG in the l/vlPAG of vgat-Cre mice (Fig. 5a). Three weeks later, we injected EnvA-dG-rabies-mCherry in the same location (Fig. 5b). Numerous double-labeled starter cells were observed in the l/vlPAG (Fig. 5c). Rabies-infected mcherry-expressing inputs to l/vlPAG vgat cells were seen in several regions known to control hunting, including the medial preoptic area, the central amygdala, the lateral hypothalamic area, the bed nucleus of the stria terminalis and the ZI (Fig. 5d). Interestingly, activation of projections from all these regions to the l/vlPAG induce hunting[7,9,11,12,14]. These data show that l/vlPAG vgat cells are a common downstream target of diverse circuits that control hunting related behaviors. However, the potential downstream targets of the l/vlPAG vgat cells that may mediate its role in eating have not been identified. We thus next studied if l/vlPAG vgat cells projected to regions that control hunting or food consumption. We injected AAV5-DIO-EFf1a-YFP in the l/vlPAG of vgat-Cre mice (Fig. 6a) and mapped the resulting projection patterns. Projections from l/vlPAG vgat cells were seen in regions known to control feeding, such as the bed nucleus of the stria terminalis and the hypothalamus (Fig. 6c, d). Interestingly, the strongest projection was seen in the ZI. We now identify a previously uncharacterized monosynaptic GABAergic projection from l/vlPAG vgat cells to the ZI (Fig. 6e–g). Though the functional role of this l/vlPAG to ZI projection is unknown, prior data have shown that activation of either ZI vgat cell bodies or of the ZI projection to the l/vlPAG causes hunting, and ZI vgat cells were also activated during food deprivation[7]. Thus, both anatomical and functional data suggest that the l/vlPAG to ZI projection may be involved in foraging.

### Activity in the ascending projection from l/vlPAG vgat cells to zona incerta can bidirectionally control food consumption

To dissect the activation patterns of the l/vlPAG vgat projection to the ZI, we obtained recordings of calcium transients in this projection. We expressed the calcium indicator GCaMP6s in l/vlPAG vgat cells and implanted fiberoptic cannulae in axons terminating in the ZI (Fig. 7a, b). Fiber photometry recordings show that both in the walnut and the cricket assay, this projection was more active prior to eating onset, as illustrated by representative traces and averaged data (Fig. 7d–g). We also studied if this activation prior to eating could be observed without touching the food. We thus built a U-shaped maze; at the end of each arm, we placed either an object or a walnut behind wire mesh screens (Fig. 7h). Our data show that the l/vlPAG vgat projection to ZI was activated during approach to a walnut, but not an object, even if the walnut was out of reach and thus could not be consumed or touched (Fig. 7i–k). These data show that the l/vlPAG vgat projection to the ZI is activated during approach to food, prior to eating onset, indicating that this bottom-up, midbrain to subthalamic region circuit may have a role in consumption and approach to food. In support of this view, l/vlPAG vgat cells encode proximity to prey (Supplementary Fig. 16f, g).

In order to test the necessity of the l/vlPAG vgat projection to ZI for appetitive drive, we expressed optogenetically inhibited Arch-expressing axon terminals of this projection in the walnut and cricket assays (Fig. 8a). Remarkably, inhibition of this projection decreased the amount of walnut consumed (Fig. 8c), increased the latency to first cricket attack and decreased the probability of hunting a cricket (Fig. 8d). We next tested if activity in this projection was sufficient to increase these behaviors. We thus expressed the excitatory opsin ChR2 in l/vlPAG vgat cells and implanted fiberoptic cannulae over the ZI (Fig. 8a). Optogenetic activation of this projection with delivery of blue light increased walnut consumption (Fig. 8e), decreased the latency to cricket predation and increased the probability of killing the cricket (Fig. 8f). It is possible that optogenetic terminal activations induce antidromic action potentials. However, our optogenetic projection inhibition of l/vlPAG vgat terminals in ZI decreased food-seeking, and these data are not confounded by potential antidromic effects (Fig. 8c, d). Our data thus indicate that higher activity in the l/vlPAG vgat projection to the ZI increases feeding. As the l/vlPAG vgat cells are GABAergic, it is expected that activation of this projection will inhibit the ZI. However, prior data are in disagreement with our results, as it has been shown that ZI inhibition decreases food-seeking[7,8]. To resolve this issue, we investigated the dynamics of this circuit. We expressed GCaMP6f in l/vlPAG vgat cells and ZI vgat cells contralaterally to avoid contamination of cell body signals with signals from long-range axon terminals. Fiber photometry recordings from contralateral hemispheres show that vgat cells from both the l/vlPAG and the ZI are activated during approach to food (Supplementary Fig. 17). These data suggest that the l/vlPAG vgat cells are not broadly inhibiting ZI vgat cells.

To causally test this idea, we expressed the red-shifted opsin ChRmine3.3 in l/vlPAG vgat cells and the calcium indicator GCaMP6f in ZI vgat cells (Supplementary Fig. 18a, b). Remarkably, optogenetic excitation of l/vlPAG vgat cells in freely-moving mice increased neural

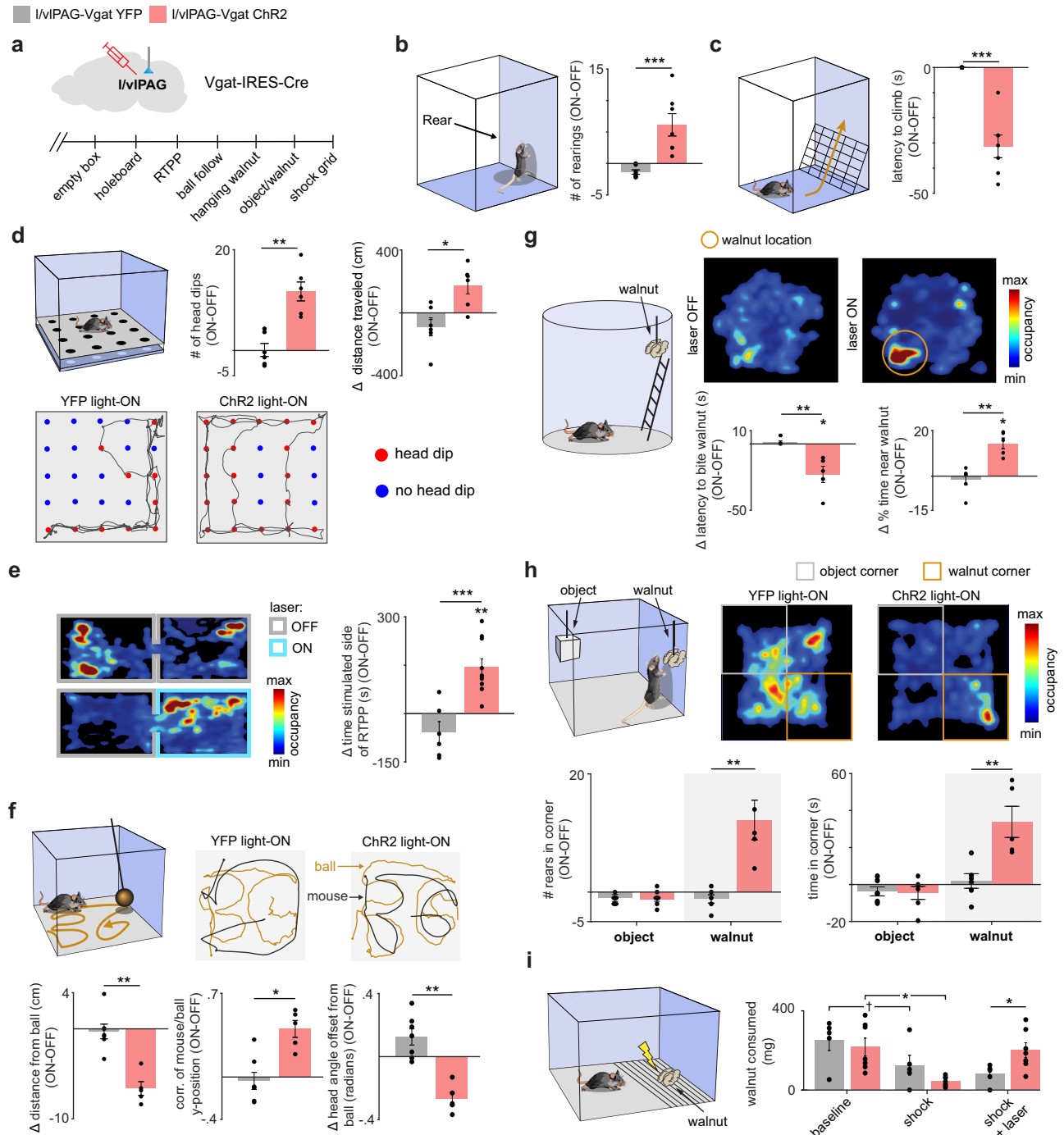

**Fig. 4 | Optogenetic excitation of l/vlPAG vgat cells produces exploratory foraging, place preference and compulsive eating. a** Experimental timeline (bottom). **b** L/vlPAG vgat stimulation increases rearing in an empty box (YFP $n = 7$, ChR2 $n = 7$; Wilcoxon rank-sum, z-value = 3.07, $p < 0.001$) and also **c** climbing in an environment with a ladder (YFP $n = 7$, ChR2 $n = 7$; Wilcoxon rank-sum, z-value = −3.28, $p < 0.001$). **d** L/vlPAG vgat activation increased headdips into holes in the holeboard test, and also increased distance traveled. Lower rows: Representative tracks (YFP n = 6, ChR2 n = 6; left: Wilcoxon rank-sum, z-value = −2.81, $p = 0.005$; right: Wilcoxon rank-sum, z-value = −2.428, $p = 0.015$). **e** Left: Behavioral heat maps showing l/vlPAG vgat stimulation induced real-time place preference (YFP $n = 6$, ChR2 $n = 10$; Wilcoxon rank-sum, z-value = −3.48, $p < 0.001$; Wilcoxon signed rank (ChR2), z-value = 3.09, $p = 0.002$). **f** A ball was moved on a trajectory spelling "BG" (upper left). L/vlPAG vgat activation decreased distance to ball, increased correlation of the position of mouse and ball and decreased the head angle offset between mouse and ball (YFP $n = 7$, ChR2 $n = 5$; Wilcoxon rank-sum; left: z-value = 2.76, $p = 0.006$; middle: z-value = −2.43, $p = 0.015$; right: z-value = 2.76, $p = 0.006$). **g** L/

vlPAG stimulation decreased latency to climb a ladder and then bite the walnut, and increased time spent in the walnut zone (YFP $n = 6$, ChR2 $n = 5$; bottom left: Wilcoxon rank-sum, z-value = 2.71, $p = 0.007$; Wilcoxon signed rank (ChR2), z-value = −2.02, $p = 0.043$; bottom right: Wilcoxon rank-sum, z-value = −2.65, $p = 0.008$; Wilcoxon signed rank (ChR2), z-value = 2.02, $p = 0.043$). **h** An object and a walnut were suspended in opposite corners (upper left). L/vlPAG vgat activation increased rearing and time spent in the walnut corner, but not the object corner (YFP $n = 6$, ChR2 $n = 5$; bottom left: Wilcoxon rank-sum (walnut), z-value = −2.65, $p = 0.008$; bottom right: Wilcoxon rank-sum (walnut), z-value = −2.65, $p = 0.008$). **i** To measure compulsive eating, walnuts were placed on a shocking grid to evaluate if mice endured foot-shocks to access walnuts. L/vlPAG vgat stimulation (shock+laser epoch) increased consumption. (YFP $n = 5$, ChR2 $n = 7$; Wilcoxon signed rank (baseline vs. shock) YFP: z-value = 1.75, $p = 0.08$; ChR2: z-value = 2.37, $p = 0.018$; Wilcoxon rank-sum (shock+laser), z-value = 2.76, $p = 0.01$) ***$p < 0.001$, **$p < 0.01$, *$p < 0.05$, † $p = 0.08$. Data are presented as mean +/- SEM. Source data is in the Source Data File.

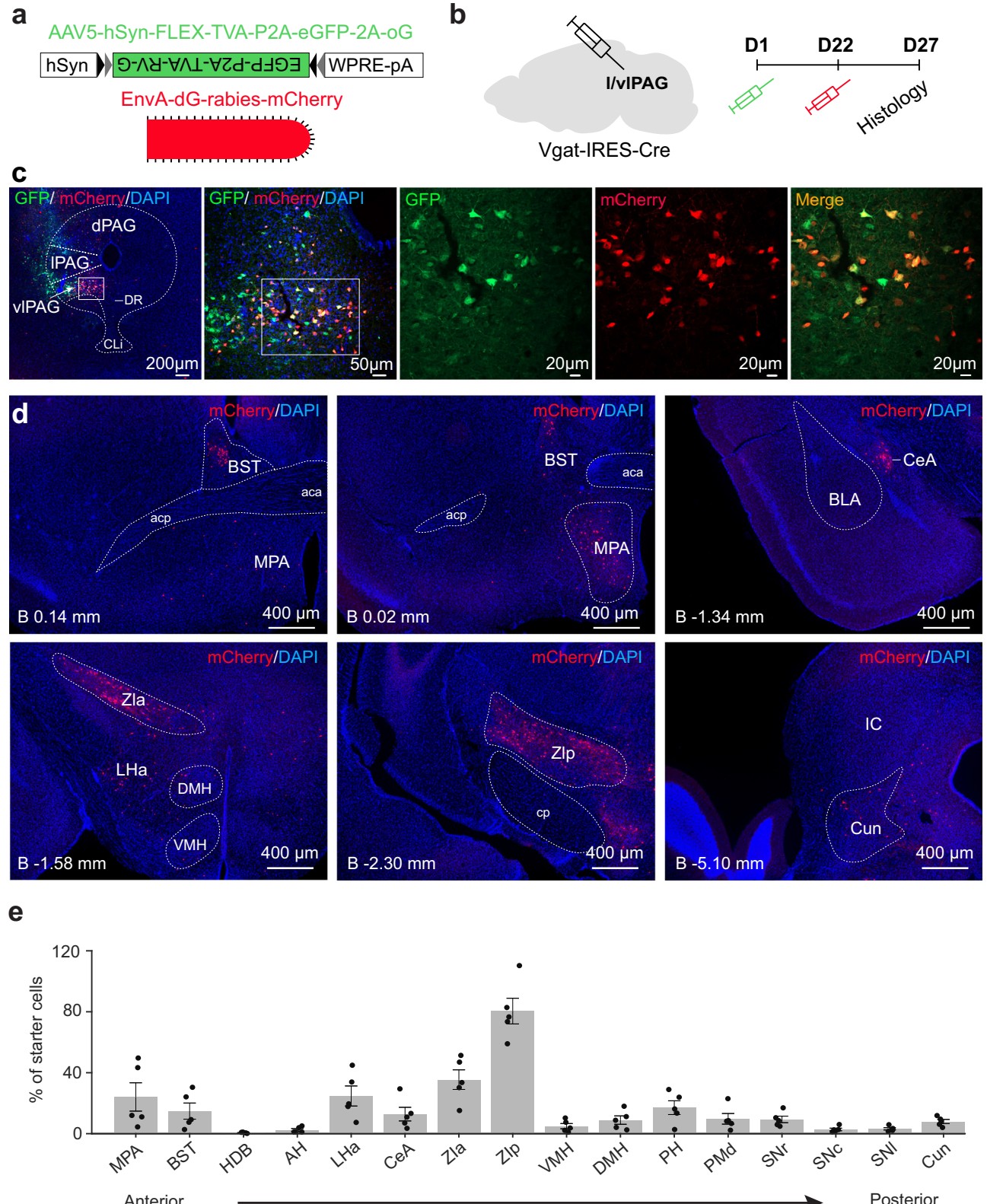

activity in ZI vgat cells time-locked to laser ON epochs. The same manipulation had no effect in ZI vgat cells during isoflurane anesthesia. This result indicates that the increase in ZI vgat GCaMP6f fluorescence during laser ON epochs is due to increased ZI neural activity, and this effect was not caused by contamination by laser light artifacts (Supplementary Fig. 18c, d). Thus, both correlative (Supplementary Fig. 17) and perturbative data (Supplementary Fig. 18) show that the l/vlPAG

input activates ZI vgat cells in freely-moving mice. These results reconcile our data with prior reports showing that higher ZI vgat activity causes increased food-seeking[7,8]. Taken together, our data show, activity in the l/vlPAG vgat to ZI projection is sufficient and necessary for appetitive drive of both non-prey and prey food types, showing that a bottom-up mechanism originating in the midbrain can potently affect food consumption.

**Fig. 5 | L/vlPAG vgat cells receive inputs from regions that control feeding. a** In order to identify inputs to PAG vgat cells, vgat-Cre mice were injected with AAV5-hsyn-FLEX-TVA-P2A-eGFP-2A-oG in the PAG. Three weeks later, mice were injected with EnvA-dG-rabies-mCherry in the same location. **b** Timeline of viral injections. **c** Expression of viral vectors in the vlPAG and the lPAG in vgat-Cre mice. Double-labelled GFP (green) and mCherry (red)-expressing vgat PAG cells (shown in yellow in 'Merge' panel) show starter cells for monosynaptic retrograde tracing. **d** mCherry-expressing (red) cells that project to PAG vgat cells in the bed nucleus of the stria terminalis (BST), medial preoptic area (MPA), central amygdala (CeA), Zona incerta anterior and posterior regions (ZIa and ZIp, respectively), lateral hypothalamic area (LHa) and cuneiform nucleus (Cun). **e** Quantification of retrogradely-labelled cells across brain regions. Structures are ordered from anterior to posterior locations. Note that the Zona incerta (ZI) provides the most prominent input to PAG vgat cells (*n* = 6807 starter cells and 16,704 retrogradely

labeled cells). Anterior-posterior coordinates in mm from Bregma (B) is displayed in the lower left corner of each image. acp anterior commissure, posterior part, aca anterior commissure, anterior part, AH anterior hypothalamus, BST bed nucleus of the stria terminalis, BLA basolateral amygdala, CeA central amygdala, Cun cuneiform nucleus, cp cerebral peduncle, DMH dorsomedial hypothalamus, DR dorsal raphe, HDB nucleus of the horizontal limb of the diagonal band, IC inferior colliculus, LHa lateral hypothalamus area, MPA medial preoptic area, dPAG lPAG and vlPAG dorsal, lateral and ventrolateral periaqueductal gray, respectively, PH posterior hypothalamus, PMd dorsal premammillary nucleus, SNl, SNc, SNr lateral, compact and reticular parts of the substantia nigra, respectively, VMH ventromedial hypothalamus, ZIa and ZIp zona incerta anterior and posterior regions, respectively. Data are presented as mean values +/- SEM. Source data are provided as a Source Data File.

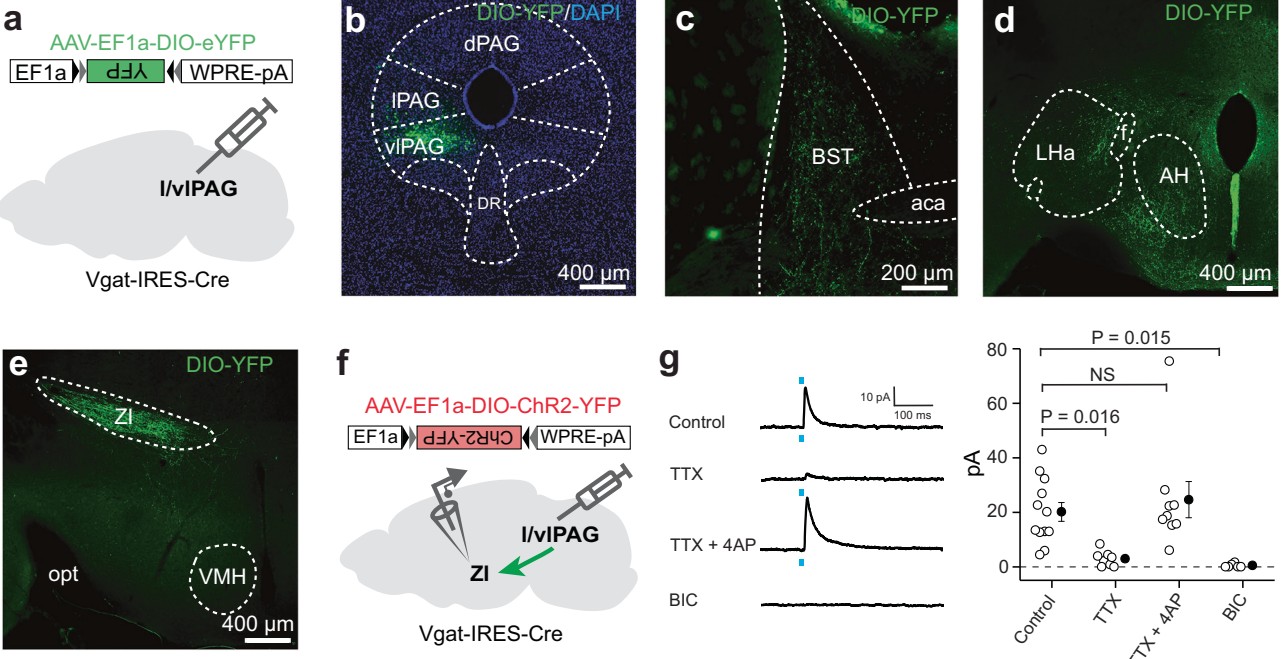

**Fig. 6 | Bottom-up l/vlPAG vgat cells' projections to the zona incerta and other regions that control feeding. a** In order to characterize the anterograde connectivity of l/vlPAG vgat cells, vgat-Cre mice were injected with AAV5-EF1a-DIO-YFP in the PAG. **b** Expression of YFP in the l/vlPAG of vgat-Cre mice. **c** YFP axon terminals that project to the bed nucleus of the stria terminalis (BST), **d** hypothalamus and **e** zona incerta (ZI). **b–e** Similar images were obtained in all 5 animals used. **f** Scheme showing injections used to characterize monosynaptic GABAergic input from PAG vgat cells to the zona incerta. **g** (Left panel) Representative traces of ex vivo slice recordings of zona incerta neurons following optogenetic excitation of ChR2-expressing axon terminals of vgat l/vlPAG cells. Traces show recordings in the

absence of drugs (control), 0.5 μM tetrodotoxin (TTX), 0.5 μM TTX + 100 μM 4-Aminopyridine (4AP) and 25 μM bicuculline (BIC) (from the top to the bottom). **g** (Right panel) Quantitative analysis of the amplitude of light-evoked inhibitory postsynaptic currents from the zona incerta for the control (n = 12 cells from 5 mice), 0.5 μM TTX (*n* = 9 cells from 5 mice), 0.5 μM TTX + 100 μM 4-Aminopyridine (*n* = 9 cells from 5 mice) and 25 μM bicuculline conditions (*n* = 6 cells from 3 mice). One-way ANOVA test following by a means comparison Tukey test, NS Not Significant, aca anterior commissure, AH anterior hypothalamus, LHa lateral hypothalamus area, VMH ventromedial hypothalamus. Data are presented as mean values +/- SEM. Source data are provided as a Source Data File.

## Discussion

Actions related to exploration and investigation are of central importance, as these processes are the initial actions that lead to foraging, hunting and other behaviors. Despite their relevance, the mechanisms controlling such actions remain elusive. Here, we show that l/vlPAG vgat cells encode and bidirectionally control these behaviors, indicating a role in foraging (Fig. 3). Indeed, l/vlPAG vgat activity is sufficient and necessary for foraging and consumption. Optogenetically activating these cells produces foraging, exploration (Fig. 4b–h), and even induces aspects of compulsive eating (Fig. 4i). We also show that vgat l/vlPAG cells are bidirectionally connected with known feeding nodes, and that these cells control feeding via a projection to the ZI. Though a role for the l/vlPAG in mice hunting crickets has been described, feeding in most modern humans does not involve hunting and mostly consists of non-prey foods. Since the l/vlPAG vgat to ZI

projection controls foraging and eating of both prey and non-prey foods (Fig. 8), it may be relevant for feeding in humans as well. The data suggest the l/vlPAG may have a broad role in feeding, rather than a species-specific role in cricket hunting in mice. Lastly, though several higher order inputs to the PAG have been shown to control hunting[7,9,11,12,14], here we identify a bottom-up foraging and consumption circuit originating in the PAG.

### Individual vgat l/vlPAG neurons show conserved eating-related modulation across different feeding assays

We characterized PAG vgat activity during appetitive and consummatory actions induced both by prey and non-prey food. These cells displayed increased activity preceding consumption (Fig. 2c, d) and prominent decreased activity following eating onset (Fig. 2e, f). This pattern was conserved across both feeding assays tested

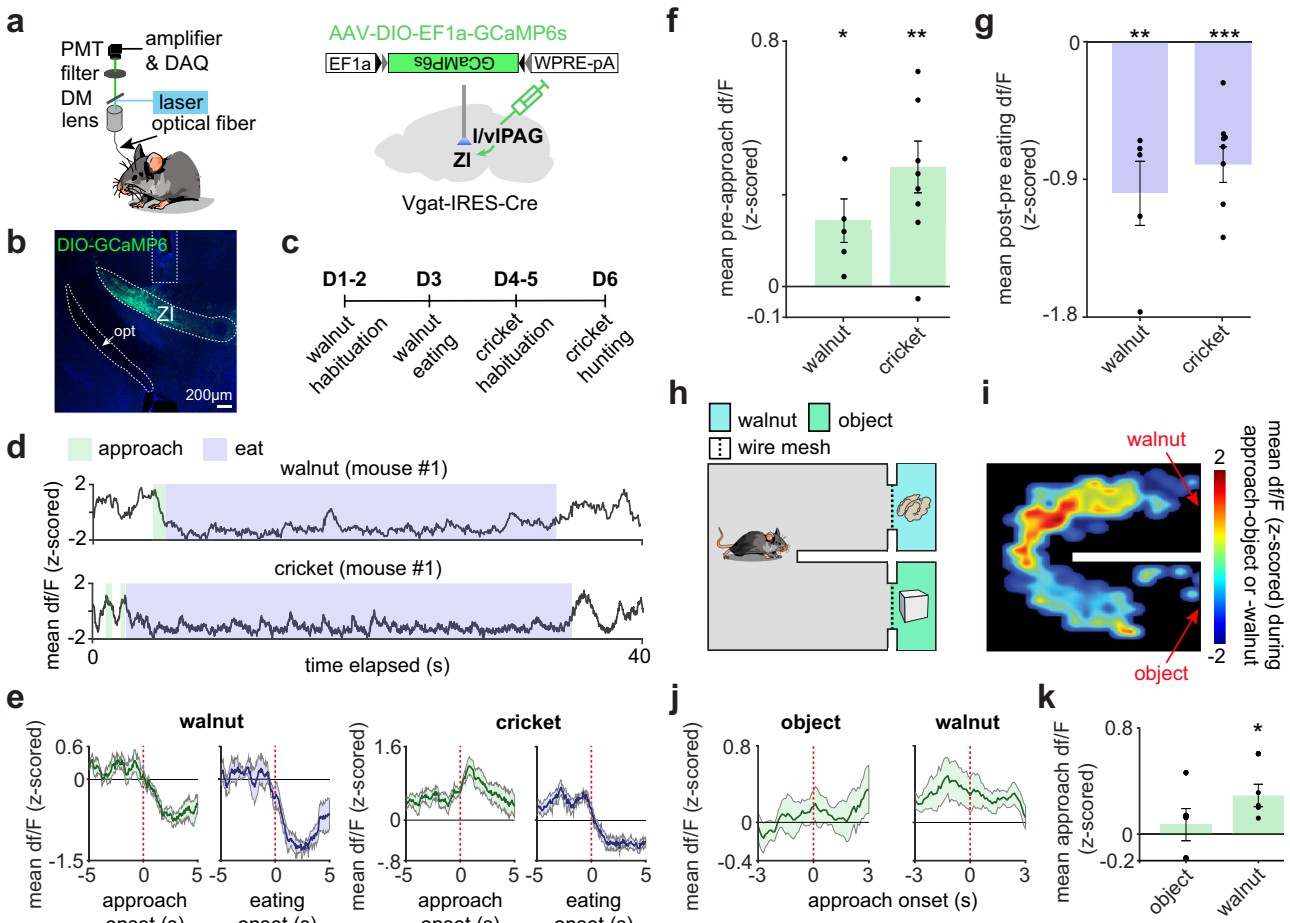

**Fig. 7 | Vgat l/vlPAG projection to zona incerta is activated during approach to food. a** (left) Scheme for fiber photometry recording. (right) vgat-Cre mice were injected with a vector encoding cre-dependent GCaMP6s in the l/vlPAG. A fiber-optic cannula was implanted above the zona incerta (ZI) to record calcium transients in l/vlPAG vgat axon terminals in the zona incerta. **b** Image showing GCaMP6s-expressing l/vlPAG vgat axon terminals in the ZI. Similar images were obtained in all 8 animals used. **c** Experimental timeline. **d** Example recording showing that activity in the l/vlPAG vgat projection to zona incerta is higher during approach to walnut (green), and lower during walnut eating (blue) (top). Bottom: same as top, but for recordings from the same mouse in the cricket assay. **e** Left: Average activity in the l/vlPAG vgat projection to zona incerta during walnut approach and eating. Right: same as left, but for the cricket assay. The l/vlPAG vgat projection to zona incerta is activated prior to approach (**f**) and is inhibited

following eating onset (**g**) in both assays (walnut $n = 5$, cricket $n = 8$; one-sample t-test, approach t-statistics: walnut=3.03, $p = 0.039$, cricket = 4.37, $p = 0.003$; eat t-statistics: walnut = −4.73, p = 0.009, cricket = −6.85, $p < 0.001$). **h** Mice were exposed to the environment depicted in the scheme. Mice were able to see and smell the walnut through the wire mesh but were not able to eat or touch it. **i** Heat map showing approach activity in the l/vlPAG vgat projection to the zona incerta for the assay shown in **h**. Note activity is highest in the entrance to the walnut corridor. **j** Average activity during approach onset to object or walnut.
**k** Significantly elevated activity was observed during approach to walnut, but not object ($n = 5$; one-sample *t*-test, t-statistic (walnut) = 3.45, $p = 0.026$). ***$p < 0.001$, **$p < 0.01$, *$p < 0.05$. Data are presented as mean values +/- SEM. All error bars and error bands correspond to +/- SEM. Source data are provided as a Source Data File.

(Fig. 2k, m). Interestingly, cells that were inhibited during consumption showed correlated activity profiles across cricket and walnut assays (Fig. 2m). In contrast, cells activated by consumption were uncorrelated across assays (Fig. 2m). These data show negative modulation of vgat PAG cells during eating (Fig. 2m) is the most conserved pattern during the consummatory phase, regardless of food type.

## PAG vgat neurons induce context-specific foraging actions

Environmental cues interact directly with the motivation to promote behavioral adaptation. Accordingly, we have shown previously that other PAG circuits function to integrate multisensory signals[25] and encode approach or avoidance states[26]. Our data showed that vgat PAG neurons were mostly activated in the moments that precede eating in different situations (Fig. 2c, d). Indeed, a diversity of behavioral assays suggest that the activation of these neurons is associated with increased exploration (Fig. 4b–d), appetitive effects (Fig. 3e, f) and positive motivation (Fig. 4e,f,i). These effects were context-dependent, as unique behaviors were displayed in different contexts during PAG

vgat activation (Fig. 4). Furthermore, activating these cells also elicited a strong motivation to pursue and follow objects (Fig. 4f). Here we show that these targeted-directed actions towards objects are flexible and depend on the resources available in the environment. In the absence of food, vgat l/vlPAG activation creates exploration and engagement with objects, but in the presence of food the directed actions preferentially target food (Fig. 4h).

Activation of l/vlPAG vgat cells does not induce locomotion per se, but rather elicits exploration of the available stimuli of interest, which may be holes in the holeboard task, an object or a hanging walnut in the other task (Fig. 4). An increase in speed during optogenetic stimulation was only observed when the exploratory action required higher locomotion (Supplementary Fig. 12). Consequently, optogenetic stimulation did not alter speed in the empty open field (an environment that does not have anything to explore - Supplementary Fig. 7c) or in the hanging walnut assay (where exploration of the walnut did not require increased movement - Supplementary Fig. 12). Similarly, endogenous l/vlPAG vgat activity also did not correlate with

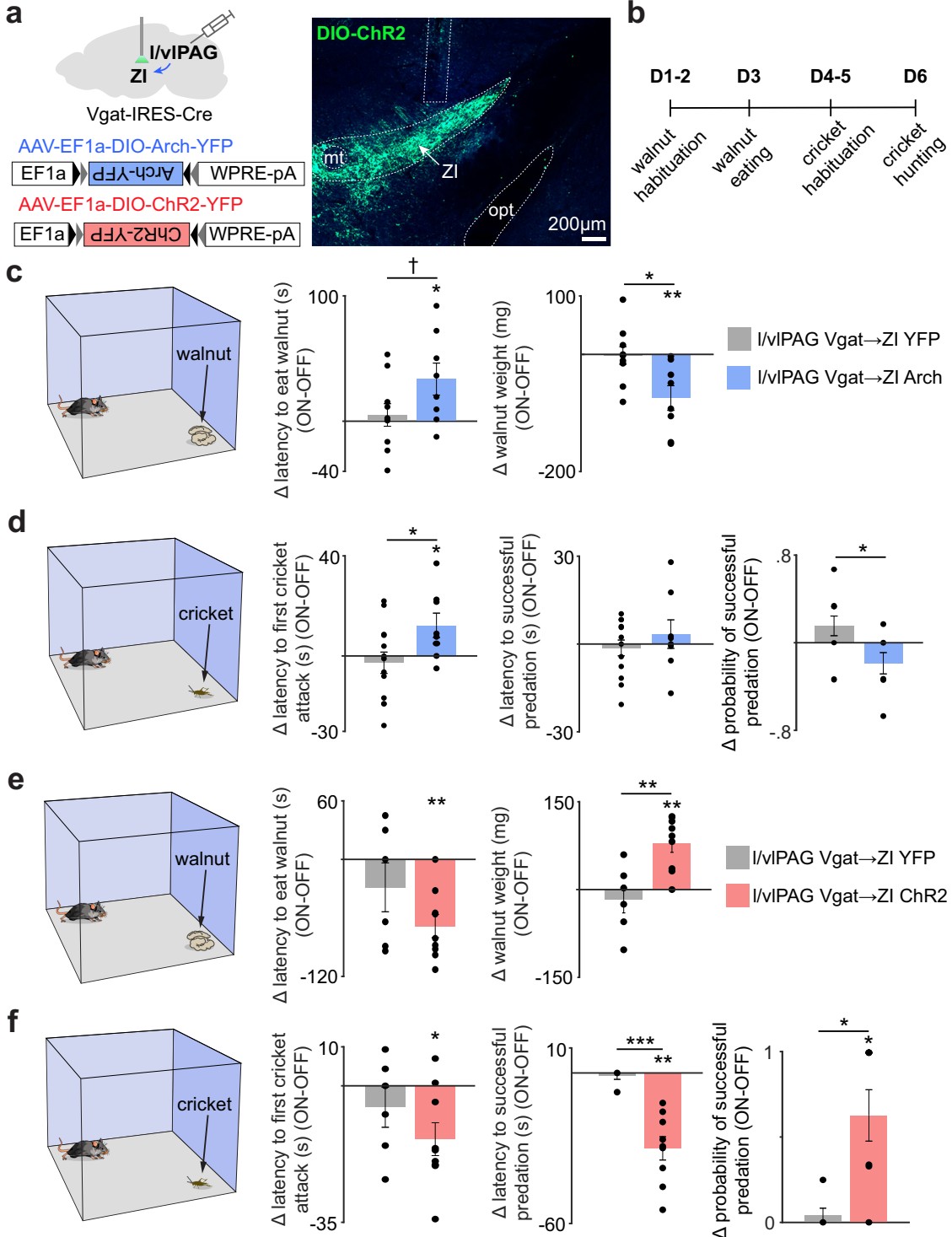

**Fig. 8 | Activity in the l/vlPAG vgat projection to zona incerta is necessary and sufficient for consumption of walnut and crickets. a** Left: scheme showing viral strategy to express the opsins Arch (inhibitory) or ChR2 (excitatory) in l/vlPAG vgat cells. A fiberoptic cannula was placed over the zona incerta (ZI) to inhibit or excite the l/vlPAG vgat projection to the zona incerta. Right: Image showing ChR2-YFP expressing l/vlPAG axon terminals in the ZI. Opt: optic tract. Similar images were obtained in all 21 animals used. **b** Experimental timeline. **c** Inhibition of the l/vlPAG vgat projection to the zona incerta decreased eating of walnut (YFP $n$ = 10, Arch $n$ = 8; left: Wilcoxon rank-sum, z-value = −1.60, $p$ = 0.11; right: Wilcoxon rank-sum, z-value = 2.45, $p$ = 0.014; Wilcoxon signed rank (Arch), z-value = −2.52, $p$ = 0.006). **d** This manipulation also increased latency to first cricket attack and decreased the probability of killing the cricket (YFP $n$ = 13, Arch $n$ = 8; left: Wilcoxon rank-sum, z-

value = −2.03, $p$ = 0.042, Wilcoxon signed rank (Arch), z-value = 2.12, $p$ = 0.034; right: Wilcoxon rank-sum, z-value = 2.06, $p$ = 0.039). **e** Excitation of the l/vlPAG vgat projection to the zona incerta increased consumption of walnut (YFP $n$ = 6, ChR2 $n$ = 9; left: Wilcoxon signed rank (ChR2), z-value = −2.52, $p$ = 0.006; right: Wilcoxon rank-sum, z-value = −2.53, $p$ = 0.006; Wilcoxon signed rank (ChR2), z-value = 2.52, $p$ = 0.006). **f** This manipulation also decreased latency to successful cricket predation and increased the probability of killing the cricket (YFP $n$ = 6, ChR2 $n$ = 9; middle: Wilcoxon rank-sum, z-value = 3.18, $p$ < 0.001, Wilcoxon signed rank (ChR2), z-value = −2.67, $p$ = 0.008; right: Wilcoxon rank-sum, z-value = −2.47, $p$ = 0.014; Wilcoxon signed rank (ChR2), z-value = 2.46, $p$ = 0.014). ***$p$ < 0.001, **$p$ < 0.01, *$p$ < 0.05, † $p$ = 0.11. Data are presented as mean values +/- SEM. Source data are provided as a Source Data File.

speed, and their neural activity could not be used to decode speed (Supplementary Fig. 7b). Taken together, these data show that l/vlPAG activity neither controls nor correlates with overall locomotion. In contrast, it is critical to point out that inhibition of feeding-inducing dorsal raphe vgat cells increases locomotion speed[27,28]. As l/vlPAG vgat cells do not directly affect overall locomotion, our data indicate that the observed results in our work were not due to optogenetic activation of raphe vgat cells. Accordingly, viral expression did not reach the dorsal raphe (Supplementary Fig. 9).

Activation of l/vlPAG vgat neurons also induced perseverative eating in the presence of an aversive stimuli indicative of involvement of this neural population in a strong motivational process potentially related to compulsive eating behaviors (Fig. 4i). Furthermore, inhibition of these neurons impaired both walnut consumption and reduced the probability of predation (Fig. 3d-c). These effects are not related to a decrease in fear or anxiety, as no correlation was observed between neural activity and threat proximity, and neither optogenetic activation nor inhibition impacted approach to threat (Supplementary Fig. 15), revealing the presence of PAG circuits that are relatively specialized for the pursuit of caloric resources.

Intriguingly, activation of the projection from lateral hypothalamus to PAG induces both hunting and attacking of conspecifics concomitantly with real-time place aversion[14]. In contrast, activation of PAG vgat cells induces foraging and real-time place preference (Fig. 4), but not aggression towards other mice (Supplementary Fig. 11), indicating these cells selectively control foraging-related behaviors. These results suggest that vgat PAG neurons produce positive motivation associated with driving appetitive responses.

## PAG vgat neurons are bidirectionally connected to a food-seeking network

Here, we describe a bottom-up ascending pathway mediated by vgat PAG neurons to control appetitive and consummatory behaviors. Though the PAG has been viewed as a downstream target of both approach[7,9,11,12,14] and avoidance behaviors[22,29–32], there is bidirectional connectivity between the PAG and the ZI[7], and other diencephalic regions[16,20,21,33]. Thus, while PAG neurons are downstream of forebrain control of both instinctive and learned behaviors, these regions may also be influenced by neural signals generated by the PAG[17,18,34]. Our data suggest that vgat PAG neurons project to forebrain structures and comprise a network loop with other brain regions engaged in appetitive and consummatory behaviors (Figs. 5 and 6). This pathway could provide the sensory and motivational information signals to ZI, which integrates afferent signals to influence investigatory motivation and initiation. For example, the ZI is essential for the decision to investigate an object or a conspecific, novelty-seeking[5], and feeding[7,8]. Thus, the existing bidirectional connectivity of the PAG with forebrain hubs controlling motivated behaviors[16,20,21,33] suggest the PAG may be critical in orchestrating motivational systems that influence both defensive[29,31,35] and appetitive responses[7,9,11,12,14]. This idea is further corroborated by our data showing that vgat PAG cells receive inputs from circuits known to control both positively and negatively motivated behaviors (Fig. 5). The PAG projects to the forebrain to influence a wide array of brain states, including complex responses and learning related to positive and negative motivation[15,19]. We now show that the activation of l/vlPAG vgat cells is reinforcing, and increases the propensity to forage and to pursue food (Fig. 4). Thus, higher l/vlPAG vgat neural activity positively correlates with and causes appetitive drive, foraging and food-seeking actions, and as a consequence, an increase in food consumption.

## GABAergic circuits involved in feeding and predation

Prior studies have shown that the recruitment of a complex neural circuit composed of several GABAergic inputs to the PAG induces food-foraging and hunting. To this end it has been shown that activation of inhibitory inputs to the PAG from the lateral hypothalamus[7,9],

bed nucleus of the stria terminalis[10] the central amygdala nucleus[11] and the ZI[7] induce hunting. However, these previous reports did not establish that l/vlPAG vgat cells were the relevant downstream target of these long-range inhibitory inputs. Thus, it is important to consider that these inhibitory inputs can exert their behavioral functions by targeting different cell types in the PAG (i.e., gad2, vgat and vglut2 neurons). One possible mechanism is that these inputs exert their influence on hunting by acting on PAG vglut2 cells. It is well-established that PAG vglut2 cells promote high fear states during which foraging and feeding is inhibited. Indeed, a switch from defense to hunting has been observed in mice first exposed to crickets[36]. According to this view, activation of long-range GABAergic inputs from the lateral hypothalamus and other regions may increase the propensity to hunt by inhibiting fear-promoting PAG vglut2 neurons. For example, hunting elicited by activation of both central amygdala[11] or lateral hypothalamus[14] projections to PAG is impaired by the activation of PAG vglut2 cells. Taken together, these data suggest that activation of diverse inhibitory long-range inputs to the PAG can elicit hunting through complex neural dynamics of distinct subpopulations of l/vlPAG neurons.

As observed in a previous work, activation of medial preoptic area camk2α projections to PAG also promotes prey pursuit and hunting[9]. These fascinating data demonstrate that complex neural dynamics are involved in the induction of predation, as activation of this pathway can trigger both inhibition and excitation in l/vlPAG neurons and thus both excitatory and inhibitory PAG inputs can elicit similar behaviors. Similarly, activation of inhibitory ZI neurons induced both excitatory and inhibitory responses in downstream PAG cells[5]. This complexity is further increased considering that it is likely that all these inputs probably target both vgat and vglut2 PAG neurons, as for the central amygdala[11] and ZI[37] inputs. These prior results do not provide an exact mechanism through which the PAG induces approach to food, but they strongly indicate that the relevant circuit elements display considerable functional heterogeneity and complexity, allowing both mono and poly-synaptic influences of excitatory or inhibitory mechanisms originating in the PAG or in PAG inputs to elicit foraging and feeding.

It is noteworthy that prior data have shown that higher activity of PAG gad2 cells decrease feeding and inhibit defensive behaviors[10,22], while our data show that activation of PAG vgat cells increase feeding (Fig. 3) but do not affect defensive actions (Supplementary Fig. 15). This discrepancy is explained by the majority (58%) of PAG gad2 cells do not express vgat (Supplementary Fig. 10), and thus different results may be expected when targeting vgat or gad2 cells in the PAG. Differences in the co-expression of gad2 and vgat have also been reported in other brain areas[38,39].

It is also important to point out that similar to a previous report[1], we observed heterogeneous responses in l/vlPAG vgat neurons. While most l/vlPAG vgat cells showed increases in activity during approach to food, a minority of cells were inhibited during approach (Supplementary Fig. 2), suggesting that l/vlPAG vgat cells may have distinct functions, such that though most cells promote approach, some may inhibit this action. Thus, another potential candidate downstream cells in the l/vlPAG for the long-range inhibitory inputs discussed above include approach-inhibited vgat PAG cells. Therefore, it is possible that inhibitory long-range projections from the central amygdala, ZI, lateral hypothalamus and other regions, are also promoting foraging and feeding by inhibiting vgat PAG cells that inhibit these behaviors. Our view is that the role of the PAG as a downstream region of food-seeking circuits results in part on the dynamic integration of these input modalities by its different cell types.

## Control of food-seeking by the l/vPAG vgat projection to ZI

We show that activation of the inhibitory l/vlPAG vgat projection to the ZI increases foraging. However, prior studies indicate that inhibition of ZI cells decreases food-seeking[7,8]. At first glance these results appear to

be in disagreement, since the l/vlPAG vgat projection releases the inhibitory transmitter GABA. Our dual photometry data indicate that vgat cell bodies in both ZI and PAG are activated during approach to food (Supplementary Fig. 17), showing that higher activity in the inhibitory PAG vgat neurons is not resulting in broad suppression of the ZI during approach to food, and vice versa. Furthermore, optogenetic excitation of l/vlPAG vgat cells increased neural activity in ZI vgat cells (Supplementary Fig. 18). These data suggest l/vlPAG vgat cells are not causing broad inhibition of the ZI. Instead, the data support a model in which l/vlPAG vgat cells increase ZI activity polysynaptically, despite the existence of direct monosynaptic inhibition (Fig. 6g). One possibility is that l/vlPAG vgat cells may preferentially target local ZI inhibitory interneurons, causing disinhibition of ZI cells that cause food-seeking. Indeed, ZI inhibitory interneurons inhibit ZI projection cells[40]. Intriguingly, average in vivo firing rates of ZI cells increase following intra-ZI injections of GABA agonists[41,42]. Thus, both our data and published reports[41,42] indicate that GABAergic transmission within the ZI in vivo can result in higher ZI activity through a polysynaptic mechanism. The most straightforward model to explain how the PAG vgat input increases ZI activity is by hypothesizing that this input is causing disinhibition of ZI interneurons. However, other mechanisms are also possible. For example, PAG vgat cells may co-release an unidentified excitatory neurotransmitter along with GABA.

Our data are compatible with a model in which inhibitory inputs may poly-synaptically increase the activity of downstream targets to induce food-seeking. Intriguingly, emerging data suggests that this mechanism may be a common feature of other feeding circuits. Accordingly, activation of vgat cells in either lateral hypothalamus[43] or peri-locus coeruleus induces feeding[44]. However, counterintuitively, activation of the inhibitory lateral hypothalamus vgat-projection to the peri-locus coeruleus also induces eating[44]. Similarly, activation of inhibitory somatostatin+ cells from the lateral septum induces food approach, and the stimulation of inhibitory terminals from this region increases neural activity in downstream lateral hypothalamus cells[45]. Likewise, here we show higher activity in the PAG vgat neurons can lead to increased activity in ZI cells leading to increased food-seeking. This view is in agreement with reports showing that higher ZI activity increases foraging and food consumption[7,8]. Our data thus add to prior evidence discussed above indicating that GABA-induced disinhibition induced by long-range projections may be a key motif of food-seeking circuits.

Lastly, it is noteworthy that the ZI is highly heterogeneous and contains cells with distinct functions. Accordingly, the ZI participates in numerous behaviors, including feeding[7,8,46], sleep[47], anxiety[40,48], exploratory curiosity[5,6], and express distinct markers[49,50]. Future studies are needed to investigate how PAG vgat inputs to distinct subpopulations of ZI may control unique motivational processes. We propose that such circuits controlling appetitive and consummatory behaviors can be viewed as a broad complex loop network[51,52] of functionally and computationally distinct nodes, requiring further functional characterization.

## Methods

All procedures conformed to guidelines established by the National Institutes of Health and have been approved by the University of California, Los Angeles Institutional Animal Care and Use Committee, protocols 2017-011 and 2017-075. All procedures conform to ARRIVE guidelines.

### Mice

Vgat-Cre mice (Jackson Laboratory stock No. 028862, strain C57BL/6J) were used for all experiments. Male and female mice between 2 and 6 months of age were used in all experiments. Mice were maintained on a 12-hour reverse light-dark cycle with food and water ad libitum at 22–24 °C and 35–55% humidity. All mice were handled for a minimum

of 5 days prior to any behavioral task. vgat-Cre mice were used to direct the GCaMP6s expression in GABAergic neurons, since vgat, or vesicular GABA transporter, is a gene specifically expressed in GABAergic neurons. No animals were excluded. This study is reported in accordance with ARRIVE guidelines.

### Viral Vectors

All vectors were purchased from Addgene, except for AAV5-hSyn-FLEX-TVA-P2A-eGFP-2A-oG and EnvA-dG-rabies-mCherry, which were purchased from the Salk Vector core.

### Surgeries

Ten-week-old mice were anesthetized with 1.5–3.0% isoflurane and affixed to a stereotaxic apparatus (Kopf Instruments). A scalpel was used to open an incision along the midline to expose the skull. After performing a craniotomy, 50 nL of virus was injected into the l/vlPAG using a Hamilton 0.5 μL Neuros Model 7000.5 KH with a beveled needle, and the bevel was placed to face medially (unilateral and counterbalanced for optogenetic activation and fiber photometry experiments, bilateral for inhibition experiments). The syringe was slowly retracted 12 min after the start of the infusion. For l/vlPAG, infusion location measured as anterior-posterior, medial-lateral, and dorso-ventral from bregma were −4.1 mm, ± 1.00 mm, −2.75 mm using a 15-degree angle. For optogenetic activation of vgat l/vlPAG cells, 50 nL of AAV5-Ef1a-DIO-eYFP or AAV5-EF1a-DIO-ChR2(H134R)-eYFP-WPRE-HGHpA of 2.3 ×10$^{13}$ titer was delivered unilaterally to the l/vlPAG of vgat-Cre mice. For optogenetic inhibition, 50 nL of AAV5-Ef1a-DIO-eYFP or AAV9-FLEX-Arch-GFP of 1.0 ×10$^{13}$ titer was delivered unilaterally to the l/vlPAG of vgat-Cre mice. Mice used in optogenetic experiments received a fiber optic cannula (0.22 NA, 200 mm diameter; Doric Lenses) 0.2 mm above viral infusion sites. For ZI optogenetic experiments, fiber optic devices location measured as anterior-posterior, medial-lateral, and dorso-ventral from bregma were −1.95 mm, ± 1.50 mm, −4.25 mm.

For photometry recordings, 50 nL of AAV9.Syn.-Flex.GCaMP6s.WPRE.SV40 of 2.3 × 10$^{*12}$ titer was injected into the l/vlPAG of vgat-Cre mice and an optical fiber was implanted (0.48 NA, 400 mm diameter; Neurophotometrics) 0.2 mm above the injection site or above ZI at the same dorso-ventral coordinate aforementioned. For dual fiber photometry recordings, a similar procedure was performed using GCaMP6f. Contra-lateral to the l/vlPAG GCaMP injection site, an injection of 50nL of AAV9.Syn.Flex.GCaMP6f.WPRE.SV40G was also performed 0.25 mm below the ZI coordinate aforementioned. For the fiber photometry recordings of ZI activity during l/vlPAG vgat optogenetic stimulation, AAV9.Syn.Flex.GCaMP6f.WPRE.SV40G was injected into the ZI, and an excitatory red shifted opsin was expressed in the l/vlPAG by injecting AAV8-nEF-Con/Foff-ChRmine3.3-oScarlet ipsilaterally into the l/vlPAG of vgat-Cre mice. Optic fibers were implanted above the l/vlPAG and ZI injection sites. Dental cement (The Bosworth Company, Skokie, IL, USA) was used to securely attach the fiber optic cannula to the skull. Half the mice in each cage were randomly assigned to YFP/GFP control or ChR2/Arch groups. Only mice with opsin expression restricted to the intended targets were used for behavioral assays.

For miniaturized microscope surgeries, after performing a craniotomy, 50 nl of virus was injected into the l/vlPAG (coordinates in mm, from skull surface): −4.10 anteromedial, ±1.00 lateral, −2.75 depth, 15-degree angle. Five days after virus injection, the animals underwent a second surgery in which two skull screws were inserted and a microendoscope was implanted above the injection site. A 0.5 mm diameter, ~4-mm-long gradient refractive index (GRIN) lens (Inscopix, Palo Alto, CA) was implanted above the l/vlPAG (−2.35 mm ventral to the skull surface) (Reis et al, 2021). The lens was fixed to the skull with cyanoacrylate glue and adhesive cement (Metabond; Parkell, Edgewood, NY, USA). The exposed end of the GRIN lens was protected

with transparent Kwik-seal glue and animals were returned to a clean cage. Two weeks later, a small aluminum base plate was cemented onto the animal's head on top of the previously formed dental cement. Animals were provided with analgesic and anti-inflammatory (carprofen).

For rabies virus tracing, we injected 50 nL of AAV5-hSyn-FLEX-TVA-P2A-GFP-2A-oG virus of titer $8.4 \times 10^{12}$ in the l/vlPAG of vgat-Cre mice. Three weeks later, 200 nL of EnvA-dG-rabies-mcherry of $2.17 \times 10^{8}$ titer was injected in the same location. Mice were perfused 5 days after the injection of the second virus. Images were acquired with a Zeiss confocal microscope (AXIO Observer.Z1/ 7), using a x 10 objective. We counted manually along every slice using Zen Image Processing software the number of starter cells within the l/vlPAG around the injection site (mCherry and GFP double-positive cells). To analyze monosynaptic inputs of vgat+ l/vlPAG neurons we quantified the number of mCherry+ cells from a series of input brain areas with ImageJ software. We use the following nomenclature for the nomenclature for the brain areas evaluated: medial preoptic area (MPA), bed nucleus of the stria terminalis (BST, including all subdivisions), nucleus of the horizontal limb of the diagonal band (HDB), anterior hypothalamus (AH, including the entire anterior hypothalamic area and the latero-anterior hypothalamic nucleus), lateral hypothalamic area (LHa, including the parasubthalamic nucleus), central nucleus of the amygdala (CeA), zona incerta (ZI, we used the posterior hypothalamus to separate the most anterior portion - ZIa, and the most posterior portion - ZIp), ventromedial hypothalamic nucleus (VMH), dorsomedial hypothalamus (DMH), posterior hypothalamic nucleus (PH), premammillary nucleus (PMd), substantia nigra (SN, SNr - reticular part, SNc - compact part, and SNl - latera partl) and cuneiform nucleus (Cun). Sections between the following anterior-posterior coordinates were collected to quantify cells in each of the structures below:

MPA: from Bregma +0.62 mm to −0.22 mm.
BST: from bregma +0.38 mm to −0.10 mm.
HDB: from bregma +0.62 mm to −0.10 mm.
AH: from bregma −0.58 mm to −1.06 mm.
LHa: from bregma −0.46 mm to −2.30 mm.
CeA: from bregma −1.06 mm to −1.70 mm.
ZIa: from bregma −1.06 mm to −1.70 mm.
ZIp: from bregma −1.82 mm to −2.70 mm.
VMH: from bregma −1.22 mm to −1.94 mm.
DMH: from bregma −1.58 mm to −2.06 mm.
PH: from bregma −1.82 mm to −2.54 mm.
PMd: from bregma −2.46 mm to −2.70 mm.
SN:from bregma −2.70 mm bregma −3.64 mm.

For each mouse all quantifications were performed on 40 µm coronal sections stained with DAPI. The outlines of these brain areas were delineated according to the Paxinos Mouse Brain Atlas.

## Perfusion and histological verification

Mice were anesthetized with Fatal-Plus and transcardially perfused with PBS, followed by a solution of 4% paraformaldehyde. Extracted brains were stored for 12 h at 4 °C in 4% paraformaldehyde. Brains were then placed in sucrose for a minimum of 24 h. Brains were sectioned in the coronal plane in a cryostat, washed in phosphate buffered saline and mounted on glass slides using PVA-DABCO. Images were acquired using a Keyence BZ-X fluorescence microscope with a 4x, 10x or 20x air objective.

## Tryptophan hydroxylase 2 (TPH2) immunohistochemistry

Three weeks after microinjecting 50 nL of AAV5-Ef1a-DIO-eYFP in vgat-Cre mice, animals were perfused, the brains were later sliced in a cryostat and coronal sections were collected. A series of sections was processed for immunohistochemistry with the blocking buffer for one hour (1X phosphate buffered saline / 5% donkey serum / 0.3% Triton X-100) and then incubated with primary antibody (TPH2(D3E5I); Rabbit

mAb #51124) at a dilution of 1:1000. After that, brain slices were rinsed three times in 1X phosphate buffered saline for 5 min each. Then, the brain slices were incubated in the secondary antibody (Alexa fluor 594, 1:500) for 2 h. Both antibodies (primary and secondary) were diluted in the blocking buffer. After rinsing three times in 1X phosphate buffered saline for 5 min each, the slices were then stained with DAPI before sections were mounted.

## Optogenetic light delivery

For ChR2 mice, blue light was generated by a 473 nm laser (Dragon Lasers, Changchun Jilin, China) at 4.5 mW unless otherwise indicated. For the experiments that used ChRmine, light was generated by a 589 nm laser at 7 mW (Dragon Lasers). Green light was generated by a 532 nm laser (Dragon Lasers), and bilaterally delivered to mice at 7 mW. Mice were food-deprived for 18 h prior to inhibition experiments. The power of the light was adjusted to the indicated intensity at the tip of the optic fiber before the experiments. A Master-8 pulse generator (A.M.P.I., Jerusalem, Israel) was used to drive the 473 nm and the 589 nm laser pulses of 5 milliseconds at 20 Hz. This stimulation pattern was used for all ChR2 and ChRmine experiments. The laser output was delivered to the animal via an optical fiber (200 µm core, 0.22 numerical aperture, Doric Lenses, Canada) coupled with the fiberoptic implanted on the animals through a zirconia sleeve.

## Behavioral protocols

Preparation of the behavioral tests. On the day of the behavioral test, the animals were transferred to the testing room and habituated to the room conditions for 2 h before the experiments started. All behavioral tests were conducted during the dark phase of the circadian period. For optogenetic experiments, the behavioral measures were taken as the difference between laser off and on epochs for each mouse and then averaged.

## Walnut assay

To investigate behavioral and neural correlates of appetitive and consummatory responses related to feeding, we used walnuts as a high palatable and caloric food resource. Mice were allowed to eat walnuts in their home cages 2 days prior to the test. They were habituated to the test chamber (dimension: 13 × 13 × 25 cm) for 5 min on the day prior to the test. On the test day, mice were introduced to the chamber in the presence of a walnut and allowed to eat it. Each photostimulation consisted of 2-min epochs and presented once, totalling 4-min exposure. Epoch presentation (off or on first) was counterbalanced across mice. Walnut weight was measured before and after each epoch. To test the effects of long periods of stimulation (30-min epoch), mice were tested in two consecutive days in a counterbalanced manner. To test if stimulation temporally restricted to food-seeking is sufficient to increase subsequent consumption, stimulation occurred only during exploration and approach to food, and was turned off at the onset of food consumption. Each epoch lasted for 2 min, totalling 4-min exposure.

## Cricket assay

Mice were trained to hunt crickets during 2 days before the start of experiments. Each mouse was allowed to kill up to 3 crickets on each day. Mice were acclimated to the experimental chamber (dimension: 33 cm diameter, 22 cm walls) for 10 min on the day prior to the test. On the test day, mice were allowed to hunt a single cricket per epoch. During "on" trials, light was delivered in 1-min epochs.The same procedure was conducted during the "off" trials but no light was delivered. Each mouse was submitted to three to five epochs per light condition. For behavioral scoring, latency to first cricket attack was defined as the latency to start the attack and bite towards the cricket. Latency to successful predation was considered the latency to kill or remove a limb of the cricket. Probability of successful predation was calculated

using a binary score (1 for a successful trial and 0 for an unsuccessful trial) if the mouse killed or removed a limb off the cricket during the 60 s epoch.

### Empty chambers

Mice were allowed to acclimate to the chamber (dimension: 13 × 13 × 25 cm) for 2 min before the start of the experiments. In a first experiment mice were allowed to explore the empty chamber and the frequency of rearing behavior was measured. In a following experiment, to investigate context-specific actions, a wired mesh (8 cm wide situated 7 cm above the floor) was added at one of the walls of the chamber allowing the occurrence of climbing behavior. In both assays, blue light was delivered in alternating 1-min epochs for four epochs, totalling 4-min exposure. The frequency of rearing behavior and the latency were quantified during each epoch and analyzed from videos recorded from a side camera. These data are plotted in Fig. 4b, c.

### Holeboard task

Exploratory behavior was analyzed using a modified version of the hole-board apparatus, consisting of a test box (46 cm × 46 cm × 36 cm) and a hole-board frame with 25 holes in a grid-pattern (1.7 cm diameter, 9 cm apart), placed 4 cm above the floor of the testing box. Odor-impregnated bedding from cages of the same sex, which is a strong exploratory motivator, was placed below the hole-board frame. Mice were exposed only once and alternating 1-min epochs for six epochs totalling 6-min exposure. As a measure of exploration, the number of head-dips (hole visits) was measured by visual inspection of an experimenter blind to treatment. These data are plotted in Fig. 4d.

### Real-time place preference

Mice were placed in a two-chamber context (20 × 42 × 27 cm) for 10 min to freely explore the environment. Both chambers are identical. During the next 10 min blue light was delivered to the l/vlPAG of vgat-Cre mice expressing either ChR2 or YFP when they entered one of the chambers. Laser stimulation was only delivered during exploration of the stimulation chamber.

### Following ball

The experiment was performed similarly as previously described[9]. Mice were habituated to a rectangular chamber (46 cm × 46 cm × 36 cm) for 10 min on the day prior to the test. The contours of the letters "B" and "G" were marked on the bottom of the chamber. During the test day, after an acclimation period of 10 min, with mice at the left corner of the chamber, a table tennis ball connected to a stick was gently placed on the start point of the letter "B", and photostimulation started. Each trial consisted in moving the ball so the mouse could follow the "BG" tracks. Speed was adjusted to prevent the mouse from biting the ball. The trial was terminated after the ball completed the "G" track, and photostimulation was discontinued. Video recordings were made of each mouse performing 5 trials for each epoch condition. The tracks of the mice and objects, and the distance between them were analyzed using custom Matlab code (MathWorks, USA).

### Climbing assay

To test how context-specific actions of vgat l/vlPAG stimulation are selected when food is available in the environment, mice were exposed to an arena (dimension: 33 cm diameter, 27 cm walls) in which walls were covered with a wired mesh. In this context, the appetitive stimuli were available on the walls 16 cm above the floor. Thus, mice were able to climb the walls and approach the food. For each mouse, the 'latency to bite walnut' and 'time in walnut corner' were taken as the average across all off and all on trials alternating 1-min epochs for five epochs.

### Object versus walnut preference assay

The mice were habituated to the arena (dimension: 24 cm × 24 cm × 24 cm) on the day prior to the experiment. Unreachable object and a walnut of similar size were hung on the opposite corner of the arena 14 cm above the floor. Blue light was delivered in alternating 3-min epochs for three epochs, totalling 9-min exposure. Mouse behavior at each corner was analyzed from videos recorded from a camera situated at the top of the arena.

### Compulsive eating

The mice were habituated to an experimental chamber (dimension: 12 × 18 × 16 cm, with 50% of the floor area covered by a shock grid) for 10 min and food deprived for 18 h on the day prior to the test. On the test day, a walnut was introduced into the environment on top of the grid and the amount of food eaten was measured during a 5 min baseline period (shock and light were off). Mice could only have access to the walnut by entering the grid area. The walnut was attached to the shocking grid with double-sided adhesive tape. After baseline, a shock generator was turned on and was only turned off at the end of the test. Shocks (0.08 mA, 0.1 s, 1 Hz) were strong enough to decrease the amount of food consumed, but weak enough not to completely abolish the behaviors similarly as previously described[53]. Each shock period (laser off or laser on) lasted 5 min. Walnut weight was measured before and after each period (baseline, shock-laser off, and shock-laser on). In order to evaluate if optogenetic manipulation of vgat-expressing cells can affect walnut intake by reducing the pain caused by foot shocks, the effects of footshocks on time spent on top of the shock grid was also measured in a separate cohort of mice in the absence of walnuts.

### Heated floor pain sensitivity assay

The assay was done on top of a metallic heating plate (14 × 14 cm) (Faithful Magnetic Stirrer model SH-3) that was heated at 50 °C, which is sufficient to cause pain-related reactions in most mice (paw licking or jumping) within one minute. A transparent box (14 × 14 × 24 cm) was placed on top of the heated plate. The latency to display a pain-related reaction was recorded. All mice showed pain responses within 30 s.

### Elevated plus maze

The arms of the elevated plus maze were 30 × 7 cm. The height of the closed arm walls were 20 cm high. The maze is elevated 65 cm from the floor and is placed in the center of the behavior room away from other stimuli. Mice were placed in the center of the elevated plus maze facing a closed arm. For optogenetic experiments, light was delivered in alternating 2-min epochs for five epochs, totalling 10-min exposure.

### Rat assay

We used a long rectangular chamber (70 × 25 × 30 cm) as previously described[25]. Mice were acclimated to this environment for at least two days for 10 min each day. During rat exposure, a live rat is restrained to one end of the chamber using a harness attached to a cable with one end taped to the chamber wall. For the optogenetic experiments light alternated off or on in 3-min epochs for 4 epochs, totalling in a 12-min trial.

### U-shaped maze with walnut and object

The mice were habituated to a U-shaped maze enclosed arena (dimension: 24 × 20 × 13 cm) for 10 min on the day before the fiber photometry experiment. Each corridor (8 cm wide) was separated by an 18 cm wall, and had a window situated on its end. Then, a plastic cap (3 cm diameter) or a piece of walnut was positioned at the end and outside of the window of each corridor. The interaction with the object or the food was limited by a wired mesh positioned at the window. Mouse approaches to the windows were recorded from a camera situated at the top of the maze. The mean df/F activity during approach to object or walnut was used for the quantitative analyses.

## Aggression

Mice were placed in a small chamber of dimensions 29 × 19 ×13 cm for 8 min in the presence of a novel conspecific of the same sex. Baseline was recorded for 5 min and optogenetic stimulation was performed alternating 1-min epochs for three epochs.

## Fiber photometry

Photometry was performed as described in detail previously[54]. Briefly, we used a 405-nm LED and a 470-nm LED (Thorlabs, M405F1 and M470F1) for the $Ca^{2+}$-dependent and $Ca^{2+}$-independent isosbestic control measurements. The two LEDs were band-pass filtered (Thorlabs, FB410-10 and FB470-10) and then combined with a 425-nm long-pass dichroic mirror (Thorlabs, DMLP425R) and coupled into the microscope using a 495-nm long-pass dichroic mirror (Semrock, FF495-Di02-25 3 36). Mice were connected with a branched patch cord (400 mm, Doric Lenses, Quebec, Canada) using a zirconia sleeve to the optical system. The signal was captured at 20 Hz (alternating 405-nm LED and 470-nm LED). To correct for signal artifacts of a non-biological origin (i.e., photo-bleaching and movement artifacts), custom Matlab scripts leveraged the reference signal (405-nm), unaffected by calcium saturation, to isolate and remove these effects from the calcium signal (470-nm). Correlation of df/F activity and threat proximity during EPM test and Rat assay were calculated as previously described[26].

## Simultaneous ZI vgat photometry recording and l/vl/PAG optogenetics

To investigate the effects of l/vlPAG vgat stimulation in ZI vgat neural correlates, we optogenetically stimulated (20 Hz, 5 ms pulses, 2-minute light ON epochs) l/vlPAG vgat cells using ChRmine, an excitatory red-shifted opsin while recording ZI vgat activity. Mice were habituated to the test chamber for 10 min on the day prior to the test. On the test day, mice were introduced to an empty chamber without food, and ZI activity was recorded during l/vlPAG optogenetic stimulation. Each photostimulation consisted of 2-min epochs and were presented five times. Average activity in ZI vgat cells during light ON and light off epochs was calculated by averaging the mean signal for all light ON and light off epochs separately for each mouse, with a total average for both conditions calculated from all 4 mice. Epochs were identified using timestamp files defining laser onset and laser offset for all mice. On-off calculation is the difference in mean df/f (average light ON df/f minus the average light off df/f).

## Miniscope video capture

All videos were recorded at 30 frames/sec using a Logitech HD C310 webcam and custom-built head-mounted UCLA miniscope. Open-source UCLA Miniscope software and hardware (http://miniscope.org/) were used to capture and synchronize the neural and behavioral videos[25].

## Miniscope post-processing

The open-source UCLA miniscope analysis package (https://github.com/daharoni/Miniscope_Analysis) was used to motion correct the miniscope videos[55]. They were then temporally downsampled by a factor of four and spatially downsampled by a factor of two. The cell activity and footprints were extracted using the open-source package Constrained Nonnegative Matrix Factorization for microEndoscopic data (CNMF-E; https://github.com/zhoupc/CNMF_E)[56]. Only cells whose variance was greater than or equal to 10% of the maximum variance among non-outliers were used in the analysis[25]. Neurons were coregistered across sessions using the open-source probabilistic modeling package CellReg (https://github.com/zivlab/CellReg)[57].

## Behavioral quantification

To extract the pose of freely-behaving animals in the described assays, we implemented DeepLabCut[58], an open-source convolutional neural network-based toolbox, to identify mouse nose, ear and tailbase xy-coordinates in each recorded video frame, as well as, when applicable, cricket xy-coordinates. These coordinates were then used to calculate speed and position at each time point.

## Behavioral classification

Approach was defined as movement epochs for which the mouse (1) had continuous movement towards the food source, (2) the distance towards the food source (either walnut or cricket) decreased by at least 10 cm and (3) the movement maintained a minimum velocity of 2 cm/s at all times. Eating was defined in timepoints in which the mouse is visibly biting or swallowing food. 'Rearing' was defined as epochs for which the mouse extended its body up the enclosure wall of either the walnut or cricket assay. This was automatically detected as samples when the mouse head position exceeded a minimum distance from either (1) the center of the circular cricket assay enclosure or (2) a minimum vertical threshold for the walnut eating assay. The DeepLabCut coordinates were not conducive to automatic classification of 'cricket eating', 'walnut approach', or 'walnut eating' behaviors. These behaviors were therefore scored manually by the experimenters.

## Principal component analysis and silhouette score

For individual cricket or walnut sessions, the multidimensional neural activity from each session was reduced to its top three principal components using Matlab function 'pca'. For combined cricket/walnut session analysis, the session pairs were first coregistered using CellReg[57], and the cell activity of putative neurons identified across sessions was first concatenated. The samples that corresponded with 'approach', 'eat', or 'rear' were plotted in 3-dimensional space. The silhouette score was calculated across these three behavioral clusters using the Matlab function 'silhouette'. The Mahalanobis distance between cricket/walnut 'approach' and cricket/walnut 'eat' was calculated using Matlab function 'mahal'. The pairwise Euclidean distance between the dimensionally-reduced 'approach' and 'eat' samples and their mean for concatenated cricket/walnut sessions was calculated using the Matlab function 'pdist'.

## Behavior decoding

Discrete classification of behaviors was performed using multinomial logistic regression. Time-points following approach or eat by 2 s were selected, and a matched number of non-behavior time-points were randomly selected for training and validation. Each time point was treated as an individual data point. Training and validation were performed using five-fold cross-validation, with a minimum of 10 s between training and validation sets. As equal numbers of behavior and non-behavior samples were used to build the training and validation sets, chance accuracy was 50%. Sessions with less than five behaviors were excluded from the analysis.

## Behavior cell classification

We used a Generalized Linear Model (GLM) to identify cells that showed increased calcium activity during each behavior. We fit this model to each cell's activity, with behavior indices as the predictor variable and behavior coefficients as the measure of fit. Behavior onset times were then randomized 100 times and a bootstrap distribution built from the resulting GLM coefficients. A cell was considered a behavior-categorized cell if its coefficient was either greater than (+ cell) or less than (- cell) 95% of the bootstrap coefficient values.

## Brain Slice Preparation and electrophysiological recordings in the zona incerta

Coronal slices were prepared from vgat-Cre mice. Briefly, animals were deeply anesthetized with isoflurane and decapitated with sharp shears. The brains were placed and sliced in ice-cold modified artificial CSF (aCSF) containing the following: 194 mM sucrose, 30 mM

NaCl, 4.5 mM KCl, 1 mM MgCl$_2$, 26 mM NaHCO$_3$, 1.2 mM NaH2PO$_4$, and 10 mM D-glucose, saturated with 95% O$_2$ and 5% CO$_2$. A vibratome (DSK Microslicer; Ted Pella, Inc.) was used to cut 300 μm brain sections. The slices were allowed to equilibrate for 30 min at 33 °C in normal aCSF containing: 124 mM NaCl, 4.5 mM KCl, 2 mM CaCl$_2$ 1 mM MgCl$_2$, 26 mM NaHCO$_3$, 1.2 mM NaH2PO$_4$, and 10 mM D-glucose continuously bubbled with 95% O$_2$ and 5% CO$_2$. Slices were then stored at 21–23 °C in the same buffer until use. All slices were used within 6 h of slicing. Procedures have been described in detail[59,60]. Electrophysiological recordings were performed as described previously for striatal slices[59,61–64]. The brain slice recordings were performed at RT (21–23 °C). Whole-cell patch-clamp recordings were made from neurons in the zona incerta using patch pipettes with a typical resistance of 3–5 MΩ. Neurons were morphologically and electrophysiologically identified. The intracellular solution for neurons membrane properties and evoked comprised the following:135 mM potassium gluconate, 3 mM KCl, 10 mM HEPES, 1 mM EGTA, 0.3 mM Na-ATP, 4 mM Mg-ATP, 0.1 mM CaCl$_2$, 8 mM Na$_2$-phosphocreatine, with pH adjusted to 7.3. To evoke IPSCs specifically from the l/vlPAG vgat cells, AAV5-EF1a-DIO-ChR2(H134R)-eYFP-WPRE-HGhpA was injected in the l/vlPAG and afference in the zona incerta were optogenetically stimulated with an LED light source (Lambda TLED + , sutter instrument) at 2 mW for 5 ms. Neurons were voltage-clamped at −60 mV and pre-incubated with drugs 5 min before recording. Evoked inhibitory postsynaptic currents were recorded in control condition, after incubation with 0.5 μM TTX (Tocris, #1069), 0.5 μM TTX + 100 μM 4-Aminopyridine (Tocris, #0940) or 25 μM bicuculline (Tocris, #0130). All recordings were performed at RT, using pCLAMP11.2 (Axon Instruments, Molecular Devices) and a MultiClamp 700B amplifier (Axon Instruments, Molecular Devices). Cells with access resistance that exceeded 20 MΩ were excluded from analysis. Analysis was performed using ClampFit 10.7 software.

## Mean peak amplitude

To calculate the mean peak amplitude for a cell on a given session, the local maxima were identified from the raw calcium activity (Matlab function 'findpeaks'), and these maxima amplitudes were averaged.

## Merfish data

To identify the cell specificity of vgat and gad2 cells in the PAG, a new analysis of the data set from ref. 65 was performed using the same procedures as previously described.

## Statistics

Nonparametric two-tailed Wilcoxon signed rank or rank-sum tests were used unless otherwise stated using MATLAB. Two-tailed tests were used throughout with α = 0.05. Asterisks in plots indicate the $p$ values. The standard error of the mean was plotted in each figure as an estimate of variation. Multiple comparisons were adjusted with the false discovery rate method. All error bars in the plots represent the standard error of the mean.

## Reporting summary

Further information on research design is available in the Nature Portfolio Reporting Summary linked to this article.

## Data availability

Source Data are provided with this paper. Raw miniscope data is available at the following link: https://datadryad.org/stash/share/cWBUCf87LWhzH9NTAfZwi26nEpWueEOd3drFnQdgXpA. Source data are provided with this paper.

## Code availability

The associated code can be found at https://github.com/schuettepeter/vgat_dpag.

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

## Acknowledgements

This work was supported by the National Institute for Mental Health (R00 MH106649 and R01 MH119089) (A.A.), the National Diabetes and Digestive and Kidney Diseases (R01DK139605 to A.A.), the Brain & Behavior Research Foundation (Grants # 29204 and # 22663 respectively to J.C.K., A.A., and # 27654, # 31093 to F.M.C.V.R.), the National Science Foundation (NSF-GRFP DGE-1650604, P.J.S.), the UCLA Affiliates fellowship (P.J.S.) and the Hellman Foundation (A.A.). We thank Judy

Genshaft and Steven Greenbaum for funding the 2022 Young Investigator Award (Grant # 31093, F.M.C.V.R.) and The Nancy and Jon Glaser Family Friends of Semel Scholar Award (F.M.C.V.R.). A.T. was supported by F31MH127943-01A1. B.A.M. was supported by an NSF-GRFP award. JMI was supported with FAPESP grants #2016/17329-3 and #2019/17677-0. AHK was supported with grants from FAPESP #2019/17892-8 and CNPq #312047/2017-7. We thank Prof. C. Dulac for mentoring E. Vaughn and guiding the generation and analysis of the PAG MERFISH dataset.

## Author contributions

FMCVR performed behavioral experiments, surgeries, experimental and analysis design, conceptualized the project and wrote the paper. SMP performed behavioral experiments and data analysis. PJS performed experimental analysis and made the figures. MO and BK performed the electrophysiology experiments and analyzed the data. EV analyzed the MERFISH data. ES performed behavioral experiments and histology. DCTN, FTHY and EI performed histology. MS, JMI and AT assisted in behavioral and viral tracing experiments. BK, AJS, AHK and JCK provided mentoring for students and advice on data analysis. MC performed behavioral experiments. MS and AJS assisted with rabies tracing. BAM performed data analysis on dual site photometry and optogenetics-related recordings. AA wrote the paper and guided experimental and analysis design.

## Competing interests

The authors declare no competing interests.
