## [Peer Review File · Nature Communications]

Control of feeding by a bottom-up midbrain-subthalamic pathwayREVIEWER COMMENTS:

Reviewer #1 (Remarks to the Author):

In this study, Reis and colleagues record calcium transients from vgat-expressing GABAergic l/vIPAG cells through implanted miniaturized microscopes in free-moving animals and report that vgat-expressing GABAergic l/vIPAG cells encode approach to food and consumption of both live prey and non-prey foods. More specifically, with optogenetic stimulation and inhibition approaches, the authors discover that vgat l/vIPAG activation produces exploratory foraging and compulsive eating without altering defensive behaviors. Furthermore, the vgat l/vIPAG projection to the zona incerta controls the approaches towards food leading to consumption. Hence, the authors conclude that vgat l/vIPAG neurons involve both exploratory foraging and pursuit of prey and non-prey foods; and vgat l/vIPAG neurons can influence these behaviors through a bottom-up pathway.

Food foraging and consumption are fundamental and composed of a broad range of behaviors that include environmental risk assessment, caloric resource detection, pursuit strategy, mastication, and digestion. It is one primary interest in ethology and neuroscience. The present manuscript could have contributed to our understanding of the topic. However, several limitations need to be addressed before precluding a firm conclusion.

Major:

1. In Fig. 1, both the walnut and cricket assays show that vgat l/vIPAG cells were active during exploration and food-seeking and were negatively modulated during the consummatory phase, regardless of food type. However, in Fig. 3, optogenetic inhibition of vgat l/vIPAG cells decreases walnut consumption while stimulation increases. The authors should explain the incoherence between the recording and the behavioral tests.

2. The same issue as above, in Fig. 7, vgat l/vIPAG projection to zona incerta is inhibited during eating; however, in Fig. 8, optogenetic inhibition of vgat l/vIPAG projection to zona incerta reduced appetite.

3. In Fig. 6, rabies tracing from vgat l/vIPAG neurons indicates direct input from the bed nucleus of the stria terminalis (BST), medial preoptic area (MPA), central amygdala (CeA), Zona incerta anterior and posterior regions (ZIa and ZIp, respectively), lateral hypothalamic area (LHa) and cuneiform nucleus (Cun). Previous studies indicated that GABAergic projections from CeA (Han et al., 2017), ZI (Zhao et al., 2019), MPA (Park et al., 2018), and BST (Hao et al., 2019) promote hunting and/or eating, which means inhibiting vgat l/vIPAG neurons would also promote hunting and/or eating. However, data from Fig.3 shows activating these cells decreases latency to eat walnut and attack crickets, as well as increase walnuts consumption. The authors should address these major conflicts with existing pieces of literature.

Reviewer #2 (Remarks to the Author):

In this manuscript, Reis and colleagues studied how the lateral and ventrolateral periaqueductal gray (l/vIPAG) vgat+ neurons encode eating behaviour in mice. In the calcium-imaging and encoding experiment, the authors show that the l/vIPAG vgat+ neurons encode the approaching behaviour during eating. In the gain/off-function studies, using a variety of behavioral quantification methods, the authors find that l/vIPAG vgat+ neurons and their projections to zona incerta (ZI) represent foraging and consumption behaviours, both sufficiently and necessarily. In general, these studies are interesting. However, the authors did not provide a reasonable explanation for the bottom-up modulation for food-seeking. The following concerns should be addressed before the manuscript can be judged for publication.

Major concerns:

1. In the encoding experiment, the authors decode the activity of l/vIPAG vgat+ neurons predominantly with approaching and eating behaviours, which are two different states for food.

However, they are also different in other ways, such as speed. How can the decoding model rule out the influences of other factors?

2. In fig3 e-f, optogenetic activation of I/vIPAG vgat+ neurons shortens the latency for food. In other words, this manipulation should accelerate the locomotion. Optogenetic activation of I/vIPAG vgat+ neurons also promotes exploratory behaviour (fig4 b-d), which is also likely to promote locomotion. However, in Supplementary Fig. 6, the authors show that optogenetic activation of I/vIPAG vgat+ neurons has no effect on locomotion velocity. Please clarify these inconsistency. In addition, did the authors test whether activation of I/vIPAG vgat+ neurons can initiate locomotion?

3. The authors showed that the projection from I/vIPAG vgat+ neurons to ZI contribute to food-foraging behaviour. Since that the projection from ZI to PAG have been proven the pivotal circuit for foraging and hunting in mice, the authors suggested that this is a bidirectional function to control food-approaching behaviour. However, neurons in ZI involved in food seeking are GABAergic, how could those ZI GABAergic neurons function with I/vIPAG vgat+ neurons? The authors have mentioned that the LH cam2 α , which is also the inputs to I/vIPAG, may be excitatory. But there are more inhibitory inputs from LH, CeA and BNST, which are crucial cells for food foraging. How these inhibitory inputs anatomically and functionally interact with I/vIPAG vgat+ neurons, monosynaptically or polysynaptically?

Similarly, the authors showed that activation of the projection from I/vIPAG vgat+ neurons to ZI promote food-consumption (fig 8). How the GABAergic PAG neurons interact with ZI neurons, since activation of ZI GABA neurons promote seeking for food? Thus, a reasonable explanation for the bottom-up modulation on food-seeking is absolutely needed.

4. In the fiber photometry experiment, the authors recorded the calcium signals of the terminals from PAG vgat neurons to ZI during food-approaching, and find this projection was more active prior to the onset of eating. It is notable that the signal was increased during approaching to the live cricket. Thus, these signals maybe associate with locomotion, and more analysis on speed is necessary?

5. This manuscript is focused on the PAG-ZI and food-consumption. These results in fig 5 are about the defensive behaviour, not so closely related to the main topic of manuscript. Maybe this part can be moved to the supplementary figures.

Minor concerns:

1. Periaqueductal gray, anatomic interface between the forebrain and the lower brainstem, is located in the midbrain, but not brainstem itself. So, it may not suitable to term PAG as to be 'brainstem' (line1, line 45, and so on).

2. In experiments to measure compulsive eating behaviour, whether the food in the shocking-position is free to move when the mice hold and bite, or not? How much cost that the mice should be paid for food adventure?

3. line 214, "the I/vIPAG is also known to be a central node for the coordination of innate defensive behaviors". This is an important statement. Please cite the original literatures.

Reviewer #3 (Remarks to the Author):

In the manuscript "Control of feeding by a bottom-up brainstem-subthalamic pathway", the authors employed state-of-the-art techniques to investigate the role of vgat-expressing GABAergic I/vIPAG cells in exploration, foraging and hunting behaviors. They found that activation of I/vIPAG GABAergic

cells produces exploratory foraging and compulsive eating without altering defensive behaviors. They further identified zona incerta as the downstream nucleus of PAG in such feeding-related behaviors. The results are interesting, but there are a number of concerns including data interpretations and experimental details that need to be addressed before the consideration of publication.

Major points

1. A major concern is that the authors' manipulations in the l/vIPAG are extensive - i.e. hit many neurons in and around the l/vIPAG. The broad nature of these manipulations diminishes the value of the findings and related to that, raises several questions. For example, how do the authors know that the effects they see are indeed mediated by l/vIPAG neurons, but not a neighboring nucleus dorsal raphe (DR)? Since studies have shown similar results that activation of vgat-expressing DR neurons induces feeding behavior, the authors need to clarify their optogenetic manipulation is restricted in l/vIPAG. A Series of sections representing virus expression patterns in manipulated areas is needed.
2. It is somewhat inconsistent with Tovote et al. (2016) who shown that inhibition of GABAergic neurons in vIPAG induced freezing. Authors need at least to discuss the possible reasons underlying these differences.
3. Previous studies including Han et al. (2017), Li et al. (2018), Zhao et al. (2019), showed that excitation of GABAergic inputs from the CeA, LH, and ZI respectively to the l/vIPAG triggers predation. In addition, a study by Hao et al. (2019) showed inhibition of anterior vIPAG GABAergic cells produces feeding behavior. Results from these prior studies are opposite to the results represented here which indicate activation of GABAergic l/vIPAG cells evokes feeding and predation. A careful discussion to compare the differences between this study and other is necessary.
4. Some references were incorrectly cited. For example, from lines 40-42, the author mentioned activation of ZI cells causes eating both prey and non-prey foods. While the cited reference Zhao et al. (2019) only observed predation, but not eating of prey. Please cite the correct reference.
5. The amount of walnut consumed by activation of l/vIPAG vgat-expressing cells is minor (around 80-90 mg). This review understands that this result may be due to the walnut assay being only recorded for 4-min. To confirm it is indeed feeding, the authors may want to test walnut-consuming experiments for a longer time, such as half an hour.
6. For the relationship between ZI and l/vIPAG, are neurons in l/vIPAG receiving GABAergic inputs from ZI innervated to the l/vIPAG projecting ZI neurons, or other l/vIPAG non-projecting ZI neurons?
7. Since ZI and l/vIPAG are projected to each other reciprocally, what's the activation sequencing between these two nuclei during predation?
8. Electrophysiological recording (e.g. through slice physiology with TTX + 4-AP) to confirm of monosynaptic projection from l/vIPAG to ZI is missing.
9. In figure 4i the author explained the results by compulsive eating. Since studies have shown that PAG participates in analgesia, could this result be interpreted by optogenetic manipulation vgat-expressing cells reduce the pain caused by foot shock, therefore does not affect walnut intake?

Minor points

1. On line 946, "cgat cells" should be "vgat cells"

REVIEWER COMMENTS

Reviewer #1 (Remarks to the Author):

In this study, Reis and colleagues record calcium transients from vgat-expressing GABAergic l/ vIPAG cells through implanted miniaturized microscopes in free-moving animals and report that vgat-expressing GABAergic l/vIPAG cells encode approach to food and consumption of both live prey and non-prey foods. More specifically, with optogenetic stimulation and inhibition approaches, the authors discover that vgat l/vIPAG activation produces exploratory foraging and compulsive eating without altering defensive behaviors. Furthermore, the vgat l/vIPAG projection to the zona incerta controls the approaches towards food leading to consumption. Hence, the authors conclude that vgat l/vIPAG neurons involve both exploratory foraging and pursuit of prey and non-prey foods; and vgat l/vIPAG neurons can influence these behaviors through a bottom-up pathway.

Food foraging and consumption are fundamental and composed of a broad range of behaviors that include environmental risk assessment, caloric resource detection, pursuit strategy, mastication, and digestion. It is one primary interest in ethology and neuroscience. The present manuscript could have contributed to our understanding of the topic. However, several limitations need to be addressed before precluding a firm conclusion.

Major:

1. In Fig. 1, both the walnut and cricket assays show that vgat l/vIPAG cells were active during exploration and food-seeking and were negatively modulated during the consummatory phase, regardless of food type. However, in Fig. 3, optogenetic inhibition of vgat l/vIPAG cells decreases walnut consumption while stimulation increases. The authors should explain the incoherence between the recording and the behavioral tests.

We thank the reviewer for this important observation. The recording shows that there is higher l/vIPAG vgat neural activity during exploration and hunting, but lower activity during consumption. These data indicate that activation of these cells may increase exploratory foraging related actions. Indeed, activation of l/vIPAG vgat cells increases exploratory foraging and pursuit of food (Figure 4b-h). Our interpretation of these data is that higher l/vIPAG vgat neural activity positively correlates with and causes foraging and food-seeking actions. Thus, the direct immediate effect of exciting l/vIPAG vgat cells is to increase the propensity to forage. In turn, an increase in foraging indirectly causes more food consumption. Consequently, optogenetically increasing l/vIPAG vgat activity causes an increase in consumption because eating is an indirect secondary effect of foraging. We realize that in some portions of the manuscript the language used indicated that l/vIPAG vgat activity directly causes eating. We have now corrected this issue, to indicate that the primary effect of activating these cells is exploratory foraging and appetitive drive.

Furthermore, as the Reviewer noted, there is a decrease in the average activity during the actual consummatory phase. This result indicates that higher l/vIPAG vgat activity does not cause consumption. This behavior is likely controlled by a network that does not include the l/vIPAG vgat cells. This network may include a subset of PAG glutamatergic cells along with regions that have been demonstrated to cause consumption behaviors related to hunting (biting), such as gabaergic cells in the lateral hypothalamus.

2. The same issue as above, in Fig. 7, vgat l/vIPAG projection to zona incerta is inhibited during eating; however, in Fig. 8, optogenetic inhibition of vgat l/vIPAG projection to zona incerta reduced appetite.

Our interpretation for this result is similar to what has been discussed above in question 1. Higher activity in the l/vIPAG vgat projection to the zona incerta creates an increased propensity to forage. Indeed, we see higher activity in this projection specifically during approach to food (Figure 7h-k). An indirect effect of increased foraging is higher consumption. However, consumption is not driven by activity in the projection to the zona incerta, as

there is lower activity in this projection during consumption. We have revised the language of the manuscript to indicate that the direct effect of higher activity in the l/vIPAG vgat projection to the zona incerta is increased approach to food, not higher consumption.

3. In Fig. 6, rabies tracing from vgat l/vIPAG neurons indicates direct input from the bed nucleus of the stria terminalis (BST), medial preoptic area (MPA), central amygdala (CeA), Zona incerta anterior and posterior regions (ZIa and ZIp, respectively), lateral hypothalamic area (LHa) and cuneiform nucleus (Cun). Previous studies indicated that GABAergic projections from CeA (Han et al., 2017), ZI (Zhao et al., 2019), MPA (Park et al., 2018), and BST (Hao et al., 2019) promote hunting and/or eating, which means inhibiting vgat l/vIPAG neurons would also promote hunting and/or eating. However, data from Fig.3 shows activating these cells decreases latency to eat walnut and attack crickets, as well as increase walnut consumption. The authors should address these major conflicts with existing pieces of literature.

We thank the Reviewer for giving us the opportunity to discuss this issue. The marker vgat is pan-GABAergic, and thus includes all GABAergic subtypes. It has been shown in other regions that GABAergic cells are highly heterogeneous and can include cells with different and even opposing functions. For example, in the central amygdala, somatostatin cells increase freezing while CRF cells increase flight, even though both cell types are GABAergic (Fadok et al., 2017). Additionally, GABAergic cells in cortex and hippocampus include multiple distinct cell types, such as parvalbumin and somatostatin-expressing cells, among other types. These cortical GABAergic cell types are known to have unique functions. Similarly, recent evidence indicates that PAG GABAergic and glutamatergic cells are also highly diverse in both expression of unique markers and in their anatomical location within the PAG (Vaughn et al., 2022). For example, cholecystokinin-expressing cells are a well-characterized vgat subtype in cortex, but cholecystokinin cells are actually a subtype of vglut2-expressing glutamatergic cell in the PAG (La-Vu et al., 2022).

Another important issue to consider is that the inhibitory inputs from the central amygdala, ZI, MPA, LH, BNST and other regions can exert their behavioral functions by targeting different types of cells in the PAG (i.e., gad2, vgat and vglut2 neurons).

Han et al., (2017) showed that the central amygdala projections to PAG that control hunting has monosynaptic connectivity with both vgat and vglut neurons. Thus, one possibility is that the majority of these inputs target excitatory vglut2 PAG cells. It is well-established that vglut2 PAG cells promote high fear states during which foraging and feeding is inhibited. Thus, inhibiting vglut PAG cells is likely a mechanism for increased foraging.

Hao et al., (2019) showed that the BNST is monosynaptically connected with gad2 neurons in the PAG, which is a specific neural population in the PAG and different from vgat cells (almost 50% of gad2 cells do not express vgat, see new Supplementary figure 10).

Zhao et al., (2019) showed that ZI GABAergic projections to PAG can promote hunting and that these effects can be blocked by pan-neurally activating downstream neurons using hM3D(Gq). Additionally, Chou et al., (2018), showed that PAG vglut2 cells are inhibited by zona incerta projections to the PAG. Another study that investigated the role of ZI GABAergic projections to PAG found that the activating of this pathway can cause both inhibition and excitation of l/vIPAG neurons (Ahmadlou et al., 2021).

As correctly noted by the Reviewer, activation of MPA projections to PAG also promotes prey pursuit and hunting. These fascinating data demonstrate that complex neural dynamics are involved in the induction of predation, as activation of this pathway can trigger both IPSP (in ~20% of the cells) and EPSP (in ~80% of cells) in l/vIPAG neurons (Park et al., 2018) and both excitatory and inhibitory PAG inputs can elicit similar behaviors. This complexity is further increased considering that it is likely that all these inputs probably target both vgat and vglut2 PAG neurons.

Our data indicates that there likely is functional heterogeneity among PAG vgat cells. For example, though most l/vIPAG vgat cells show increased activity during foraging, there are also cells that are inhibited during approach (Supplementary Figure 2). Our view is that the role of the PAG as a downstream region of hunting and/or eating circuits results in part on the dynamic integration of these input modalities by its different cell populations.

We have now discussed the important issue outlined by the Reviewer to allow readers to more clearly interpret our data and how it can be placed among other published findings.

Reviewer #2 (Remarks to the Author):

In this manuscript, Reis and colleagues studied how the lateral and ventrolateral periaqueductal gray (l/vIPAG) *vgat*⁺ neurons encode eating behaviour in mice. In the calcium-imaging and encoding experiment, the authors show that the l/vIPAG *vgat*⁺ neurons encode the approaching behaviour during eating. In the gain/off-function studies, using a variety of behavioral quantification methods, the authors find that l/vIPAG *vgat*⁺ neurons and their projections to zona incerta (ZI) represent foraging and consumption behaviours, both sufficiently and necessarily. In general, these studies are interesting. However, the authors did not provide a reasonable explanation for the bottom-up modulation for food-seeking. The following concerns should be addressed before the manuscript can be judged for publication.

Major concerns:

1. In the encoding experiment, the authors decode the activity of l/vIPAG *vgat*⁺ neurons predominantly with approaching and eating behaviours, which are two different states for food. However, they are also different in other ways, such as speed. How can the decoding model rule out the influences of other factors?

This is an important concern, because as noted by the Reviewer, there are differences in speed between approach and eating. To address this issue, we show that l/vIPAG cells cannot decode speed above chance levels, and *vgat* PAG activity does not correlate with speed. These results indicate that PAG *vgat* cells are indeed encoding approach to food, rather than general locomotion or speed (Supplementary figure 7).

Figure R1. (a) Scheme of open field assay. (b) Diagram of mouse with implanted miniscope. (c) Left: Correlation of speed and df/F (z-scored) for an example open field session. Speed was not significantly correlated with calcium activity. Right: The correlation of speed and df/F is not significantly different from zero for animals in the open field. ($n=4$; Wilcoxon signed rank test). (d-e) A generalized linear model was trained with miniscope-recorded neural data (predictor variable) and mouse speed or acceleration in the open field assay (response variable). The model was then used to predict speed or acceleration, given a withheld test dataset, and performed at a level no better than chance (Wilcoxon signed rank test; $n=4$).

2. In fig3 e-f, optogenetic activation of I/vIPAG vgat+ neurons shortens the latency for food. In other words, this manipulation should accelerate the locomotion. Optogenetic activation of I/vIPAG vgat+ neurons also promotes exploratory behaviour (fig4 b-d), which is also likely to promote locomotion. However, in Supplementary Fig. 6, the authors show that optogenetic activation of I/vIPAG vgat+ neurons has no effect on locomotion velocity. Please clarify these inconsistency. In addition, did the authors test whether activation of I/vIPAG vgat+ neurons can initiate locomotion?

As requested by the Reviewer, we now show that activation of I/vIPAG vgat cells does not change locomotion relative to light stimulation onset in an empty open field (Figure R2. This result likely was observed because there is no salient feature to explore in an empty open field. Interestingly, performing the same manipulation in the hole board test produces a marked change in locomotion speed (Figure R3), because the holes in this test are spread along the entire environment, so exploring all the holes requires increased locomotion. However, there are also situations in which changes in speed are not necessary for exploration. In another assay (see Figure 4g), we suspended a walnut from the top. In this assay, the focus of the exploration is the walnut, and thus during optogenetic activation of vgat cells mice spend significantly more time near the walnut, trying to reach it and sniff it. This sort of exploration does not require changes in motion, and consequently, optogenetic activation of vgat cells does not induce changes in locomotion in this assay either.

Taken together, these results show that activation of I/vIPAG vgat cells does not induce locomotion per se, but rather elicits exploration of the available stimuli of interest, which may be holes in the holeboard task, an object or a hanging walnut in the other task. An increase in speed during optogenetic stimulation was only observed when the exploratory action requires higher locomotion. Consequently, optogenetic stimulation did not alter speed in the empty open field (an environment that does not have anything to explore) or in the hanging walnut assay (where exploration of the walnut does not require increased movement). Similarly, endogenous I/vIPAG vgat activity also does not correlate with speed, and their neural activity cannot be used to decode speed (see above in Figure R1).

We have added supplementary analysis and text to the discussion considering how vgat neural activity relates to overall locomotion speed.

Figure R2. The trace on the left shows the mean speed (+/- s.e.m) of ChR2 mice in the open field assay, centered at laser onset. The speed pre- and post-laser onset were not significantly different (Wilcoxon signed rank test; n=10)

Figure R3. (a) Activation of I/vIPAG vgat cells increased the mean speed in the hole board test. (Wilcoxon rank sum test; YFP n=6, ChR2 n=6). **(b)** Bars depict the difference in mean speed (ON-OFF) in the hanging walnut assay. (Wilcoxon rank sum test; YFP n=6, ChR2 n=5). * p<0.05.

3. The authors showed that the projection from I/vIPAG vgat+ neurons to ZI contributes to food-foraging behaviour. Since that the projection from ZI to PAG have been proven the pivotal circuit for foraging and hunting in mice, the authors suggested that this is a bidirectional function to control food-approaching behaviour. However, neurons in ZI involved in food seeking are GABAergic, how could those ZI GABAergic neurons function with I/vIPAG vgat+ neurons? The authors have mentioned that the LH cam2 α , which is also the inputs to I/vIPAG, may be excitatory. But there are more inhibitory inputs from LH, CeA and BNST, which are crucial cells for food foraging. How these inhibitory inputs anatomically and functionally interact with I/vIPAG vgat+ neurons, monosynaptically or polysynaptically?

We thank the Reviewer for the opportunity to address this important question. Indeed, as noted by the Reviewer prior work has shown that activation of several GABAergic inputs to the PAG induce food-foraging and hunting. However, it is important to point out that none of those previous reports established that I/vIPAG vgat cells were the relevant downstream target of these long-range inhibitory inputs.

While other models are possible, the simplest explanation is that the inhibitory long range projections from LH, CeA and other regions, are promoting foraging and feeding by inhibiting PAG cells that inhibit these behaviors.

A possible mechanism is that these inputs exert their influence on food-foraging by also targeting excitatory vglut2 I/vIPAG cells. It is well-established that vglut2 PAG cells promote high fear states during which foraging and feeding is inhibited. According to this view, activation of GABAergic inputs from CeA, BNST, LH and ZI may be inducing foraging and feeding by inhibiting fear-promoting PAG vglut2 neurons. In accordance with this possibility, Han et al., (2017) show that glutamatergic I/vIPAG cells are an important downstream target of the CeA-vgat inhibitory pathway that promotes hunting. The results from that work show that activation of this CeA terminal hyperpolarizes vglut2 cells in the PAG. Furthermore, the results from Li et al., (2018) also suggest that a reduction in the activity of PAG vglut2 neurons might be needed for predatory behavior mediated by PAG-projecting LH neurons.

Similarly, PAG vglut2 cells are inhibited by zona incerta projections to the PAG (Chou et al., 2018), providing evidence in favor of the model discussed above. Also, another study that investigated the role of ZI GABAergic projections to PAG found that the activation of this pathway can cause both inhibition and excitation of various I/vIPAG neurons (Ahmadlou et al., 2021), potentially through a polysynaptic mechanism.

As correctly noted by the Reviewer, activation of medial preoptic area-cam2 α projections to PAG also promotes prey pursuit and hunting (Park et al., 2018). These fascinating data demonstrate that complex neural dynamics are involved in the induction of approach behavior during hunting, as activation of this pathway can trigger both IPSP and EPSP in I/vIPAG neurons (Park et al., 2018) and thus both excitatory and inhibitory PAG inputs can elicit similar behaviors.

This complexity is further increased considering that it is likely that all these inputs probably target both vgat and/or vglut2 PAG neurons. Indeed, this has been shown to be the case for CeA (Han et al., 2017) and ZI (Chou et al., 2018) inputs. Taken together, these prior results do not provide an exact mechanism through which the PAG induces approach to food, but they strongly indicate that the relevant circuit elements display considerable functional heterogeneity and complexity, allowing both mono and poly-synaptic influences of excitatory or inhibitory projections to elicit foraging.

It is also important to point out that we observed heterogeneous responses in l/vIPAG vgat neurons. While most l/vIPAG vgat cells showed increases in activity during approach to food, a minority of cells were inhibited during approach (Supplementary Figure 2). This contrast suggests that l/vIPAG vgat cells may have distinct functions, such that though most cells promote approach, some inhibit this action. Thus, another potential candidate downstream cells in the l/vIPAG include approach inhibited vgat cells. As explained above, it is likely that the inhibitory long-range projections from LH, CeA and other regions, are also promoting foraging and feeding by inhibiting PAG cells that inhibit these behaviors. Our view is that the role of the PAG as a downstream region of hunting and/or eating circuits results in part on the dynamic integration of these input modalities by its different cell populations.

We now explore these models in the Discussion section of the manuscript.

(continuation of point 3.) Similarly, the authors showed that activation of the projection from l/vIPAG vgat+ neurons to ZI promotes food-consumption (fig 8). How the GABAergic PAG neurons interact with ZI neurons, since activation of ZI GABA neurons promote seeking for food? Thus, a reasonable explanation for the bottom-up modulation on food-seeking is absolutely needed.

The ZI has been recently found to participate in the regulation of feeding (Zhang and van den Pol, 2017; Zhao et al., 2019), but also sleep (Liu et al., 2017), anxiety (Li et al., 2021; Venkataraman et al., 2019), and exploratory curiosity (Ahmadlou et al., 2021; Ogasawara et al., 2022). Indeed, as stated by the Reviewer, activation of ZI GABAergic neurons induces food-consumption. However, it is critical to consider that ZI cells are highly diverse, and express distinct markers, suggesting that different ZI populations have different functions (Fratzl & Hofer, 2022; Liu et al., 2017; Wang et al., 2020). This diversity feature has been also demonstrated in anxiety assays, where prior data has shown heterogeneity among ZI cells in anxiety, as higher activity of vglut2 and somatostatin cells increase anxiety, while activation of calretinin cells has the opposite effect (Li et al., 2021). Similarly, it is likely that distinct ZI populations have different effects on food consumption. For example, it is expected that higher activity of ZI somatostatin cells suppresses feeding, as these cells induce anxiety, and higher defensive states decrease foraging and consumption. Furthermore, it has also been shown that the ZI contains local interneurons, and activating these cells can poly-synaptically induce both excitation and inhibition. These inhibitory interneurons may decrease consumption by inhibiting consumption-inducing ZI cells (Li et al., 2021). Considering these data, it seems that there may be at least two mechanisms that may decrease food consumption: higher activation of anxiety inducing cells or activation of local inhibitory interneurons. Thus, both anxiety-inducing cells as well as inhibitory interneurons may be putative downstream targets of the inhibitory PAG vgat projection. Inhibition of either of these populations by PAG vgat cells would decrease food-seeking, providing a plausible model to explain our results in the context of prior data.

4. In the fiber photometry experiment, the authors recorded the calcium signals of the terminals from PAG vgat neurons to ZI during food-approaching, and find this projection was more active prior to the onset of eating. It is notable that the signal was increased during approaching to the live cricket. Thus, these signals maybe associate with locomotion, and more analysis on speed is necessary?

As shown for PAG vgat cell body activity (see Figure R1 above), PAG vgat terminals in the ZI also do not show any correlation with speed, as shown below (Figure R4). Lastly, activation of PAG vgat cells in an empty open field does not increase velocity (Figure R2). Taken together, these data show that PAG vgat signals do not control or encode overall locomotion.

Figure R4. (left) Vgat cre mice were injected with a vector encoding cre-dependent GCaMP6s in the I/vIPAG. A fiberoptic cannula was implanted above the zona incerta (ZI) to record calcium transients in I/vIPAG vgat axon terminals in the zona incerta. (middle) Mice were exposed to the environment depicted in the scheme. Mice were able to see and smell the walnut through the wire mesh but were not able to eat or touch it. (right) The recorded fiber photometry signal was not significantly correlated with mouse speed (Wilcoxon signed rank test; n=5).

5. This manuscript is focused on the PAG-ZI and food-consumption. These results in fig 5 are about the defensive behaviour, not so closely related to the main topic of manuscript. Maybe this part can be moved to the supplementary figures.

We have moved this figure to the supplementary material as requested.

Minor concerns:

1. Periaqueductal gray, anatomic interface between the forebrain and the lower brainstem, is located in the midbrain, but not brainstem itself. So, it may not suitable to term PAG as to be 'brainstem' (line1, line 45, and so on).

We thank the Reviewer for this anatomical correction and have substituted 'brainstem' for 'midbrain' throughout the manuscript, including in the title.

2. In experiments to measure compulsive eating behaviour, whether the food in the shocking-position is free to move when the mice hold and bite, or not? How much cost that the mice should be paid for food adventure?

In the compulsive eating experiment the walnut was attached to the shocking grid with double-sided adhesive tape. Consequently, the mouse was unable to collect the food in the shock zone and then bring it to the safe zone for consumption. In order to eat the walnut mice had to endure the shocks. In fact, vgat-cre mice expressing Chr2 in I/vI PAG spent more time on the grid during shock epochs in the presence of a walnut compared to the control group during laser ON (new Supplementary Figure 13). We now increased the methodological detail of this experiment in the "Methods" section.

3. line 214, "the I/vIPAG is also known to be a central node for the coordination of innate defensive behaviors". This is an important statement. Please cite the original literatures.

We have added citations of prior work that shows involvement of the I/vIPAG in defensive behaviors, as requested by the Reviewer.

Reviewer #3 (Remarks to the Author):

In the manuscript "Control of feeding by a bottom-up brainstem-subthalamic pathway", the authors employed state-of-the-art techniques to investigate the role of vgat-expressing GABAergic I/vIPAG cells in exploration, foraging and hunting behaviors. They found that activation of I/vIPAG GABAergic cells produces exploratory foraging and compulsive eating without altering defensive behaviors. They further identified zona incerta as the

downstream nucleus of PAG in such feeding-related behaviors. The results are interesting, but there are a number of concerns including data interpretations and experimental details that need to be addressed before the consideration of publication.

Major points

1. A major concern is that the authors' manipulations in the l/vIPAG are extensive - i.e. hit many neurons in and around the l/vIPAG. The broad nature of these manipulations diminishes the value of the findings and related to that, raises several questions. For example, how do the authors know that the effects they see are indeed mediated by l/vIPAG neurons, but not a neighboring nucleus dorsal raphe (DR)? Since studies have shown similar results that activation of vgat-expressing DR neurons induces feeding behavior, the authors need to clarify their optogenetic manipulation is restricted in l/vIPAG. A Series of sections representing virus expression patterns in manipulated areas is needed.

The Reviewer raises an important potential confound, as higher activity of GABAergic raphe cells also increases consumption, similarly to our data showing that activation of l/vIPAG vgat cells induces foraging. We now provide detailed histological characterization showing that our viral expression spread is restricted to the l/vIPAG throughout the anterior posterior axis of the PAG, without spreading to the adjacent dorsal raphe nucleus (Figure R5).

To further demonstrate that the viral spread did not include the dorsal raphe, we also performed antibody staining of the marker tph2, which is specific to raphe serotonergic cells. We now show that there is no overlap between the targeted l/vIPAG vgat cells and the expression of tph2, which is restricted to the dorsal raphe. (l/vIPAG vgat cells are shown in green and tph+ cells are depicted in red in Figure R6 below).

It is also critical to point out that there are important differences induced by manipulations of PAG vgat and dorsal raphe vgat cells. Prior reports show that optogenetic inhibition of dorsal raphe vgat cells increases locomotion speed (Nectow et al., 2017; Schneeberger et al., 2019). In contrast, in our data, neither activation or inhibition of l/vIPAG vgat cells altered locomotion speed in an empty open field (Supplementary Fig. 7c). These differences also support our assertion that our manipulation did not involve activation of dorsal raphe vgat cells.

In conclusion, data from characterization of viral spread, Tph2 expression and optogenetic activation all provide converging evidence showing that the effects seen by activation of l/vIPAG vgat cells is not due to viral spread to the neighboring dorsal raphe vgat population.

Figure R5. Expression of dio-ChR2 in *vgat* I/vIPAG cells throughout the anterior posterior axis of the PAG. Representative serial images from a single mouse showing expression of YFP (green) restricted to the I/vIPAG.

Figure R6. Viral vector spread was restricted to the I/vIPAG throughout the anterior posterior axis of the PAG and did not spill over into the dorsal raphe. Representative images from a single mouse showing that expression of YFP in I/vIPAG cells (green) was contained within the boundaries of the PAG. The outline of the dorsal raphe is made visible by tyrosine hydroxylase 2 (Tph2) - expressing cells (orange). Note that the green YFP-expressing region shows no overlap with the orange dorsal raphe region throughout a large range of the anterior-posterior axis.

2. It is somewhat inconsistent with Tovote et al. (2016) who showed that inhibition of GABAergic neurons in vIPAG induced freezing. Authors need at least to discuss the possible reasons underlying these differences.

Indeed, as noted by the Reviewer, our results differ from Tovote. The most notable experimental difference between the two reports is that we targeted *vgat* cells while Tovote targeted *gad2* cells. Though both genes are ubiquitously used as GABAergic markers, it is noteworthy that recent evidence indicates that *vgat* and *gad2* cells are not identical populations. For example, it has been shown that in the medial preoptic area in the hypothalamus co-expression of the glutamatergic marker *vglut2* was much higher among *gad2* cells than *vgat* cells (Moffitt et al., 2018). Similarly, in the lateral hypothalamus *gad2* and *vgat* cells are also different populations as 50% of *gad2* cells do not express *vgat* (Kosse et al., 2017). Until recently it was unclear if in the PAG *vgat* and *gad2* cells targeted non-identical populations, as was shown in the medial preoptic area and the lateral hypothalamus.

Interestingly, new analysis of the Vaughn et al., (2022) dataset shows that though the vast majority (92%) of PAG vgat cells co-express gad2, only 42% of gad2 cells co-express vgat, demonstrating that vgat and gad2 PAG cells are not identical populations (Supplementary figure 10), as vgat cells are a sub-population of gad2 cells. This difference is likely a major factor underlying the differences between prior studies (Hao et al., 2019, Tovote et al., 2016), which used a gad2 cre line, and our study with a vgat cre line. Previous work in other regions such as the lateral hypothalamus has also shown that around 50% of gad2 cells do not express vgat, similarly to our findings in the PAG (Kosse et al., 2017). We now discuss these crucial views in the manuscript.

Lastly, despite not observing changes on freezing or locomotion during inhibition of l/vIPAG vgat cells, we did replicate one key result from Tovote, as a complementary analysis of our neural recording data set suggests that l/vIPAG vgat neurons show less activity during freezing, as reported by Tovote's study with gad2 cells.

Figure R7. l/vIPAG vgat cells are inhibited during freezing. (left) Trace shows the mean df/F (+/- s.e.m.) of approach + cells from the walnut assay during freezing to rat. (n cells=25). (right) Bars depict the mean df/F pre- and post-freeze behavior, as shown to the left, during exposure to rat. (Wilcoxon sign rank test; n cells=25) *** p<0.001.

Figure R8. A large fraction of PAG gad2-expressing cells do not co-express vgat. (a) Analysis of the MERFISH data from Vaughn et al., (2022) quantifying the number of gad2, vgat and gad2*vgat expressing cells in the PAG per mouse. (b) Note that the vast majority (92%) of vgat cells co-express gad2. (c) Graph showing that 42% of gad2-expressing cells also co-express vgat. Data are plotted as mean (+/- s.e.m.). n=784863 gad2, 366011 vgat, 342886 vgat.vgat, n=31 mice.

3. Previous studies including Han et al. (2017), Li et al. (2018), Zhao et al. (2019), showed that excitation of GABAergic inputs from the CeA, LH, and ZI respectively to the l/vIPAG triggers predation. In addition, a study by Hao et al. (2019) showed inhibition of anterior vIPAG GABAergic cells produces feeding behavior. Results from these prior studies are opposite to the results represented here which indicate activation of GABAergic l/vIPAG cells evokes feeding and predation. A careful discussion to compare the differences between this study and other is necessary.

We thank the Reviewer for the opportunity to address this important point regarding the complex neural circuit that controls predation and feeding behavior. Indeed, as noted by the Reviewer prior work has shown that activation of several GABAergic inputs to the PAG induce food-foraging and hunting. However, it is important to point out that none of those previous reports established that l/vIPAG vgat cells were the relevant downstream target of these long-range inhibitory inputs. Thus, it is important to consider that the inhibitory inputs from the central amygdala, ZI, medial preoptic area, LH, BNST and other regions can exert their behavioral functions by targeting different cell types in the PAG (i.e., gad2, vgat and vglut2 neurons). While other models are possible, the simplest explanation is that the inhibitory long range projections from LH, CeA and other regions, are promoting foraging and feeding by inhibiting PAG cells that inhibit these behaviors.

A possible mechanism is that these inputs exert their influence on food-foraging by also targeting excitatory vglut2 PAG cells (widely expressed in the l/vIPAG). It is well-established that vglut2 PAG cells promote high fear states during which foraging and feeding is inhibited. According to this view, activation of GABAergic inputs from CeA, BNST, LH and ZI may be inducing foraging and feeding by inhibiting fear-promoting PAG vglut2 neurons.

In accordance with this possibility, Han et al., (2017) show that glutamatergic l/vIPAG cells are an important downstream target of the CeA-vgat inhibitory pathway that promotes hunting. The results from that work show that activation of this CeA terminal hyperpolarizes vglut2 cells in the PAG. The results from Li et al., (2018) also suggest that a reduction in the activity of PAG vglut2 neurons might be needed for predatory behavior mediated by PAG-projecting LH neurons.

Similarly, PAG vglut2 cells are inhibited by ZI projections to the PAG (Chou et al., 2018), providing evidence in favor of the model discussed above. Another study that investigated the role of ZI GABAergic projections to PAG found that the activation of this pathway can cause both inhibition and excitation of various l/vIPAG neurons (Ahmadlou et al., 2021), potentially through a polysynaptic mechanism.

As observed in a previous work, activation of medial preoptic area-cam2 α projections to PAG also promotes prey pursuit and hunting (Park et al., 2018). These fascinating data demonstrate that complex neural dynamics are involved in the induction of predation, as activation of this pathway can trigger both inhibition and excitation in l/vIPAG neurons (Park et al., 2018). These data suggest that both excitatory and inhibitory PAG inputs can elicit similar behaviors.

This complexity is further increased considering that it is likely that all these inputs probably target both vgat and/or vglut2 PAG neurons. Indeed, this has been shown to be the case for CeA (Han et al., 2017) and ZI (Chou et al., 2018) inputs.

An additional layer of complexity is that, our new data show that a large proportion of PAG gad2 cells do not co-express vgat (Supplementary figure 10). Thus, different results are expected when targeting PAG gad2 or vgat cells, and this issue is likely the explanation for the differences between Hao et al and our study.

Taken together, these prior results do not provide an exact mechanism through which the PAG induces approach to food, but they strongly indicate that the relevant circuit elements display considerable functional heterogeneity and complexity, allowing both mono and poly-synaptic influences of excitatory or inhibitory circuits to elicit foraging and feeding.

It is also important to point out that we observed heterogeneous responses in l/vIPAG vgat neurons. While most l/vIPAG vgat cells showed increases in activity during approach to food, a minority of cells were inhibited during approach (Supplementary figure 2). This contrast suggests that l/vIPAG vgat cells may have distinct functions, such that though most cells promote approach, some inhibit this action. Thus, another potential candidate downstream cells in the l/vIPAG include approach inhibited vgat cells. As explained above, it is likely that the inhibitory long-range projections from LH, CeA and other regions, are also promoting foraging and feeding by inhibiting PAG cells that inhibit these behaviors. Our view is that the role of the PAG as a downstream region of hunting and/or eating circuits results in part on the dynamic integration of these input modalities by its different cell populations.

We now explore these models in the Discussion section of the manuscript.

4. Some references were incorrectly cited. For example, from lines 40-42, the author mentioned activation of ZI cells causes eating both prey and non-prey foods. While the cited reference Zhao et al. (2019) only observed predation, but not eating of prey. Please cite the correct reference.

We apologize for this mistake. We have now corrected the description of the results published by (Hao et al., 2019) describing that activation of ZI cells caused predation and eating of high fat food.

5. The amount of walnut consumed by activation of l/vIPAG vgat-expressing cells is minor (around 80-90 mg). This reviewer understands that this result may be due to the walnut assay being only recorded for 4-min. To confirm it is indeed feeding, the authors may want to test walnut-consuming experiments for a longer time, such as half an hour.

We have performed the experiment requested by the Reviewer and we observed that stimulation of l/vIPAG vgat cells for a longer amount of time (30 min) causes a larger increase in food consumption.

Figure R9. Optogenetic activation of I/vIPAG vgat cells for longer times induces a higher amount of food consumption. (Left) Data depicting the amount of walnut consumed during 2 or 3-minute optogenetic activation of ChR2 or YFP -expressing vgat I/vIPAG cells. Data are plotted as mean +-s.e.m. of walnut consumed. Wilcoxon rank sum test; * $p < 0.001$, YFP $n=9$, ChR2 $n=7$).

6. For the relationship between ZI and I/vIPAG, are neurons in I/vIPAG receiving GABAergic inputs from ZI innervated to the I/vIPAG projecting ZI neurons, or other I/vIPAG non-projecting ZI neurons?

We agree with the Reviewer that this would be interesting to evaluate if the same vgat cells in both regions are reciprocally interconnected. However, it is not possible to functionally identify this circuit with cell-type specificity for vgat cells across regions using currently available tools. The issue is that although there are cell-type specific retrograde vectors (such as the rabies-tva system), but there is no cell-type specific anterograde transsynaptic vector that is well-validated and commercially available. Thus, unfortunately, this issue raised by the Reviewer cannot be investigated.

Our main claim is that inhibition of the I/vIPAG vgat projection to the ZI decreases foraging. This is possible regardless of whether the same vgat cells in both structures are monosynaptically connected or not, thus addressing this comment does not impinge on the validity of our conclusions. Of course, knowing the exact connectivity map among individual cells in this circuit would be valuable to determine the mechanism underlying our behavioral effect, but this question cannot be addressed with existing approaches.

7. Since ZI and I/vIPAG are projected to each other reciprocally, what's the activation sequencing between these two nuclei during predation?

The synaptic delay between the PAG and ZI is expected to be less than 15 ms, given the distance between the two structures. Considering the slow kinetics of calcium indicators, it is not possible to reliably find the order of activation of these structures, thus the exact sequence of activation cannot be measured *in vivo* in a freely moving mouse.

Despite the aforementioned temporal limitations of calcium imaging, in order to characterize the activity of the reciprocally connected ZI and I/vIPAG circuit we performed a dual photometry recording experiment in vgat-expressing cells from both regions simultaneously. AAV-FLEX-GCaMP6f was injected in vgat cre mice in the ZI and PAG. Recordings of calcium transients were obtained from vgat ZI and vgat PAG cells from contralateral hemispheres. These data show that in both regions vgat cells are activated during approach to cricket or walnut. This data indicates that the main effect of PAG vgat cells is not to simply inhibit ZI all cells. Rather, these inhibitory interconnected regions are likely acting in complex polysynaptic disinhibitory ways, as their activities are positively correlated during approach. One possibility is that vgat cells in both regions are functionally heterogeneous. Indeed, our single cell data show that though the majority of PAG vgat cells are activated during approach, some cells are inhibited during behavior, suggesting these cells have distinct functions. Similarly, published single cell ZI data also show both excitatory and inhibitory responses during hunting (Zhao et al., 2019). Taken together, these data are compatible with a model in which each region inhibits downstream cells in the other region that decreases foraging, allowing the majority of foraging-inducing cells to initiate foraging behaviors.

Figure R10. Dual photometry recording experiment in vgat-expressing cells from I/vIPAG and ZI. AAV-FLEX-GCaMP6f was injected in the I/vIPAG and ZI of vgat cre mice. Recordings of calcium transients were obtained from vgat ZI and vgat I/vIPAG cell bodies from contralateral hemispheres. The data show that in both regions vgat cells are activated during approach to cricket or walnut.

8. Electrophysiological recording (e.g. through slice physiology with TTX + 4-AP) to confirm monosynaptic projection from I/vIPAG to ZI is missing.

Figure R11. Characterization of monosynaptic GABAergic input from PAG vgat cells to the zona incerta. (A) Relationship of injected current to number of action potentials (APs - left), traces of MSN membrane responses to current injection and membrane properties of neurons in the zona incerta (right). (B) Representative traces of light-evoked IPSCs from zona incerta neurons for the control, 0.5 μ M tetrodotoxin (TTX), 0.5 μ M TTX + 100 μ M 4-Aminopyridine (4AP) and 25 μ M bicuculline (BIC) conditions (from the top to the bottom). (C) Quantitative analysis of the amplitude of light-evoked IPSCs from the zona incerta for the control (n = 12 cells from 5 mice), 0.5 μ M TTX (n = 9 cells from 5 mice), 0.5 μ M TTX + 100 μ M 4-Aminopyridine (n = 9 cells from 5 mice) and 25 μ M bicuculline conditions (n = 6 cells from 3 mice). One-way ANOVA test following by a means comparison Tukey test, NS = Not Significant.

We performed the requested experiment as shown in the figure above, and have characterized the vgat PAG projection to ZI in an *ex vivo* slice preparation. We now show that optogenetic activation of ChR2-expressing vgat PAG terminals evokes an inhibitory current in ZI cells that is abolished in the presence of tetrodotoxin and then rescued after the addition of 4-aminopyridine. We also show that the response is abolished in the presence of bicuculline, demonstrating that PAG vgat cells release GABA in the PAG.

9. In figure 4i the author explained the results by compulsive eating. Since studies have shown that PAG participates in analgesia, could this result be interpreted by optogenetic manipulation vgat-expressing cells reduce the pain caused by foot shock, therefore does not affect walnut intake?

Prior studies have shown that inhibition of PAG vgat cells (Samineni et al., 2017) or excitation of PAG vglut2 cells (Tovote et al., 2016) increase analgesia. In our compulsive eating experiment we optogenetically excited PAG vgat cells. Based on the prior results cited above, this manipulation is not expected to cause analgesia. To test if excitation of PAG vgat cells caused analgesia we performed the heated floor pain sensitivity test, in which the latency of the mouse to display pain induced by heat is measured. Indeed, as predicted from these published results, we observed that excitation of PAG vgat cells did not cause analgesia (see Figure R12 and R13 below). Thus, our results in the compulsive eating assay cannot be explained by analgesia caused by I/vIPAG vgat cells.

To further investigate the role of PAG vgat cells in the compulsive eating assay we repeated this experiment in the absence of the walnut reward. This experiment shows that though ChR2-expressing mice spend more time in the shock grid to obtain the walnut reward, these mice do not spend more time in the shock grid in the absence of reward. This result indicates that PAG vgat cell activation does not decrease sensitivity to the shock, as ChR2 mice avoid the shock grid as much as YFP mice in the absence of walnut. Thus, the increase in eating following excitation of vgat cells is not due to analgesia or pain sensitivity-related processes.

Figure R12. Optogenetic excitation of I/vIPAG vgat cells did not change latency for nociceptive reactions in the heated floor assay. Graph depicting average latency for pain reactions in the heated floor assay for mice expressing either YFP or ChR2 in I/vIPAG vgat cells (Wilcoxon rank sum test; $p > 0.05$, YFP $n = 7$, ChR2 $n = 11$).

Figure R13. Optogenetic activation of I/vIPAG vgat cells increases time on the shock grid only in the presence of walnut reward. Mice were exposed to an environment with a safe floor and a shock grid floor. In the presence of shock, but absence of walnut (first two bars) both YFP and ChR2 mice avoid the shock grid to similar extent, indicating that activation of I/vIPAG vgat cells does not decrease pain sensitivity. We then added walnut reward to the shock grid, and observed that activation of ChR2-expressing I/vIPAG vgat cells increases time spent on the grid (yellow rectangle) (Wilcoxon rank sum test; * $p < 0.05$, $n = 5$ YFP, $n = 7$ ChR2).

Minor points

1. On line 946, “cgat cells” should be “vgat cells

Typo corrected.

References

- Ahmadlou, M., Houba, J. H. W., van Vierbergen, J. F. M., Giannouli, M., Gimenez, G.-A., van Weeghel, C., Darbanfouladi, M., Shirazi, M. Y., Dziubek, J., Kacem, M., de Winter, F., & Heimel, J. A. (2021). A cell type-specific cortico-subcortical brain circuit for investigatory and novelty-seeking behavior. *Science*, 372(6543), eabe9681.
- Chou, X.-L., Wang, X., Zhang, Z.-G., Shen, L., Zingg, B., Huang, J., Zhong, W., Mesik, L., Zhang, L. I., & Tao, H. W. (2018). Inhibitory gain modulation of defense behaviors by zona incerta. *Nature Communications*, 9(1), 1151.
- Fadok, J. P., Krabbe, S., Markovic, M., Courtin, J., Xu, C., Massi, L., Botta, P., Bylund, K., Müller, C., Kovacevic, A., Tovote, P., & Lüthi, A. (2017). A competitive inhibitory circuit for selection of active and passive fear responses. *Nature*, 542(7639), 96–100.
- Fratzl, A., & Hofer, S. B. (2022). The caudal prethalamus: Inhibitory switchboard for behavioral control? *Neuron*, 110(17), 2728–2742.
- Han, W., Tellez, L. A., Rangel, M. J., Motta, S. C., Zhang, X., Perez, I. O., Canteras, N. S., Shammah-Lagnado, S. J., van den Pol, A. N., & de Araujo, I. E. (2017). Integrated Control of Predatory Hunting by the Central Nucleus of the Amygdala. *Cell*, 168(1), 311–324.e18.
- Hao, S., Yang, H., Wang, X., He, Y., Xu, H., Wu, X., Pan, L., Liu, Y., Lou, H., Xu, H., Ma, H., Xi, W., Zhou, Y., Duan, S., & Wang, H. (2019). The Lateral Hypothalamic and BNST GABAergic Projections to the Anterior Ventrolateral Periaqueductal Gray Regulate Feeding. *Cell Reports*, 28(3), 616–624.e5.
- Kosse, C., Schöne, C., Bracey, E., & Burdakov, D. (2017). Orexin-driven GAD65 network of the lateral hypothalamus sets physical activity in mice. *Proceedings of the National Academy of Sciences of the United States of America*, 114(17), 4525–4530.
- La-Vu, M. Q., Sethi, E., Maesta-Pereira, S., Schuette, P. J., Tobias, B. C., Reis, F. M. C. V., Wang, W., Torossian, A., Bishop, A., Leonard, S. J., Lin, L., Cahill, C. M., & Adhikari, A. (2022). Sparse genetically defined neurons refine the canonical role of periaqueductal gray columnar organization. *eLife*, 11. <https://doi.org/10.7554/eLife.77115>
- Liu, K., Kim, J., Kim, D. W., Zhang, Y. S., Bao, H., Denaxa, M., Lim, S.-A., Kim, E., Liu, C., Wickersham, I. R., Pachnis, V., Hattar, S., Song, J., Brown, S. P., & Blackshaw, S. (2017). Lhx6-positive GABA-releasing neurons of the zona incerta promote sleep. *Nature*, 548(7669), 582–587.
- Li, Y., Zeng, J., Zhang, J., Yue, C., Zhong, W., Liu, Z., Feng, Q., & Luo, M. (2018). Hypothalamic Circuits for Predation and Evasion. *Neuron*, 97(4), 911–924.e5.
- Li, Z., Rizzi, G., & Tan, K. R. (2021). Zona incerta subpopulations differentially encode and modulate anxiety. *Science Advances*, 7(37), eabf6709.
- Moffitt, J. R., Bambah-Mukku, D., Eichhorn, S. W., Vaughn, E., Shekhar, K., Perez, J. D., Rubinstein, N. D., Hao, J., Regev, A., Dulac, C., & Zhuang, X. (2018). Molecular, spatial, and functional single-cell profiling of the hypothalamic preoptic region. *Science*, 362(6416). <https://doi.org/10.1126/science.aau5324>
- Nectow, A. R., Schneeberger, M., Zhang, H., Field, B. C., Renier, N., Azevedo, E., Patel, B., Liang, Y., Mitra, S., Tessier-Lavigne, M., Han, M.-H., & Friedman, J. M. (2017). Identification of a Brainstem Circuit Controlling Feeding. *Cell*, 170(3), 429–442.e11.
- Ogasawara, T., Sogukpinar, F., Zhang, K., Feng, Y.-Y., Pai, J., Jezzini, A., & Monosov, I. E. (2022). A primate temporal cortex-zona incerta pathway for novelty seeking. *Nature Neuroscience*, 25(1), 50–60.
- Park, S.-G., Jeong, Y.-C., Kim, D.-G., Lee, M.-H., Shin, A., Park, G., Ryoo, J., Hong, J., Bae, S., Kim, C.-H., Lee, P.-S., & Kim, D. (2018). Medial preoptic circuit induces hunting-like actions to target objects and

- prey. *Nature Neuroscience*, 21(3), 364–372.
- Samineni, V. K., Grajales-Reyes, J. G., Copits, B. A., O'Brien, D. E., Trigg, S. L., Gomez, A. M., Bruchas, M. R., & Gereau, R. W., 4th. (2017). Divergent Modulation of Nociception by Glutamatergic and GABAergic Neuronal Subpopulations in the Periaqueductal Gray. *eNeuro*, 4(2).
<https://doi.org/10.1523/ENEURO.0129-16.2017>
- Schneeberger, M., Parolari, L., Das Banerjee, T., Bhawe, V., Wang, P., Patel, B., Topilko, T., Wu, Z., Choi, C. H. J., Yu, X., Pellegrino, K., Engel, E. A., Cohen, P., Renier, N., Friedman, J. M., & Nectow, A. R. (2019). Regulation of Energy Expenditure by Brainstem GABA Neurons. *Cell*, 178(3), 672–685.e12.
- Tovote, P., Esposito, M. S., Botta, P., Chaudun, F., Fadok, J. P., Markovic, M., Wolff, S. B., Ramakrishnan, C., Fenno, L., Deisseroth, K., Herry, C., Arber, S., & Luthi, A. (2016). Midbrain circuits for defensive behaviour. *Nature*, 534(7606), 206–212.
- Vaughn, E., Eichhorn, S., Jung, W., Zhuang, X., & Dulac, C. (2022). Three-dimensional Interrogation of Cell Types and Instinctive Behavior in the Periaqueductal Gray. In *bioRxiv* (p. 2022.06.27.497769).
<https://doi.org/10.1101/2022.06.27.497769>
- Venkataraman, A., Brody, N., Reddi, P., Guo, J., Gordon Rennie, D., & Dias, B. G. (2019). Modulation of fear generalization by the zona incerta. *Proceedings of the National Academy of Sciences of the United States of America*, 116(18), 9072–9077.
- Wang, X., Chou, X.-L., Zhang, L. I., & Tao, H. W. (2020). Zona Incerta: An Integrative Node for Global Behavioral Modulation. *Trends in Neurosciences*, 43(2), 82–87.
- Zhang, X., & van den Pol, A. N. (2017). Rapid binge-like eating and body weight gain driven by zona incerta GABA neuron activation. *Science*, 356(6340), 853–859.
- Zhao, Z.-D., Chen, Z., Xiang, X., Hu, M., Xie, H., Jia, X., Cai, F., Cui, Y., Chen, Z., Qian, L., Liu, J., Shang, C., Yang, Y., Ni, X., Sun, W., Hu, J., Cao, P., Li, H., & Shen, W. L. (2019). Zona incerta GABAergic neurons integrate prey-related sensory signals and induce an appetitive drive to promote hunting. *Nature Neuroscience*, 22(6), 921–932.

REVIEWER COMMENTS:

Reviewer #1 (Remarks to the Author):

The authors have addressed my concerns.

Reviewer #2 (Remarks to the Author):

The authors addressed all of my concerns in the revision. I support the publication of NCOMMS-22-22181A.

Reviewer #3 (Remarks to the Author):

The author responded to some of my comments satisfyingly which resulted in an improved overall manuscript. However, some points raised by the reviewers have not been fully addressed. Specifically, the authors' results are at variance with the results from many other groups of investigators (Tovote et al. 2016; Han et al. 2017; Zhang et al. 2017; Li et al. 2018; Zhao et al. 2019; Hao et al. 2019; Yu et al. 2021). The discrepancies with results from other labs are not fully addressed.

Major points,

1) The authors proposed that activation of l/vIPAG vgat cells may increase exploratory foraging-related actions, and an increase in foraging indirectly causes more food consumption. However, optogenetic manipulation will time-lock the firing of targeted neurons. Therefore, during food consumption in the behavioral test, these l/vIPAG vgat cells are still activated by optogenetic manipulation. The incoherence between the calcium recording and the behavioral tests is not resolved.

2) Hunting has a specific sequential behavioral action, including prey search, pursuit, attack, and consumption. A recent study (Yu et al. 2021) suggested that IPAGVgat neurons are required for prey detection, chase and attack, while IPAGVglut2 neurons are selectively required for the attack. They also found that photo-inhibition of IPAGVgat neurons has no effects on crickets' consumption, which is opposite to the conclusion of this study. In addition, food and prey approaches are key hallmarks to the analysis of calcium recording and the behavioral tests. Whereas in the methods part, a detailed description of how to determine an approach is not reported. This crucial behavioral readout needs to be unambiguously defined.

3) The explanation for the discrepancy between this study and CeAGABA-vIPAG pathway (Han et al. 2017) in hunting is that the majority of these inputs target excitatory vglut2 PAG cells is not convincing. No pieces of evidence support that GABAergic inputs from CeA only innervated or preferred to project to vglut2 but not vgat PAG cells.

4) The explanation for the discrepancy between this study and Hao et al. 2019 is that gad2 and vgat l/vIPAG cells might be two different cell populations. However, they still have a large number of overlapping cells (gad2 positive and vgat positive). The opposite role of gad2 and vgat l/vIPAG cells in feeding regulation is not fully addressed.

5) The authors explained the discrepancy between this study and prior studies (Zhang et al. 2017; Zhao et al. 2019) showing that activation of ZI GABAergic cells promotes food seeking by that the functional diversity of ZI cells and vIPAG vgat cells may decrease food consumption through modulating ZI anxiety inducing cells or local inhibitory interneurons. The authors proposed that both anxiety-inducing cells, as well as inhibitory interneurons, may be putative downstream targets of the inhibitory PAG vgat projection. Inhibition of either of these populations by PAG vgat cells would decrease food-seeking. However, this study concludes that activation of PAG GABAergic inputs to ZI, that is inhibition of ZI, promotes food-seeking and food consumption, which is controversial to the authors' own hypothesis.

6) The answer to the question of what activation sequencing between ZI is and vIPAG during predation is the cornerstone of this study. Because this question could answer whether it is indeed the bottom-

up modulation of feeding from vIPAG to ZI that exist. Current optogenetic manipulation of I/vIPAG vgat-ZI terminals data could not rule out the possibilities of retrograde activation or inhibition of vIPAG cells.

Minor points

- 1) Video showing optogenetic activation of I/vIPAG vgat cells induces hunting is necessary.
- 2) A related question, in cricket prey test, does optogenetic activation of I/vIPAG vgat cells cause consuming of cricket?

REVIEWER COMMENTS

Reviewer #1 (Remarks to the Author):

The authors have addressed my concerns.

We are happy to know that Reviewer #1 has no remaining concerns.

Reviewer #2 (Remarks to the Author):

The authors addressed all of my concerns in the revision. I support the publication of NCOMMS-22-22181A.

We are delighted to learn that Reviewer #2 supports the publication of our manuscript.

Reviewer #3 (Remarks to the Author):

The author responded to some of my comments satisfyingly which resulted in an improved overall manuscript. However, some points raised by the reviewers have not been fully addressed. Specifically, the authors' results are at variance with the results from many other groups of investigators (Tovote et al. 2016; Han et al. 2017; Zhang et al. 2017; Li et al. 2018; Zhao et al. 2019; Hao et al. 2019; Yu et al. 2021). The discrepancies with results from other labs are not fully addressed.

We now more thoroughly compare our results with the ones cited by the Reviewer.

Zhang et al., 2017¹ and Zhao et al., 2019² showed that excitation of the zona incerta (ZI) increases food-seeking. Here, we show that activation of the inhibitory l/vIPAG vgat projection to the ZI increases food-seeking. We agree with the Reviewer that in this case our work appears to be in disagreement with prior work. We have now performed a new experiment showing that *in vivo* stimulation of l/vIPAG vgat cells results in increased activity of ZI vgat cells (Supplementary Fig. 18 and Fig. R6 below), likely via poly-synaptic disinhibition, though other mechanisms are also possible. Thus, taking into account this new result, our work shows that activation of l/vIPAG vgat input results in higher ZI vgat activity, which leads to increased food-seeking. This model is in full agreement with the data from Zhang et al.¹ and Zhao et al.² showing that higher ZI activity induces more food-seeking.

Hao et al., 2019³ and Tovote et al., 2016⁴ studied gad2 cells in the PAG and found different results from our study on PAG vgat cells. However, as explained in question 4 below (see also Fig. R3 below), the majority (58%) of PAG gad2 cells are not vgat positive. Thus, gad2 and vgat constitute substantially different populations within the PAG. Consequently, it is expected that these two cell populations may have different roles.

Han et al., 2017⁵ showed that activation of the vgat projection from the central amygdala to PAG causes hunting. They also showed that this effect was mediated via inhibition of PAG vglut2 cells (see detailed answer in question 3 below). It is unclear to us why the Reviewer states this result from Han et al.⁵ is in disagreement with our data.

Li et al., 2018⁶ showed that activation of GABAergic projections from lateral hypothalamus to the PAG induces hunting. Zhao et al., 2019² showed that excitation of the GABAergic projection from ZI to PAG also elicits hunting. However, neither Li et al. nor Zhao et al. identified the relevant downstream population in the PAG that mediated those effects. Since Li et al.⁶ and Zhao et al.² did not study any PAG vgat populations, their results cannot be in direct disagreement with us, as they studied different circuits.

Yu et al., 2021⁷ is the only study that investigated PAG vgat cells in food-seeking. As explained in response to

question 2 below, both Yu et al.⁷ and we found that inhibition of PAG vgat cells decreases cricket hunting (Fig. 3h of Yu et al.⁷, and Fig. 3d of our paper). Both of these studies did not measure the amount of cricket eaten. Furthermore, Yu et al. and we report that PAG vgat cells display decreased activity during consumption (Fig. 2c of Yu et al., 2021⁷ and Fig. 2c of our report). Both papers also report that activation of PAG vgat cells causes real-time place preference (Supplementary Fig. 7c of Yu et al., 2021⁷ and Fig. 4e of the current paper). Thus, both Yu et al.⁷ and we found the same results in three comparable and independent experiments (optogenetic control of cricket hunting, recordings of PAG vgat neural activity, and optogenetic induction of real-time place preference).

Lastly, note it is not possible for us to be in agreement with all these papers cited because some of these papers are in disagreement with each other. Most notably, Yu et al., 2021⁷ found that inhibition of PAG vgat cells decreases food-seeking, while Hao et al., 2019³ reported the opposite results with PAG gad2 cells. Considering that the papers cited by the Reviewer are already “at variance” with each other, it is logically impossible for us (or any other paper) to be simultaneously in agreement with both Yu et al.⁷ and Hao et al.³ Instead, the papers cited above demonstrate that gad2 and vgat PAG cells have distinct roles in hunting. Another internal discrepancy in the papers cited by the Reviewer is that Tovote et al., 2016⁴ shows that the central amygdala gad2 projection to the PAG causes freezing, while Han et al., 2017⁵ reports no effect on freezing following the activation of the central amygdala vgat projection to the PAG. Han et al.⁵ states that: “Surprisingly, we failed to observe any occurrences of freezing upon optical stimulation of CeA”. Thus, Han et al.⁵ and Tovote et al.⁴ also support our argument that gad2 and vgat cells can have different roles.

Additionally, Tovote et al.⁴ showed that inhibition of PAG gad2 cells causes freezing. Tovote et al.⁴ state that “Optical inhibition of GAD2+ neurons induced freezing in naive mice”. In stark contrast, Hao et al., 2019³ reports that inhibition of the very same PAG gad2 cells induced more eating without affecting freezing. Hao et al.³ states that “Light activation of vIPAG GABAergic cells did not induce freezing”. Thus, these two papers mentioned by the Reviewer are in fundamental disagreement even when comparing two reports that used the same cre line (gad2) in the same region (PAG), measuring the same behavior (freezing).

Given this state of affairs, it is difficult to see how we can be fully consistent with papers that studied different long-range projections and used different cre lines from our work. As explained above, there are numerous additional inconsistencies between the reports cited by the Reviewer. It is not logically possible for us, or any other paper, to be in agreement with all the papers listed simultaneously, since these papers already are in disagreement with each other on multiple fronts. As explained earlier, we are in complete agreement with Yu et al., 2021⁷ which is the only paper that studied vgat PAG cells in hunting, similarly to our paper.

Major points,

1) The authors proposed that activation of l/vIPAG vgat cells may increase exploratory foraging-related actions, and an increase in foraging indirectly causes more food consumption. However, optogenetic manipulation will time-lock the firing of targeted neurons. Therefore, during food consumption in the behavioral test, these l/vIPAG vgat cells are still activated by optogenetic manipulation. The incoherence between the calcium recording and the behavioral tests is not resolved.

We thank the Reviewer for giving us the opportunity to address this important concern. Indeed, our data show that PAG vgat cells are active during food-seeking, but not during consumption. In contrast, our optogenetic excitation was done both during food-seeking and consumption. We thus performed optogenetic excitation that mimics the endogenous pattern of PAG vgat activation explained above.

Our data show that l/vIPAG vgat cells are more active during food-seeking, while they are mostly inhibited during consumption. We performed a new experiment where we optogenetically activated these cells only during exploration and approach to food, and then turned off the laser stimulation at the onset of food consumption.

This assay (See Figure R1 below) revealed that optogenetic activation of I/vIPAG vgat cells restricted to exploration and food approach was sufficient to increase consumption even though the stimulation was turned off during consumption. These results mirror important aspects of the endogenous neural activity of these cells and show that activating them causes consumption as an indirect downstream effect, as suggested by us previously. These data also show that optogenetic stimulation during consumption is not necessary to increase feeding, as stimulation temporally restricted to food-seeking is sufficient to increase subsequent consumption.

Figure R1 (Supplementary Fig. 6). Activation of I/vIPAG vgat cell activity increases approach to and consumption of food, but not of water. (a) Scheme showing experimental approach for expressing excitatory (ChR2) opsin in I/vIPAG vgat cells. **(b)** Optogenetic excitation with blue light was only delivered during the ON epoch during exploration and approach. Blue light was not delivered during consumption to emulate the endogenous neural activity of these cells, as they show lower activity during consumption (see Fig. 2). **(c)** Optogenetic excitation of I/vIPAG vgat cells increased the amount of food eaten, **(d)** decreased latency to eat food and increased the amount of time spent eating. (YFP n=8, ChR2 n=8, left: Wilcoxon rank sum p<0.009, 0.004 and 0.004, respectively for food eaten, latency and time spent plots).

2) Hunting has a specific sequential behavioral action, including prey search, pursuit, attack, and consumption. A recent study (Yu et al. 2021) suggested that IPAG vgat neurons are required for prey detection, chase and attack, while IPAG vglut2 neurons are selectively required for the attack. They also found that photo-inhibition of IPAG vgat neurons has no effects on crickets' consumption, which is opposite to the conclusion of this study. In addition, food and prey approaches are key hallmarks to the analysis of calcium recording and the behavioral tests. Whereas in the methods part, a detailed description of how to determine an approach is not reported. This crucial behavioral readout needs to be unambiguously defined.

To the best of our knowledge neither us nor Yu et al., 2021⁷ reported the effect of photo inhibition of IPAG vgat cells on cricket consumption. Yu et al.⁷ only reported that photo inhibition during eating did not impact eating duration when the laser was delivered restricted to the eating phase of a hunting task (panel 3e below). Amount of cricket consumed was not reported. However, Yu et al.,⁷ reported that photo inhibition during chase or attack phases impaired latency to attack, chase rounds, attack attempts, attack segments and successful capture. The authors show no data regarding cricket eating when stimulation was performed during those phases. However, it's plausible to assume that because successful capture was impaired, mice would have consumed less cricket.

We did not use this metric because typically after capture, the cricket is dismembered in many small pieces and it is not possible to accurately evaluate the amount of cricket mass leftover. Thus, it is not feasible to measure the amount (in mg) of cricket that is eaten. This is why instead of 'cricket consumption' we reported "probability of successful predation".

Similarly to us, Yu et al., 2021⁷ also did not report a direct metric of cricket consumption. The relevant figure from Yu et al., 2021⁷ is pasted below. The closest metric related to cricket consumption in the figure below, in panel **3h** is "% successful capture". They also mentioned eating duration on panel **3e** (see explanation below). Their data shows that inhibition of IPAG vgat cells decreases "% of successful capture" of crickets. While they do not report on 'cricket consumption' *per se*, it is likely that a reduction in "% successful capture" would cause a decrease in consumption, as only captured crickets can be consumed.

We also do not report a direct measure of cricket consumption (which presumably would be measured in mg of cricket eaten). Our figure plots the same exact measure of Yu et al., 2021⁷ which is "probability of successful predation", and we also show the exact same effect reported by Yu et al.⁷. Thus, both reports are in agreement.

We have, however, measured the amount of walnut eaten, and show that inhibition of these cells decreases the amount of mass (in mg) of walnuts eaten. This result, when considered along with the fact that inhibition of vgat cells decreased predation, strongly suggests that the activity of these cells also decreases mg of crickets eaten, even though this measurement cannot be obtained accurately.

The results of Yu et al., 2021⁷ are in complete agreement with our interpretation of the role of PAG vgat cells. Both our data and Yu et al.⁷ report that PAG vgat cells are more active during approach to food, and that they are inhibited during consumption (see Fig. 2c of both papers). As these cells are not active during consumption, inhibiting them during consumption is not expected to cause any effect because there is very little endogenous PAG vgat activity to inhibit during this behavior (as observed on Yu et al., 2021⁷ panel **3e**). Indeed, accordingly, Yu et al.⁷ argues, like us, that these cells are involved in chase and attack, but not in maintaining ongoing consumption. Since consumption must necessarily happen after chase, decreasing chase by optogenetically inhibiting vgat cells leads to subsequent decreases in consumption. Thus, the downstream effects on consumption are a consequence of the control that vgat cells exert on pre-consumption actions, not in ongoing eating. This view is also supported by our new finding showing that activation of PAG vgat cells only during food-seeking increases consumption even though the stimulation is turned off during eating (Fig. R1 above).

We agree with the Reviewer that it is crucial to accurately define behaviors, so now we have added a more detailed description of how the different phases of cricket hunting were defined.

Figure R4 (This is Figure 3 from Yu et al., 2021⁷). Phase-specific inhibition of LPAG vgat neurons suppresses hunting.

3) The explanation for the discrepancy between this study and CeAGABA-vIPAG pathway (Han et al. 2017) in hunting is that the majority of these inputs target excitatory vglut2 PAG cells is not convincing. No pieces of evidence support that GABAergic inputs from CeA only innervated or preferred to project to vglut2 but not vgat PAG cells.

The explanation we provided previously was originated by Han et al., 2017⁵. That report investigated this exact question directly. This paper first showed, as correctly stated by the Reviewer, that activation of the central amygdala (CeA) projection to the PAG increases hunting. Then, the authors investigated if vglut2 or vgat PAG cells are the relevant downstream population mediating the effects of the CeA on the PAG. The authors reasoned that the CeA input to the PAG was inhibitory, and thus, activating the CeA terminals on the PAG were inhibiting some target population in the PAG, which resulted in more hunting. They then tried to counteract the inhibition of the CeA in the PAG by activating either PAG vglut2 or PAG vgat cells.

To do so, they optically activate the CeA to PAG projection, while simultaneously activating either vglut2 or vgat cells in the PAG. Han et al., 2017⁵ reports that increases in hunting elicited by optogenetic activation of the CeA projection to the PAG were blocked by chemogenetic activation of PAG vglut2. They further state that “This is consistent with CeA terminals inhibiting their VGlut2-expressing target cells in PAG”.

The complete description of the relevant text from Han et al., 2017⁵ is pasted below in green for convenience:

“To assess the functional relevance of these CeA → PAG projections, vgat-ires-Cre mice were transfected with non-Cre-dependent ChR2 in CeA and optical fibers placed immediately above the CeA neuronal terminals in LPAG/VLPAG. In the same animals, the Cre-inducible excitatory chemogenetic construct AAV-hSyn-HA-

hM3D(Gq)-IRES-mCherry was injected into the PAG of both *vgat-ires-Cre* and *VGlut2-ires-Cre* mice (Figures S5F and S5G). CeA → PAG optical activation enhanced predatory hunting (Movie S6). Specifically, CeA → PAG optical activation increased pursuit velocities (Figure 5D) and shortened both latency to pursue and overall hunting duration (Figures 5E and 5F)."

"To counter the inhibitory effects of CeA on PAG neurons, we combined optical stimulation with administration of the designer drug CNO in both *VGlut2-ires-Cre* and *vgat-ires-Cre* mice. We found that all of the hunting-promoting effects produced by optical stimulation were annulled by CNO injections in *VGlut2-ires-Cre* mice (Figures 5D–5F). This is consistent with CeA terminals inhibiting their *VGlut2*-expressing target cells in PAG (Figure 5G)."

"Figure 5 legend: "(D–F) Optical activation of CeA/PAG projections elicited moderately faster prey pursuit ($n = 5$, paired t test $**p = 0.03$, D), much shorter latencies to pursuit ($**p = 0.006$, E), and more efficient hunting ($**p = 0.02$, F). However, all these effects were totally annulled by chemogenetic activation of *VGlut2* neurons in PAG."

We did not claim that there is "evidence support that GABAergic inputs from CeA only innervated or preferred to project to *vglut2* but not *vgat* PAG cells.". We did not quantify the number of post synaptic *vgat* or *vglut2* cells in the PAG that receive CeA input. We, just like Han et al., 2017⁵ claimed that activation of the inhibitory projection of the CeA to the PAG can induce hunting by inhibiting *vglut2* PAG cells.

4) The explanation for the discrepancy between this study and Hao et al. 2019 is that *gad2* and *vgat* l/vPAG cells might be two different cell populations. However, they still have a large number of overlapping cells (*gad2* positive and *vgat* positive). The opposite role of *gad2* and *vgat* l/vPAG cells in feeding regulation is not fully addressed.

Although both *vgat* and *gad2* cells are commonly used as GABAergic markers, recent emerging evidence indicates that these two populations have much less overlap than previously assumed. Prior work in regions such as the medial preoptic area and the lateral hypothalamus have shown that a large fraction of *gad2* cells do not co-express *vgat*^{8,9}. These data indicate that *vgat* is a more selective GABAergic marker, as a large fraction of *gad2*, but not *vgat* cells, are glutamatergic^{8,9}. Accordingly, Moffitt et al., 2018⁸ state that "we observed expression of the γ -aminobutyric acid (GABA) synthetic genes *Gad1* and *Gad2* in many excitatory neuronal clusters classified on the basis of expression of *Vglut2* (*Slc17a6*), with *Gad2* expression being particularly widespread (fig. S3C). By contrast, very few *Slc17a6*-positive clusters expressed the GABA transporter gene *Vgat* (*Slc32a1*). These data suggest that *Slc17a6* and *Slc32a1* are better discriminators for excitatory versus inhibitory neurons, corroborating evidence from other brain areas (32)."

Our data adds to the results above, and show that in the PAG the majority of *gad2* cells are not *vgat* cells. Thus, it is not unexpected that activating *gad2* cells produces different results from activating *vgat* cells. The population that Hao et al., 2019³ activated consisted of 42% of *vgat*+ cells and 58% of *vgat*- cells (Supplementary Fig. 10). It is entirely possible that these 58% of cells induce a different effect. These 58% of cells would constitute likely thousands of cells that we are not activating in our study of *vgat* cells. Recent data show that activation of even a very small number of cortical cells, such as twenty ($n=20$), is sufficient to robustly change behavior¹⁰. It is, consequently, expected that the activation of thousands of different *gad2*+*vgat*-cells by Hao et al.³ may produce different results from ours.

Results conceptually similar to ours have been observed previously. For example, activation of *PKC- δ* cells in the central amygdala decreases feeding¹¹. These cells are a subset of 50% GABAergic cells in the central amygdala¹². Nevertheless, activation of pan-GABAergic *vgat* cells in the central amygdala increases feeding⁵, even though this *vgat* population has a large overlap with *PKC- δ* the cells. The data above show that activation of two populations of cells may produce completely opposite results, even though the two populations share 50%

overlap. This same pattern of results seen in the central amygdala was also seen in the PAG, where activation of gad2 and vgat cells induce opposite results even though the gad2 cells overlap by 42% with the vgat cells.

Similarly to our study, when inhibiting PAG vgat neuronal population, Yu et al., 2021⁷ didn't observe an increase in consummatory behavior as Hao et al., 2019³.

Figure R3 (Supplementary Fig. 10). The majority of PAG gad2-expressing cells do not co-express vgat. (a) Analysis of the MERFISH data from Vaughn et al., 2022¹³ quantifying the number of gad2, vgat and gad2*vgat expressing cells in the PAG per mouse. (b) Note that the vast majority (92%) of vgat cells co-express gad2. (c) Graph showing that 42% of gad2-expressing cells also co-express vgat. Data are plotted as mean (+/- s.e.m.). n=784,863 gad2+, 366,011 vgat+, 342,886 vgat.gad2, n=31 mice.

5) The authors explained the discrepancy between this study and prior studies (Zhang et al. 2017; Zhao et al. 2019) showing that activation of ZI GABAergic cells promotes food seeking by that the functional diversity of ZI cells and vPAG vgat cells may decrease food consumption through modulating ZI anxiety inducing cells or local inhibitory interneurons. The authors proposed that both anxiety-inducing cells, as well as inhibitory interneurons, may be putative downstream targets of the inhibitory PAG vgat projection. Inhibition of either of these populations by PAG vgat cells would decrease food-seeking. However, this study concludes that activation of PAG GABAergic inputs to ZI, that is inhibition of ZI, promotes food-seeking and food consumption, which is controversial to the authors' own hypothesis.

We apologize for not clearly explaining our model. Prior data show that the majority of ZI cells are GABAergic vgat+ cells, and that bulk excitation of these ZI vgat cells increases feeding^{1,2}. However, the ZI is highly heterogeneous, and distinct cells have shown to have different functions¹⁴. The ZI also has local inhibitory interneurons¹⁴. Local interneurons in the ZI are thus expected to inhibit principal projection ZI cells that induce eating.

Scheme 1. Model depicting how PAG vgat cell may increase food-seeking. PAG vgat cells (blue) may release the inhibitory transmitter GABA (red dots) onto zona incerta (ZI) GABAergic local interneurons (green). Inhibition of the ZI interneuron, in turn will decrease release of GABA by ZI interneurons onto ZI projection cells that induce food-seeking (brown). Thus, activation of PAG vgat cells may decrease inhibition supplied by ZI interneurons onto ZI food-seeking cells, resulting in disinhibition. Disinhibition of ZI food-seeking cells is expected to increase foraging behaviors.

Excitation of either anxiety-inducing ZI cells or inhibitory interneurons in the ZI is expected to decrease feeding. Thus, logically, inhibition of anxiety-inducing ZI cells or of ZI interneurons are expected to increase feeding. We previously proposed that the inhibitory PAG vgat input is inhibiting these cell types, which would be expected to increase feeding. Now, in this revised submission, in order to simplify the discussion, we have deleted any mention to ZI anxiety cells being the downstream target of PAG vgat neurons. In the current manuscript we only mention ZI interneurons as a potential downstream target of PAG vgat cells.

As the Reviewer correctly states, we proposed inhibitory interneurons may be putative downstream targets of the inhibitory PAG vgat projection". However, we disagree that "Inhibition of this population by PAG vgat cells would decrease food-seeking" as stated by the Reviewer. As we discussed above, inhibition of ZI inhibitory interneurons is expected to increase feeding. Thus, inhibition of these cells by the inhibitory PAG vgat input is also expected to induce feeding by causing disinhibition of ZI principal cells (see explanatory diagram on Scheme 1 above). This disinhibitory mechanism is identical to the one proposed by Tovote et al⁴ in the freezing circuit. In that work, Tovote et al⁴ proposes that the GABAergic projection cells from the central amygdala inhibit local GABAergic interneurons in the PAG, resulting in disinhibition of freezing-inducing PAG projection cells.

According to our current view, PAG vgat input poly-synaptically disinhibits eating-inducing cells in the ZI, even though the direct monosynaptic effect of the PAG vgat input is GABA-mediated inhibition (Fig. 6g). This view is supported by Supplementary Fig. 17 (Figure R5 below), which shows that during approach to food both ZI and PAG vgat cells show an increase in activity during approach to food. Since both show an increase in activity during approach to food, it is not possible that the overall effect of the PAG vgat input during food approach is to cause strong inhibition of the majority of ZI cells.

Furthermore, we now provide direct evidence that activation of PAG vgat cells increases ZI vgat activity. To do so, we expressed the red-shifted excitatory opsin ChRmine in PAG vgat cells and GCaMP6f in ZI vgat cells. Optogenetic activation of PAG vgat cells caused time-locked increases in neural activity in ZI vgat cells in freely-moving mice. To ascertain that this result was not due to spectral interference of the optogenetic laser source causing an artifact in GCaMP6f fluorescence measurement, we repeated the same experiment in isoflurane anesthesia. In this condition, optogenetic excitation of PAG vgat cells did not cause changes in ZI neural activity. Thus, these data demonstrate that in freely-moving mice, activation of PAG vgat cells results in higher ZI activity (Supplementary Fig. 18 and Figure R6 below).

Taken together, these new data show that PAG vgat activity is not broadly inhibiting the ZI. Instead, these results are compatible with polysynaptic disinhibition. Other groups have also found support for this idea, as prior data show that average *in vivo* firing rates of ZI cells increase following intra-ZI injections of GABA agonists^{15,16}. Thus, both our data and published data^{15,16} indicate that GABAergic transmission within the ZI *in vivo* can result in polysynaptic disinhibition, which increases the neural activity of the ZI. Consequently, our data suggest that PAG vgat input to the ZI increases feeding by increasing ZI vgat activity through polysynaptic disinhibition. Similar circuit motifs controlling feeding have been reported in other circuits. For example, activation of inhibitory somatostatin+ cells from the lateral septum also induces food approach and the stimulation of inhibitory terminals increases neural activity in downstream regions¹⁷. Similarly, it has been shown that activation of the peri-locus coeruleus increases feeding. However, counterintuitively, activation of the inhibitory lateral hypothalamus vgat projection to the peri locus coeruleus also induces feeding¹⁸. It's been also reported that ZI GABAergic inputs to PAG can excite a subset of PAG neurons¹⁹.

We have added a discussion of these ideas in the main manuscript.

Figure R5 (Supplementary Fig. 17). ZI and I/vIPAG cells are activated during approach to food. Dual photometry recording experiment in vgat-expressing cells from I/vIPAG and ZI. AAV-FLEX-GCaMP6f was injected in the I/vIPAG and ZI of vgat-Cre mice. Recordings of calcium transients were obtained from vgat ZI and vgat I/vIPAG cell bodies from contralateral hemispheres to avoid contamination of the signal from long range axon terminals (**a,b**). The data show that in both regions vgat cells are activated during approach to walnut (**c,d**) and cricket (**e,f**).

Figure R6 (Supplementary Fig. 18) Optogenetic activation of /VIPAG vgat cells increases activity in zona incerta vgat cells. **a** Scheme showing mouse with simultaneous recording of GCaMP6f-expressing zona incerta vgat cells with 589 nm optogenetic excitation of ChRmine-expressing /VIPAG vgat cells. **b** Cre-dependent viral vectors encoding GCaMP6f and the red-shifted opposing ChRmine were injected, respectively, in the zona incerta and /VIPAG of a vgat-Cre mouse. Fiberoptic cannulae for fiber photometry and optogenetic excitation were, respectively implanted over the zona incerta and the /VIPAG. **c** Upper panel: Activity from GCaMP6f-expressing vgat zona incerta cells in a representative freely-moving mice with or without optogenetic stimulation of ChRmine-expressing /VIPAG vgat cells. Simulation epochs (Light ON) are indicated by shaded orange rectangles. Lower panel: Same as upper panel, but with recordings from the same representative mouse under isoflurane anesthesia. Note that optogenetic excitation of PAG vgat cells increased ZI vgat neural activity in freely-moving mice, but not in isoflurane anesthetized animals, demonstrating that the result was not due to the ChRmine excitation light causing spectral interference in GCaMP6f fluorescence. **d** Average activity in zona incerta vgat cells (plotted as light ON-light OFF). Note that optogenetic stimulation of /VIPAG vgat cells increases activity in zona incerta vgat cells in freely moving mice (blue), but not in isoflurane anesthetized mice (green). (n=4, paired t-test, $p = 0.025$, t-statistic = 4.18)

6) The answer to the question of what activation sequencing between ZI is and vIPAG during predation is the cornerstone of this study. Because this question could answer whether it is indeed the bottom-up modulation of feeding from vIPAG to ZI that exist. Current optogenetic manipulation of /VIPAG vgat-ZI terminals data could not rule out the possibilities of retrograde activation or inhibition of vIPAG cells.

According to our understanding, the Reviewer states that optogenetic manipulations of axon terminals can induce both retrograde activation and retrograde inhibition, as the Reviewer shows concern about “retrograde activation or inhibition of vIPAG cells.”

The Reviewer is correct in asserting that optical activation of axon terminals can generate retrograde action potentials. We have added text to the discussion mentioning this important caveat. These back-propagating action potentials are possible because the depolarization of the action potential is actively propagated by opening of voltage-gated sodium channels along the axon.

However, to our knowledge it is not biophysically possible for inhibition of long range terminals to generate back-propagating inhibition. There is no known way for optically-induced inhibition of the axon potential to be actively propagated from the axon terminal to the cell body. Thus, we respectfully disagree with the Reviewer, and we do not believe that optogenetic inhibition of terminals in ZI is confounded by retrograde inhibition of I/vIPAG cells. If such a mechanism of antidromic inhibition exists, we ask the Reviewer to please provide a reference showing that optical inhibition can generate antidromic retrograde inhibition that travels over long distances from the axon terminal to the cell body.

However, optical inhibition is not confounded by this same retrograde confound, as explained. Thus, the results of optical inhibition can unambiguously be assigned to manipulations done in the terminals. Consequently, we can claim that activity in the PAG vgat projection to the ZI is necessary for feeding.

Additionally, we agree with the Reviewer that it is important to determine the “activation sequencing between ZI and vIPAG during predation”. Our data from Figure R5 (above) clearly shows that PAG vgat activity (Figure R5 c,e in green) increases prior to ZI vgat activity (Figure R5c,e, yellow) during approach to both cricket and food. These data show the endogenous activity dynamics of the circuit are compatible with bottom-up influence from the PAG to the ZI. The data also indicate that the two populations are not mutually inhibiting each other, since both of them show increased activity during approach to food.

To demonstrate if higher PAG vgat activity indeed causes an increase in ZI activity, we expressed the red-shifted opsin ChRmine in PAG vgat cells and GCaMP6f in ZI vgat cells. We now show that activation of PAG vgat cells demonstrably increases activity *in vivo* of ZI vgat cells (Figure R6, or Supplementary Fig.18). However, the monosynaptic effect of this PAG input is inhibitory (Fig. 6g). Consequently, the observed increase in activity in ZI vgat cells following activation of PAG vgat cells is likely due to poly-synaptic disinhibition (though other mechanisms are also possible). Our data thus indicates that the PAG vgat input disinhibits the ZI, increasing ZI activity. This increase in ZI activity may lead to increased food-seeking, in agreement with prior data showing that higher ZI activity increases this behavior. Remarkably, both correlational (Figure R5) and perturbational measurements (Figure R6) support a model in which PAG vgat input increases ZI vgat activity. Since our new data directly shows that activation of PAG vgat cells can increase activity of ZI vgat cells, we posit this the most straight-forward explanation for how the PAG to ZI projection increases feeding. Consequently, due to the adoption of this explanation, we deleted any mention of our previous model with putative anxiety-inducing ZI cells from the text.

Alternatively, it is also possible that when the Reviewer wrote ‘retrograde’ they were referring to retrograde inhibition of viral spread. However, the serotype we used (AAV5) has very low retrograde efficiency, as shown in the figure below from Tervo et al., 2016²⁰. Furthermore, for projection optogenetic assays, the stimulation light was targeted at PAG vgat terminals in the ZI. Thus, the stimulation light would not affect hypothetical retrogradely infected cells upstream of PAG vgat cells. Additionally, we did not observe retrogradely-labeled cell bodies in the ZI, which is a region that projects to the vIPAG vgat cells (Fig. 5e). These data indicate that the effects observed due to optogenetic manipulations of PAG vgat projections to the ZI were not mediated by viral retrograde spread.

Figure R7 (Fig. 2d from Tervo et al., 2016²⁰). Retrograde transport efficiency for different AAV serotypes and for canine adenovirus type 2

Figure R8. AAV5-DIO-YFP injection in the I/vIPAG of *vgat-cre* mice resulted in infected cells in I/vIPAG, but not in zona incerta. A *vgat-cre* mouse was injected with the viral vector AAV5-DIO-YFP in the I/vIPAG, resulting in YFP-expressing cells the I/vIPAG (left panel). (Right panel) Axonal fibers from I/vIPAG *vgat* cells expressing YFP can be seen in the zona incerta. Note the absence of cell bodies within the zona incerta expressing YFP, demonstrating that zona incerta cells were not infected by retrograde transmission of the AAV5 vector injected in the I/vIPAG.

Minor points

1) Video showing optogenetic activation of I/vIPAG *vgat* cells induces hunting is necessary.

We have added the video as requested (see Supplementary Movie 3).

2) A related question, in cricket prey test, does optogenetic activation of l/vIPAG vgat cells cause consumption of cricket?

As explained in question 2 above, we did not quantify the amount of cricket consumed. Our data show that activation of these cells increases the probability of successful capture, and in all cases of successful capture we observed cricket consumption (see video added), though the amount of cricket eaten (measured in mg) could not be accurately assessed because the cricket is dismembered in many pieces while being consumed. Thus, as explained (see answer to major points, question 2 above), it is not practical to measure mg of cricket consumed. However, we do show that optogenetic activation of l/vIPAG vgat cells increases consumption of walnuts.

References

1. Zhang, X. & van den Pol, A. N. Rapid binge-like eating and body weight gain driven by zona incerta GABA neuron activation. *Science* **356**, 853–859 (2017).
2. Zhao, Z.-D. *et al.* Zona incerta GABAergic neurons integrate prey-related sensory signals and induce an appetitive drive to promote hunting. *Nat. Neurosci.* **22**, 921–932 (2019).
3. Hao, S. *et al.* The Lateral Hypothalamic and BNST GABAergic Projections to the Anterior Ventrolateral Periaqueductal Gray Regulate Feeding. *Cell Rep.* **28**, 616–624.e5 (2019).
4. Tovote, P. *et al.* Midbrain circuits for defensive behaviour. *Nature* **534**, 206–212 (2016).
5. Han, W. *et al.* Integrated Control of Predatory Hunting by the Central Nucleus of the Amygdala. *Cell* **168**, 311–324.e18 (2017).
6. Li, Y. *et al.* Hypothalamic Circuits for Predation and Evasion. *Neuron* **97**, 911–924.e5 (2018).
7. Yu, H. *et al.* Periaqueductal gray neurons encode the sequential motor program in hunting behavior of mice. *Nat. Commun.* **12**, 1–15 (2021).
8. Moffitt, J. R. *et al.* Molecular, spatial, and functional single-cell profiling of the hypothalamic preoptic region. *Science* **362**, (2018).
9. Kosse, C., Schöne, C., Bracey, E. & Burdakov, D. Orexin-driven GAD65 network of the lateral hypothalamus sets physical activity in mice. *Proc. Natl. Acad. Sci. U. S. A.* **114**, 4525–4530 (2017).
10. Jennings, J. H. *et al.* Interacting neural ensembles in orbitofrontal cortex for social and feeding behaviour. *Nature* **565**, 645–649 (2019).

11. Cai, H., Haubensak, W., Anthony, T. E. & Anderson, D. J. Central amygdala PKC- δ (+) neurons mediate the influence of multiple anorexigenic signals. *Nat. Neurosci.* **17**, 1240–1248 (2014).
12. Haubensak, W. *et al.* Genetic dissection of an amygdala microcircuit that gates conditioned fear. *Nature* **468**, 270–276 (2010).
13. Vaughn, E., Eichhorn, S., Jung, W., Zhuang, X. & Dulac, C. Three-dimensional Interrogation of Cell Types and Instinctive Behavior in the Periaqueductal Gray. *bioRxiv* 2022.06.27.497769 (2022)
doi:10.1101/2022.06.27.497769.
14. Li, Z., Rizzi, G. & Tan, K. R. Zona incerta subpopulations differentially encode and modulate anxiety. *Sci Adv* **7**, eabf6709 (2021).
15. Moon, H. C. & Park, Y. S. Reduced GABAergic neuronal activity in zona incerta causes neuropathic pain in a rat sciatic nerve chronic constriction injury model. *J. Pain Res.* **10**, 1125–1134 (2017).
16. Moon, H. C., Lee, Y. J., Cho, C. B. & Park, Y. S. Suppressed GABAergic signaling in the zona incerta causes neuropathic pain in a thoracic hemisection spinal cord injury rat model. *Neurosci. Lett.* **632**, 55–61 (2016).
17. Carus-Cadavieco, M. *et al.* Gamma oscillations organize top-down signalling to hypothalamus and enable food seeking. *Nature* **542**, 232–236 (2017).
18. Marino, R. A. M. *et al.* Control of food approach and eating by a GABAergic projection from lateral hypothalamus to dorsal pons. *Proceedings of the National Academy of Sciences* **117**, 8611–8615 (2020).
19. Ahmadlou, M. *et al.* A cell type–specific cortico-subcortical brain circuit for investigatory and novelty-seeking behavior. *Science* **372**, eabe9681 (2021).
20. Tervo, D. G. R. *et al.* A Designer AAV Variant Permits Efficient Retrograde Access to Projection Neurons. *Neuron* **92**, 372–382 (2016).

REVIEWERS' COMMENTS

Reviewer #3 (Remarks to the Author):

The rebuttal has partially addressed some of the issues. Although additional experiments and data demonstrate a clear intention to improve the conclusiveness of the study, some points raised by the reviewer remain unanswered. Therefore, it is difficult for this reviewer to support the publication of the study in its current form in Nature Communications.

Major points

- 1) The biggest concern of this review regards the authors' claim that l/vIPAG vgat cells innervated to ZI interneurons, and subsequently disinhibited ZI food-seeking cells to promote feeding. Although the new fiber-photometry data showing that activation of PAG vgat cells demonstrably increases the activity of ZI vgat cells in vivo (Figure R6, or Supplementary Fig.18) is nice, no substantial evidence to support the existence of the pathway proposed by the authors. The lack of direct evidence (either morphological or functional) to prove the preference of l/vIPAG vgat to ZI interneurons, but not ZI projection cells weakens the conclusion of a role of l/vIPAGvgat to ZI pathway in feeding regulation.
- 2) The authors use MERFISH data to support the concept that gad2 and vgat l/vIPAG cells might be two different cell populations in l/vIPAG. However, it is unclear whether those gad2+/vgat- cells are evenly distributed in the whole PAG, including the l/vIPAG, or have a specific distribution pattern. Alternatively, the variance of this study with previous reports could be a result of exact manipulation site differences. In addition, there is no fiber track in the sample image of Fig 3a, indicating this image is not obtained from an experimental animal.
- 3) In response to the reviewer's concern regarding retrograde activation or inhibition of vIPAG cells during optogenetic manipulation of l/vIPAGGABA to ZI pathway, the authors claimed "However, to our knowledge it is not biophysically possible for inhibition of long range terminals to generate backpropagating inhibition. There is no known way for optically-induced inhibition of the axon potential to be actively propagated from the axon terminal to the cell body." This statement is not correct since enhanced mosquito homolog of the vertebrate encephalopsin (ePON3) is an effective tool to selectively inhibit axonal terminal synaptic release without affecting the action potential production of soma1.

References

- 1 Mahn, M. et al. Efficient optogenetic silencing of neurotransmitter release with a mosquito rhodopsin. *Neuron* 109, 1621-1635.e1628, doi:10.1016/j.neuron.2021.03.013 (2021).

REVIEWERS' COMMENTS

Reviewer #3 (Remarks to the Author):

The rebuttal has partially addressed some of the issues. Although additional experiments and data demonstrate a clear intention to improve the conclusiveness of the study, some points raised by the reviewer remain unanswered. Therefore, it is difficult for this reviewer to support the publication of the study in its current form in Nature Communications.

Major points

1) The biggest concern of this review regards the authors' claim that l/vIPAG vgat cells innervated to ZI interneurons, and subsequently disinhibited ZI food-seeking cells to promote feeding. Although the new fiber-photometry data showing that activation of PAG vgat cells demonstrably increases the activity of ZI vgat cells in vivo (Figure R6, or Supplementary Fig.18) is nice, no substantial evidence to support the existence of the pathway proposed by the authors. The lack of direct evidence (either morphological or functional) to prove the preference of l/vIPAG vgat to ZI interneurons, but not ZI projection cells weakens the conclusion of a role of l/vIPAGvgat to ZI pathway in feeding regulation.

We appreciate the reviewer's comment and indeed, our work indicates the need of further studies investigating the mechanisms that underlie the connectivity and functions between inhibitory GABAergic circuits that control feeding, like the bottom-up PAG vgat projections. A full understanding of the PAG → ZI projection, among the other GABAergic → GABAergic circuits that control feeding still demand more mechanistic evidence on how they function at the cellular and circuit level.

We also note that our main goal with our newly included data was not to show the existence of a projection from PAG vgat cells to ZI interneurons. Instead, the goal was to resolve an important discrepancy noted by the Reviewer. Prior data showed that activation of ZI vgat cells caused more food-seeking. We showed that activation of the GABAergic vgat PAG projection to the ZI induced food-seeking. These two results are in contradiction, as the PAG vgat projection produces monosynaptic inhibition onto the ZI. Our new data show that activation of PAG vgat cells counterintuitively increases ZI vgat activity, This shows that our result is not in disagreement with prior data because exciting the PAG vgat projection to the ZI increases ZI vgat activity and elicits feeding, thus it is now in agreement with prior data showing ZI activation increases feeding. We argue that the most straight-forward model in which activation of PAG vgat cells would increase ZI activity is via disinhibition mediated by ZI interneurons. Of course, other mechanisms can also explain the observed ZI activation, such as PAG vgat cells co-releasing an unidentified excitatory peptide.

2) The authors use MERFISH data to support the concept that *gad2* and *vgat* I/vIPAG cells might be two different cell populations in I/vIPAG. However, it is unclear whether those *gad2*⁺/*vgat*⁻ cells are evenly distributed in the whole PAG, including the I/vIPAG, or have a specific distribution pattern. Alternatively, the variance of this study with previous reports could be a result of exact manipulation site differences. In addition, there is no fiber track in the sample image of Fig 3a, indicating this image is not obtained from an experimental animal.

Future studies systematically comparing the manipulation of different I/vIPAG sites in both *vgat* and *gad2* *cre* mice could elucidate differences in their spatial expression patterns in detail. Performing a stereological quantification of overlap of these two populations across the entire volume of the PAG is outside the scope of this work. The main point of our work is to show that the PAG *vgat* projection to ZI elicits food-seeking. This main result does not depend on the specific spatial distribution signatures of *gad2*⁺/*vgat*⁻ cells.

Despite not displaying the tract of the optic fiber in the image, the sample image of Fig 3a was indeed obtained from an experimental animal. Often the angle at which the brain was sliced is not perfectly parallel to the plane of the fiber, and in those cases the fiber tract may not be prominently visible. We have now added a new histology sample that displays the fiber's placement.

3) In response to the reviewer's concern regarding retrograde activation or inhibition of vIPAG cells during optogenetic manipulation of I/vIPAG GABA to ZI pathway, the authors claimed "However, to our knowledge it is not biophysically possible for inhibition of long range terminals to generate backpropagating inhibition. There is no known way for optically-induced inhibition of the axon potential to be actively propagated from the axon terminal to the cell body." This statement is not correct since enhanced mosquito homolog of the vertebrate encephalopsin (ePON3) is an effective tool to selectively inhibit axonal terminal synaptic release without affecting the action potential production of soma¹.

References

1 Mahn, M. et al. Efficient optogenetic silencing of neurotransmitter release with a mosquito rhodopsin. *Neuron* 109, 1621-1635.e1628, doi:10.1016/j.neuron.2021.03.013 (2021).

We appreciate the Reviewer's comments about the ePON3 opsin that recruits the Gi/o signaling cascade and that can effectively suppress synaptic output during illumination of presynaptic terminals. However, we couldn't find any evidence in this paper showing that optically-induced inhibition can be back-propagated to the soma, which was the concern raised by the Reviewer regarding retrograde propagation of inhibition. This paper, along with all other papers using different opsins to inhibit axon terminals, supports our point that optical inhibition of axon terminals does not back propagate.

In the previous round of review the Reviewer asked about retrograde back-propagation of optical inhibition of axon terminals. We replied saying that “to our knowledge it is not biophysically possible for inhibition of long range terminals to generate backpropagating inhibition”. The Reviewer then stated that our statement is not correct, which presumably means that according to the Reviewer, optical inhibition of terminals can create retrograde inhibition of the soma. To support their point, the Reviewer then provides a reference, that in the Reviewer’s own words, shows “that optical inhibition of terminals can selectively inhibit axonal terminal release without affecting the action potential production in the soma.” However, this is precisely what we wrote originally, as in our view optical inhibition cannot retrogradely backpropagate. We are truly perplexed as the Reviewer denied their own original concern, and now claims we are wrong while citing a reference that fully agrees with our response.